



# Comparative evaluation of rainfall-runoff modelling against flow duration curves in semi-humid catchments

Daeha Kim[1], Ilwon Jung[2], Jong Ahn Chun[1]

[1]APEC Climate Center, Busan, 48058, South Korea
5  [2]Korea Infrastructure Safety & Technology Corporation, Jinju, Gyeongsangnam-do, 52852, South Korea

*Correspondence to*: Jong Ahn Chun (jachun @apcc21.org)

**Abstract.** Streamflow prediction using rainfall-runoff models has long been a special subject in hydrological sciences, and parameter identification is still challenging in ungauged catchments. In this study, we comparatively evaluated predictive performance of rainfall-runoff modelling against the flow duration curve (FDC), which is gaining attention as signature-based parameter identification, by comparing it with conventional hydrograph-based approaches for gauged and ungauged catchments. Using a parsimonious model GR4J under a Monte-Carlo framework, we conducted rainfall-runoff modelling against observed hydrographs and empirical FDCs for 45 gauged catchments in South Korea. By treating each catchment as ungauged, the parameter calibration against regional FDCs was compared with the proximity-based parameter regionalisation in terms of hydrograph and flow signature reproducibility. Results showed that the FDC calibration could lead to noticeably weaker performance and higher uncertainty in predictions in gauged catchments due to the absence of flow timing. The calibration against regional FDCs, which were estimated by a geostatistical method, also showed weaker performance than the proximity-based parameter regionalisation. A relative merit of the FDC calibration was high performance in predicting low flows. From the evaluation of signature reproducibility, it is suggested that metrics describing flow dynamics such as the rising limb density should be added as complementary constraints for improving rainfall-runoff modelling against FDCs.

## 1 Introduction

The runoff hydrograph, a time series of streamflow, is the basis for practical resource management tasks such as water resource allocations, designing infrastructures, flood and drought forecasting, environmental impact assessment (Westerberg et al., 2014; Parajka et al., 2013). It is essential information for investigating physical controls of catchment functional behaviours because a hydrograph aggregates processes interacting within a catchment. Prediction of the runoff hydrograph has long been an important subject in hydrological sciences and is gaining increasing attention with growing concerns about environmental changes (Blöschl et al., 2013). Runoff prediction in ungauged sites has already been a special topic in hydrological sciences, e.g., a decade-long project, Prediction in Ungauged Basins (PUB) by the International Association of Hydrological Sciences (see http://iahs.info/pub/biennia.php). However, predicting hydrographs is still challenging due to





poor data availability and inadequate knowledge about complex catchment responses (Zhang et al., 2015; Blöschl et al., 2013).

A standard method for predicting daily streamflow is to employ a rainfall-runoff model that conceptualises catchment functional behaviours and simulates synthetic hydrographs from atmospheric forcing inputs (Blöschl et al., 2013; Wagener and Wheater, 2006). A prerequisite of this conceptual modelling approach is parameter identification to enable the model to imitate actual catchment responses, and is commonly achieved via calibration against observed hydrographs (referred to as the hydrograph calibration hereafter). On one hand, the hydrograph calibration provides convenience to modellers because reproducibility of the predictand (i.e., the runoff time series), which is typically taken as a performance measure, can be automatically achieved. The use of the hydrograph reproducibility for parameter identification and its validity check has a typical approach for rainfall-runoff modelling (see Hrachowiz et al., 2013). The hydrograph calibration, on the other hand, can be challenged by epistemic errors in input and output data, sensitivity to calibration criteria, and inability of parameter calibration under no or poor data availability (Westerberg et al, 2011; Zhang et al., 2008). Importantly, it is difficult to know whether or not parameters from the hydrograph calibration are unique to represent actual catchment responses since multiple parameter sets would show similar hydrograph reproducibility (Beven, 2006). This low uniqueness of calibrated parameter sets, namely the equi-finality problem in rainfall-runoff modelling, can become a significant uncertainty source when extrapolating parameters to ungauged catchments (Oudin et al., 2008).

To overcome or circumvent those disadvantages of the hydrograph calibration, one can identify the parameters with distinctive flow signatures, i.e., metrics or auxiliary data representing catchment behaviours (referred to as the signature calibration hereafter). The signature calibration is a good alternative to the hydrograph calibration when suitable model parameters are not easily obtained with observed hydrographs alone. Hingray et al. (2010), for instance, calibrated a runoff model with specific flow signatures relevant to its parameters such as snow accumulation and ablation, recession curves, and rising limb, and subsequently found enhanced performance in hourly runoff prediction in Alpine catchments. Yadav et al. (2007) used spatially extrapolated flow metrics for parameter identification, and found major streamflow indices related to catchment functional behaviours. Euser et al. (2013) proposed a framework for structuring a flexible perceptual model with multiple hydrograph signatures, and evaluated model plausibility. Other examples include the use of remotely-sensed geomorphological metrics (Fang et al., 2010), isotope concentrations (Son and Sivapalan, 2007), the baseflow index (Bulygina et al., 2009), the spectral density of streamflow observations (Winsemius et al., 2009; Montanari and Toth, 2007), and long-term hydrograph descriptors (Shamir et al., 2005).

In particular, the flow duration curve (FDC) has received great attention as a calibration criterion that can fit model parameters to catchment functional behaviours (e.g., Westerberg et al., 2011; 2014). The FDC, the relationship between the frequency and flow magnitudes, provides a summary of temporal streamflow variations at the outlet of a catchment (Vogel and Fennessey (1994). It has been useful for numerous hydrological applications. Vogel and Fennessey (1995) exemplified potential uses of FDCs in hydrological studies including wetland inundation mapping, lake sedimentation studies, instream flow assessment, hydropower feasibility analysis, contaminant and waste management, water resources allocation, and flood



frequency analysis. FDCs have been extensively used for runoff prediction (Zhang et al., 2015; Kim and Kaluarachchi, 2014; Smkhtin and Masse, 2000), land use change assessment (Zhao et al., 2012), design of power plants (Liucci et al, 2014), water quality evaluation (Morrison and Bonta, 2008), and catchment classification (Sawicz et al., 2011) among many variations. Along with those applications, FDCs or metrics from FDCs (e.g., the slope of FDCs) were often used as a single
calibration criterion (e.g., Westerberg et al., 2014, 2011; Yu and Yang, 2000) or one of constraints (e.g., Pfannerstill et al., 2014; Kavetski et al., 2011; Hingray et al., 2010; Blazkova and Beven, 2009; Son and Sivapalan, 2007; Yadav et al., 2007) for identifying behavioural parameters. The rationale behind the model calibration against FDCs is that the catchment functional behaviours can be captured by the shape of FDCs (Vogel and Fennessey, 1995; Yokoo and Sivapalan, 2011). This hypothesis also made it possible to apply runoff models to FDC prediction (Zhang et al., 2014; Yokoo and Sivapalan, 2011)
or investigation of physical controls of FDCs (e.g., Ye et al., 2012) in an inverse manner.

For prediction in ungauged catchment, the parameter calibration against FDCs (referred to as the FDC calibration hereafter) provides practical advantages in comparison to conventional parameter regionalisation. The parameter regionalisation, i.e., transferring calibrated parameters from gauged to ungauged catchments (e.g., Kim and Kaluarachchi, 2008; Parajka et al., 2007; Wagener and Wheater, 2006; Dunn and Lilly, 2001), has a critical concern of over-reliance on behavioural parameters
of gauged catchments. Although a priori parameter estimates of ungauged catchments are conveniently achieved by the parameter regionalisation, they are indirectly derived from modelling results at gauged sites with the equi-finality problem. Thus, regionalised parameters could be insufficiently reliable and highly uncertain (Oudin et al., 2008; Zhang et al., 2008; Bárdossy, 2007). To circumvent those drawbacks of the parameter regionalisation, the FDC-based calibration possibly becomes a good alternative. A number of studies have proposed regional models for predicting FDCs at ungauged sites
through regression analyses between quantile flows and catchment properties (e.g., Shu and Ouarda, 2012; Mohammoud, 2008; Smakhtin et al., 1997), geostatistical interpolation of quantile flows (e.g., Pugliese et al., 2014; Westerberg et al., 2014), and regionalisation of theoretical probability distributions (e.g., Atieh et al., 2017; Sadegh et al., 2016). In general, FDCs predicted by those regional models (referred to as the regional FDCs hereafter) well agreed with empirical FDCs; hence, the model calibration with regional FDCs was already applied and showed promising predictive performance for
ungauged catchments (e.g., Westerberg et al., 2014; Yu and Yang, 2000). The parameter identification against regional FDCs was useful even for gauged catchments in the cases of observed hydrographs with poor quality or no overlap between climatic inputs and hydrographs. Importantly, it may be more reliable than the parameter regionalisation because flow information of the catchment of interest, albeit predicted, is directly used to find behavioural parameter sets.

However, several questions arise when using the FDC calibration for gauged and ungauged catchments. First, the FDC is
simplified information with flow magnitudes only; thus, the FDC calibration could worsen the equi-finality and may be more deficient in flow prediction (van Werkhoven et al., 2009). Second, one can cast concerns about uncertainty in regional FDCs possibly introduced by errors in streamflow data and the regional models (Westerberg et al., 2011; Yu et al., 2002). If the calibration with regional FDCs yields highly unreliable quantile flows due to those error sources, it may be less pragmatic than a simple parameter regionalisation. In truth, several studies found that a simple proximity-based parameter transfer well



performed in many regions (e.g., Parajka et al., 2013; Oudin et al., 2008); thus, the calibration against the regional FDCs may be undesirable in the case. Third, there may be additional flow signatures that can improve performance of the FDC calibration. If any flow signatures are found orthogonal to FDCs, additional constraining with those signatures will enable to alleviate the equi-finality of the FDC calibration and thus enhance predictive performance. Nevertheless, it is still an open

question which flow signatures complement FDCs.

This study explored predictive performance of the FDC calibration in rainfall-runoff modelling in comparison with the conventional approaches, the hydrograph calibration and the parameter regionalisation for gauged and ungauged catchments respectively. To answer the questions given, we (1) evaluated predictive performance of the hydrograph calibration and the FDC calibration as well as their uncertainty for gauged catchment, (2) assessed the calibration against regional FDCs in

comparison with the proximity-based parameter regionalisation for ungauged catchments, and (3) gauged ability of the FDC calibration to reproduce typical flow signatures. In this work, a parsimonious 4-parameter conceptual model was used to simulate daily hydrographs from the lumped atmospheric forcing for 45 unregulated catchments in South Korea. To predict FDCs in ungauged catchment, a geostatistical regional model was adopted here. The Monte-Carlo sampling was simply used for parameter identification and uncertainty assessment. The following section presents the study area and data used in our

comparative study.

## 2 The study area and data

The study areas are 45 gauged catchments located across South Korea with no or negligible human-made alterations (e.g., river diversions and dam operations) in flow variations (Figure 1). South Korea is characterized as a temperate and semi-humid climate region with rainy summer seasons. The North Pacific high-pressure brings monsoon rainfall with high

temperatures in summer seasons, while dry and cold weather prevails in winter seasons due to the Siberian high-pressure. Typical ranges of annual precipitation are 1200-1500 and 1000-1800 mm in the northern and the southern areas respectively (Rhee and Cho, 2016). Approximately, 60-70 percent of precipitation falls in summer seasons from June to September (Bae et al., 2008). Streamflow usually peaks in the middle of summer seasons because of heavy rainfall or typhoons, and hence information of catchment responses is largely concentrated on summer-season hydrographs. Snow accumulation and ablation

are observed at high elevations, but their effects on temporal flow variations are minor due to the limited amount of winter precipitation (Bae et al., 2008). Annual temperatures range between 10 and 15 ℃ (Korea Meteorological Administration, 2011).

The study catchments shown in Figure 1 were selected based on availability of streamflow data. Although long streamflow data are available at a few river gauging stations, high-quality streamflow data across the South Korea have been produced

since establishment of the Hydrological Survey Center in 2007 (Jung et al., 2010). We collected streamflow data at 29 river gauging stations from 2007 to 2015 together with inflow data of 16 multi-purpose dams for the same data period from the Water Resources Management Information System operated by the Mistry of Land, Infrastructure, and Transport of the



Korean government (available at http://www.wamis.go.kr/). The selected catchments are listed in Table 1 together with their climatological features.

As the climatic inputs for rainfall-runoff modelling, we used gridded daily precipitation, and maximum and minimum temperatures at a 3-km grid resolution produced by spatial interpolation between 60 stations of the automated surface observing system maintained by the Korea Meteorological Administration. Jung and Eum (2015) combined the Parameter-elevation Regression on Independent Slope Model (Daly et al., 2008) with the inverse distance method for the spatial interpolation, and found improved performance for producing grid precipitation and temperature datasets across South Korea. For simulating streamflow at outlets of the study catchments, we collected the grid climatic data from 2005 to 2015. The ranges of annual mean precipitation and temperature of the selected catchments are 1145–1997 mm and 8.0–13.8 ℃ respectively for the climatic data period. Processing the climatic data for rainfall-runoff modelling will appear later in the methodology section.

## 3 Methodology

In this work, a conceptual rainfall-runoff model, GR4J (Perrin et al., 2003), was adopted to simulate daily hydrographs of the 45 catchments. GR4J conceptualises the functional catchment response to rainfall with four free parameters that regulate the water balance and water transfer functions, and is schematized in Figure 2. The four parameters (X1 to X4) conceptualises soil water storage, groundwater exchange, routing storage, and the base time of unit hydrograph respectively. GR4J is classified as a soil moisture accounting model, and computation details are found in Perrin et al. (2003). Since its parsimonious and efficient structure enables robust calibration and reliable regionalisation of model parameters, GR4J has been frequently used for modelling daily hydrographs with various purposes (e.g., Nepal et al., 2016; Tian et al., 2013). The potential evapotranspiration (PE in Figure 1) in this study was estimated by the temperature-based model of Oudin et al. (2005) proposed for lumped rainfall-runoff modelling.

### 3.1 Preliminary data processing

Before rainfall-runoff modelling with GR4J, we preliminarily processed the gridded climatic data to convert precipitation data to liquid water depths forcing catchments (i.e., rainfall and snowmelt depths) using a physics-based snowmelt model proposed by Walter et al. (2005). The preliminary processing was mainly for reducing systematic errors or bias from no snow component in GR4J, which may affect model efficiencies in catchments at high elevations. Though combining a temperature index snowmelt model with GR4J can be an alternative approach, it increases the number of parameters (i.e., higher equi-finality) and thus model uncertainty. Since contribution of snowmelt to temporal flow variation is insignificant in South Korea as described, maintaining the parsimonious structure of GR4J was considered more importantly for parameter calibration and regionalisation in this work. The error sources in the snowmelt model were assumed to yield minor impacts on runoff prediction. The snowmelt model has the same input requirement as GR4J, thus no additional data are



necessary for the processing. It simulates point-scale snow accumulation and ablation processes using empirical methods that estimate physical parameters required for the energy balance in snowpack, and produces the liquid water depths and snow water equivalent as outputs. After the snowmelt modelling, we took spatially averaged pixel values of the liquid water depths and maximum and minimum temperatures within the boundary of each catchment as lumped inputs to GR4J.

Besides, consistency between the spatially-averaged liquid water depths and observed hydrographs was checked using the current precipitation index (CPI; Smakhtin and Masse, 2000) defined as:

$$I_t = I_{t-1} \cdot K + R_t \tag{1}$$

where $I_t$ is the CPI (mm) at day t, K is a decay coefficient (0.85 d$^{-1}$), and $R_t$ is the liquid water depth (mm d$^{-1}$) at day t that forces the catchment (i.e., rainfall or snowmelt). CPI mimics temporal variations in typical streamflow data by converting

intermittent rainfall data to a continuous time series with an assumption of the linear reservoir. The consistency between model input and output was checked for each catchment using correlation between CPI and observed streamflow as in Westerberg et al. (2014) and Kim and Kaluarachchi (2014). The correlation coefficients of the 45 catchments had an average of 0.67 with a range of 0.43-0.79, and no outliers were found in the box plot of correlation coefficients. Hence, we hypothesised that acceptable consistency existed between the climatic forcing and the observed hydrographs for parameter

calibration.

## 3.2 Rainfall-runoff modelling for gauged catchments

To search behavioural parameter sets of GR4J using observed runoff time series (i.e., the hydrograph calibration), the Monte-Carlo random sampling was used within the parameter ranges given by Demirel et al. (2013). The objective function in Zhang et al. (2015) was chosen as the calibration criterion that considers together the Nash Sutcliffe Efficiency (NSE) and

the Water Balance Error (WBE) between observed and modelled hydrographs as:

$$OBJ = (1 - NSE) + 5|\ln(1 + WBE)|^{2.5} \tag{2a}$$

$$NSE = 1 - \frac{\sum_{i=1}^{N}(Q_{obs,i} - Q_{sim,i})^2}{\sum_{i=1}^{N}(Q_{obs,i} - \overline{Q_{obs}})^2} \tag{2b}$$

$$WBE = \frac{\sum_{i=1}^{N}(Q_{obs,i} - Q_{sim,i})}{\sum_{i=1}^{N} Q_{obs,i}} \tag{2c}$$

where $Q_{obs}$ and $Q_{sim}$ are the observed and simulated flows respectively, $\overline{Q_{obs}}$ is the arithmetic mean of $Q_{obs}$, and N is the total

number of flow observations. The best parameter sets for each study catchment was obtained from minimisation of the OBJ using the Monte-Carlo simulations described below.

To determine sufficient runs for the random simulations, we calibrated GR4J parameters using the shuffled complex evolution (SCE) algorithm (Duan et al., 1992) for one catchment with high input-output consistency. Then, the total number of random simulations was iteratively determined by adjusting the number of runs until the minimum OBJ of the random





simulations became adequately close to the OBJ value from the SCE algorithm. We found that approximately 20,000 runs could provide the minimum OBJ value equivalent to one from the SCE algorithm. Subsequently, GR4J was calibrated by 20,000 runs of the Monte-Carlo simulations for remaining 44 catchments, and the parameter sets with the minimum OBJ values were taken for runoff predictions. In addition, we sorted the 20,000 parameter sets in terms of corresponding OBJ values in ascending order and first 50 sets were taken for uncertainty assessment (i.e., 0.25% of the rejection threshold). For the parameter identification, the 9-year streamflow data were divided into two parts for calibration (2011-2015) and for validity check (2007-2010) respectively. A two-year warm-up period was used for initializing all runoff simulations in this study.

The FDC calibration was also conducted by the same Monte-Carlo sampling but towards minimising OBJ between the observed and modelled quantile flows. We used quantile flows at 103 exceedance probabilities (p of 0.001, 0.005, 99 points between 0.01 and 0.99 at an interval of 0.01, 0.995, and 0.999) to evaluate agreement between the observed and simulated FDCs. As conducted in the hydrograph calibration, the best parameter set was found by 20,000 random simulations and 50 behavioural parameter sets were taken.

### 3.3 Rainfall-runoff modelling for ungauged catchments

Synthetic runoff time series were generated by GR4J for the same 45 catchments by treating each catchment as ungauged. The parameters of ungauged catchments were identified by (a) local calibration against regional FDCs and by (b) transferring the calibrated sets of nearby gauged catchments (i.e., proximity-based parameter regionalisation). Following are descriptions of both approaches.

### 3.3.1 Parameter identification against regional flow duration curves

The geostatistical method recently proposed by Pugliese et al. (2014) was used to regionalise the observed FDCs. Pugliese et al. (2014) employed the top-kriging method (Skøien et al., 2006) to spatially interpolate the total negative deviation (TND), which indicates an area between the mean annual flow and below-mean flows in a normalized FDC. The top-kriging weights that interpolate TND values were used as weights to estimate the flow quantiles of ungauged catchments from empirical FDCs of neighbouring gauged catchments. Since the top-kriging weights are obtained from topological proximity between catchments, the two methods for ungauged catchments in this study are categorised as proximity-based approaches and thus of consistency. The FDC of an ungauged catchment in Pugliese et al. (2014) is estimated from the normalised FDCs of neighbouring gauged catchments as:

$$\widehat{\Phi}(w_0, p) = \widehat{\phi}(w_0, p) \cdot \overline{Q}(w_0) \tag{3a}$$

$$\widehat{\phi}(w_0, p) = \sum_{i=1}^{n} \lambda_i \cdot \phi_i(w_i, p), \quad p \epsilon (0,1) \tag{3b}$$



where $\widehat{\Phi}(w_0, p)$ is the estimated quantile flow (m³ s⁻¹) at an exceedance probability p (unitless) for an ungauged catchment $w_0$, $\widehat{\phi}(w_0, p)$ is the estimated normalized quantile flow (unitless), $\overline{Q}(w_0)$ is the annual mean streamflow (m³ s⁻¹) of the ungauged catchment, and $\phi_i(w_i, p)$ and $\lambda_i$ are the normalized quantile flows (unitless) and corresponding top-kriging weights (unitless) of gauged catchment $w_i$ respectively. The unknown mean annual flow of an ungauged catchment, $\overline{Q}(w_0)$, can be estimated with a rescaled mean annual precipitation defined as:

$$MAP^* = 3.171 \times 10^{-5} \cdot MAP \cdot A \tag{4}$$

where MAP* is the rescaled mean annual precipitation (m³ s⁻¹), MAP is mean annual precipitation (mm yr⁻¹) and A is drainage area (km²) of the ungauged catchment, and the constant of $3.171 \times 10^{-5}$ is to convert the unit of MAP* from mm yr⁻¹ km² to m³ s⁻¹.

A distinct advantage of the geostatistical method is that it enables to estimate the entire flow quantiles in a FDC with a single set of top-kriging weights. Since a parametric regional FDC (e.g., Mohamoud, 2008; Yu et al., 2002) is obtained from independent models for each flow quantile in many cases, e.g., multiple regressions between selected quantile flows and catchment properties, fundamental characteristics in a FDC continuum would be entirely or partly lost. The geostatistical method, on the other hand, treats all flow quantiles as a single object; thereby, features in a FDC continuum can be preserved. It showed promising performance to reproduce empirical FDCs using topological proximity only, and further details and discussion are available in Pugliese et al. (2014).

For regionalising empirical FDCs of the 45 catchments, we followed the same procedure of Pugliese et al. (2014). We obtained top-kriging weights ($\lambda_i$) by the geostatistical interpolation of TND values from empirical FDCs for the calibration period (2011-2015). Then, the top-kriging weights were used to regionalise flow quantiles. The number of neighbours for the TND interpolation was iteratively determined as five at which additional neighbouring TNDs are unlikely to provide better agreement between the estimated and empirical TNDs. FDCs for the calibration period were regionalised with the top-kriging weights of the TND interpolation at the 103 exceedance probabilities. Against the regional FDCs, parameters of GR4J were directly calibrated for each catchment. The parameters were identified in the same manner of 20,000 runs of the Monte Carlo simulations, but towards minimisation of the OBJ value between regional and modelled FDCs.

### 3.3.2 Proximity-based parameter regionalization

As a counterpart of the calibration against regional FDCs, the proximity-based parameter transfer was used. The parameter regionalisation can be classified into three typical categories: (a) proximity-based parameter transfer (e.g., Oudin et al., 2008); (b) similarity-based parameter transfer (e.g., McIntyre et al., 2005); and (c) regression between parameters and physical properties of gauged catchments (e.g., Kim and Kaluarachchi, 2008). Based on its competitive performance and simplicity (Parajka et al., 2013; Oudin et al., 2008), we chose the proximity-based parameter regionalisation.

For prediction in ungauged catchment, five donor catchments chosen for the FDC regionalisation were again used for transferring their parameter sets to each catchment of interest. To be consistent between the two proximity-based approaches,




we synchronised donor catchments. The five runoff simulations were averaged for representing modelled hydrographs for each catchment.

## 3.4 Evaluation of predictive performance and uncertainty

Two performance measures were used to evaluate the model predictive performance. One is NSE in Eq. 2b between the observed and modelled flows and the other is the logarithmic Nash-Sutcliffe Efficiency (LNSE) between the observed and simulated flows. These conventional measures evaluate the reproducibility of high and medium flows (NSE) and low flows (LNSE) respectively. LNSE is defined as:

$$\text{LNSE} = 1 - \frac{\sum_{i=1}^{N}[\ln Q_{obs,i} - \ln Q_{sim,i}]^2}{\sum_{i=1}^{N}[\ln Q_{obs,i} - \ln(\overline{Q_{obs}})]^2} \tag{5}$$

For uncertainty assessment, the lower and upper bounds were drawn at the values of 2.5 and 97.5 percentiles of predicted hydrographs with the collection of 50 parameter sets. Uncertainty in predicted flows was quantified by the area between the lower and upper bounds of simulated hydrographs. We took a ratio of uncertainty of the FDC calibration to that of the hydrograph calibration for each catchment and defined it as the uncertainty ratio. It should be noted that this assessment was not to estimate absolute uncertainty but to measure relative uncertainty gained by replacing a hydrograph with a FDC for model calibration.

We additionally selected three typical flow metrics to evaluate flow signature predictability; the runoff ratio ($R_{QP}$), the baseflow index ($I_{BF}$), and the rising limb density ($D_{RL}$). The three typical signatures describe aridity in a catchment, long-term baseflow contribution, and the flashness of catchment response respectively. They are defined as the ratio of runoff to precipitation, the ratio of long-term baseflow to total runoff, and the inverse of average time to peak as:

$$R_{QP} = \frac{\overline{Q}}{\overline{P}} \tag{6a}$$

$$I_{BF} = \sum_{t=1}^{T} \frac{Q_{B,t}}{Q_t} \tag{6b}$$

$$D_{RL} = \frac{N_{RL}}{T_R} \tag{6c}$$

where $\overline{Q}$ and $\overline{P}$ are the average flow and precipitation during a period, $Q_t$ and $Q_{B,t}$ (m d$^{-1}$) is the total streamflow and the base flow at time t respectively, $N_{RL}$ is the number of rising limb, and $T_R$ is the total amount of time the hydrograph is rising (days). $Q_{B,t}$ can be calculated by subtracting direct flow $Q_{D,t}$ from $Q_t$ as:

$$Q_{D,t} = c \cdot Q_{D,t} + 0.5 \cdot (1 + c) \cdot (Q_t - Q_{t-1}) \tag{7a}$$

$$Q_{B,t} = Q_t - Q_{D,t} \tag{7b}$$



where the parameter c is a value of 0.925 from a comprehensive case study by Eckhardt (2007). Reproducibility of $R_{QP}$, $I_{BF}$, and $D_{RL}$ can be evaluated by the relative absolute bias between the modelled and observed signatures as:

$$D_{FS} = \frac{|FS_{sim} - FS_{obs}|}{FS_{obs}} \tag{8}$$

where $D_{FS}$ is the relative absolute bias, $FS_{sim}$ is a flow signature of the modelled flows, and $FS_{obs}$ is that of the observed flows.

## 4 Results

### 4.1 Streamflow prediction in gauged catchments

The box plots in Figure 3 comparatively show distributions of NSE and LNSE values between the observed and modelled flows. This result clearly indicates that the hydrograph calibration outperformed the FDC calibration in prediction of high flows. The NSEs of the hydrograph calibration were generally greater than those of the FDC calibration for both calibration and validation periods. The FDC calibration was of much wider NSE ranges than the hydrograph calibration, suggesting greater uncertainty in high flow prediction. The prediction results tended to have greater medians of NSEs for the calibration period than the validation period. Because the term NSE was directly used for calibration, the parameter identification could be slightly inclined towards reproduction of high flows for the calibration period. The NSE ranges for the calibration period was smaller than those for validation period (Figure 3b). It implies that the FDC calibration has weaker temporal parameter transferability from one period to another. In low-flow prediction, the FDC calibration showed slightly weaker performance than the hydrograph calibration. Although the LNSE medians of the FDC calibration were comparable to those of the hydrograph calibration, LNSEs of the FDC calibration also showed wider ranges than the hydrograph calibration. The FDC calibration was still likely to yield significant uncertainty in low-flow predictions when parameters were temporally transferred. Unlike the NSE comparison, the median LNSE values did not decrease from the calibration to the validation periods for the both hydrograph and the FDC calibrations. This would imply that the behavioural parameter sets have more temporal consistency in low flows than high flows.

Figure 4 illustrates 1:1 scatter plots between the performance measures and correlation between CPI and observed hydrographs, indicating that consistency between model input and output meaningfully affects predictive performance of rainfall-runoff models. The performance measures were generally in positive relationships with correlation between CPI and observed hydrographs. Adequate input-output consistency seems to be a prerequisite of parameter identification to attain good high-flow predictability especially for the hydrograph calibration. For having 0.6 or higher NSE, the correlation coefficient between CPI and observed flows should be greater than 0.6 approximately. On the other hand, predictability of low flows was achieved with relatively low input-out consistency. LNSEs less than 0.4 were rarely observed than NSEs for both hydrograph and FDC calibrations. Interestingly, the FDC calibration appears to have better predictability in low flows despite the use of NSE for parameter calibration, which is a sensitive measure to high-flow reproducibility. This result



implies that the FDC calibration has some deficiency to capture catchment responses to storm events even with adequate model input-output consistency whereas it performs well for long-term low-flow or baseflow predictions.

Shortly, the FDC calibration could lead to relatively low predictive power with increased uncertainty when adopted as an alternative of the hydrograph calibration. Low predictability in high-flows can be a particular concern of the FDC calibration. The simplification of flow information appears to exacerbate the equi-finality in parameter identification. This weakness of the FDC calibration was confirmed by the uncertainty bounds of modelled hydrographs in Figure 5. The collection of 50 parameter sets from the FDC calibration showed less robust simulations than the hydrograph calibration for the three catchments even though their FDCs were fairly well reproduced by the FDC calibration. For the 45 catchments, the mean NSE between the observed and modelled FDCs was 0.95 when using the FDC calibration. In other words, parameters reproducing observed FDCs generally were less unique to represent catchment functional behaviours than ones reproducing observed hydrographs. The equi-finality in the FDC calibration is likely to become worse with decreasing performance of the hydrograph calibration (Figure 6). On average, uncertainty of predicted hydrographs was doubled for the 45 catchments when the FDC calibration substitutes for the hydrograph calibration. The prediction results from the 45 gauged catchments, hence, suggest that parameter identification with compact information of FDCs could yield weaker performance and less parameter identifiability than the hydrograph calibration.

### 4.2 Geostatistical FDC regionalisation

Figure 7a illustrates the 1:1 scatter plot between the observed and estimated TNDs of the 45 catchments. The correlation coefficient between the empirical and estimated TNDs was 0.56 (equivalent to 0.30 NSE). It is likely that use of annual precipitation for normalising flow quantiles lead to the relatively poor prediction of TNDs. In the original study of the geostatistical method (Pugliese et al., 2014), the TND prediction became poorer (NSE was decreased from 0.81 to 0.60) when using the rescaled annual precipitation instead of the observed mean annual flow. Uncertainty introduced by estimation of mean annual flows might influence predictive power of the geostatistical TND interpolation. Another possible reason is that TND is a complex signature of streamflow regime; yet, it could be descriptive in terms of functional similarity between catchments (Pugliese et al., 2016). It may be difficult to completely capture spatial variation of TNDs with topological proximity only. However, Pugliese et al. (2016) also argued that poor prediction of TND did not automatically result in poor quantile flow predictions. Their comparative study achieved successful FDC predictions for 182 catchments in the United States (0.95 of median NSE) using the top-kriging weights of TNDs in spite of low TND predictability. A further study is recommended to be directed towards effects of TND prediction on the FDC regionalisation. Because it is still unclear whether or not descriptors from FDCs well predict flow quantiles, top-kriging weights of various flow signatures need to be tested for improving the geostatistical FDC prediction as well.

The high performance in FDC prediction with poor TND prediction was replicated in this study. Overall NSE and LNSE values between the observed and predicted flow quantiles of the 45 catchments suggest good applicability of the geostatistical method to the study catchments (Figure 7b). The averages of individual NSEs and LNSEs for each catchment



were 0.83 and 0.91 with standard deviations of 0.25 and 0.11 respectively. The higher LNSEs imply that performance of the geostatistical method is better for low flows. This might be because the top-kriging weights interpolating TNDs were obtained from below-average flows only. No information of above-average flows reflected in TNDs might incline the FDC regionalisation towards low-flow predictions. Low predictive power of the regional FDC model was found at locations with low gauging density. Catchments 4, 10, 35, and 36, which recorded 0.6 or less NSEs, were with no hatching catchments and/or limited adjacent catchments; nonetheless, LNSEs of those catchments were still greater than 0.7. This result was consistent with a finding of Pugliese et al. (2016) that performance of the geostatistical method was highly sensitive to river gauging density. Transferring quantile flows of remote catchments can yield significant errors because functional similarity would not be captured between donor and receiver catchments. Overall, in spite of abovementioned shortcomings, the geostatistical FDC regionalisation was considered to be acceptable and topological proximity would to be a good predictor of FDCs across the study catchments.

### 4.3 Streamflow prediction for ungauged catchments

The box plots in Figure 8 present predictive performance of the calibration against regional FDCs (referred to as RFDC_cal hereafter) in comparison with the proximity-based parameter regionalisation (referred to as PROX_reg hereafter). The performance measures between the observed and modelled hydrographs were computed for the entire period of streamflow data (2007-2011). Distributions of NSEs clearly showed that PROX_reg outperforms the FDC calibration in prediction of high flows (Figure 8a), indicating that a priori parameter sets from neighbouring catchments should perform even better than ones from local calibrations against the observed FDCs. The average difference between NSEs of PROX_reg and RFDC_cal was 0.18 with a standard deviation of 0.25. RFDC_cal outperformed PROX_reg only for 8 out of the 45 catchments. LNSEs with PROX_reg were still of a slightly higher median than RFDC_cal. Although RFDC_cal appears to have comparable predictability in low flows, 31 out of 45 catchments were having greater LNSEs with PROX_reg.

The weaker performance of RFDC_cal in this work is consistent with the comparative study of Zhang et al. (2015), which evaluated performance of RFDC_cal using GR4J in 228 Australian catchments. Zhang et al. (2015) argued that RFDC_cal is not good enough for predicting daily hydrographs in the Australian catchments due to its much worse performance than the hydrograph calibration in gauged catchments. The information loss from simplifying hydrographs can be attributed to weaker performance and higher uncertainty of rainfall-runoff modelling agianst in FDCs. In recognition of good agreement between the empirical and regional FDCs for the study catchments, prediction errors in regional FDCs would exert minor impacts on performance of RFDC_cal.

### 4.4 Evaluation of flow signature reproducibility

Figure 9 summarises performance of the four methods applied in this study to regenerate three flow signatures of $R_{QP}$, $I_{BF}$, and $D_{RL}$. The box plots of absolute biases between the observed and modelled signatures indicate that parameter identification against FDCs showed competitive reproducibility in the long-term signatures $R_{QP}$ and $I_{BF}$, while its ability was



relatively weak to regenerate the event-based signature $D_{RL}$. $R_{QP}$ biases seem to be sensitively affected by additional uncertainty sources in the FDC regionalisation and in spatial and temporal parameter transfer, but their medians and box heights were similar between FDC-based and hydrograph-based approaches. Given their relatively competitive performance in low flows, FDC-based approaches would show strong performance to reproduce $I_{BF}$.

In contrast, the FDC-based approaches were poorer to reproduce the event-based flow signature, $D_{RL}$. It is not surprising because a FDC aggregates information of flow magnitude only. No information of flow timing in FDCs is likely a main factor that resulted in poor predictions of peak flow timing for both gauged and ungauged catchments. The FDC-based approaches could be insufficient for hydrological applications that require specific flow timings (e.g., flood forecasting). The conventional parameter regionalisation would be a more pragmatic option for the Korean catchments. From Figure 9c, we
also had an indication that predictability in peak flow timing of the hydrograph calibration was well preserved even when parameter sets were transferred to neighbouring catchments.

## 5 Discussion

### 5.1 Evaluation of rainfall-runoff modelling against regional FDCs

Regionalised flow signatures have frequently used for constraining rainfall-runoff models (e.g., Bárdossy, 2007; Boughton
and Chiew, 2007; Bulygina et al., 2009). Advantages of the approaches are that they are complementary to a priori estimation of model parameters and are similar to usual methods to directly determine the model parameters from dynamic catchment response data (Blöschl et al., 2013). An important lesson learned from previous studies was that the models would dominantly work for reproducing the flow signature of interest (Blöschl et al., 2013), albeit it appears self-evident. Thus, if one forces the model to reproduce low-flow signatures, use of the model would be appropriate for a drought forecasting
rather than a flood analysis. Likewise, multiple signatures are obviously necessary for constraining runoff models to consider various aspects of flow variation.

In this context, use of a FDC as a single calibration criterion appears to be a great choice for searching model parameters suitable for dynamic catchment behaviours. A FDC is a compact representation of runoff variability in frequency domain at all time scales from inter-annual to event-scale, and thus it embeds various aspects of multiple flow signatures (Blöschl et al.,
2013). A pilot study of Yokoo and Sivapalan (2011) discovered that the upper part of a FDC with high flows is controlled by interaction between extreme rainfall and fast runoff, while the middle and lower parts are governed by interactions between water availability, energy and water storage and by baseflow recession behaviour during dry periods respectively. The major hydrological processes within a catchment are reflected in a FDC, and therefore a runoff model constrained by a FDC can be expected to provide reliable flow predictions. The studies of Westerberg et al. (2014, 2011) and Yu and Yang (2000) are
successful examples that applied FDCs to rainfall-runoff modelling as a single calibration criterion.

The comparative evaluation in this study, however, provides a lesson that rainfall-runoff modelling against FDCs sufficiently reproduces the FDC itself, but it was insufficient to be comparable to the hydrograph calibration in gauged catchments. For





41 out of the 45 catchments, NSEs between the observed and modelled FDCs were greater than 0.9; nonetheless, hydrograph reproducibility of the FDC calibration was generally weaker. The hydrograph is an output of numerous hydrological processes interacting within a catchment, and is regarded as the most complete flow signature (Blöschl et al., 2013). Since any simplification of the hydrograph including FDCs would lose some amount of flow information, it is no surprise that the

5 FDCs calibration worsens the equi-finality problem in conceptual rainfall-runoff modelling. If one has a runoff time series with acceptable data quality and length, there should be no reason to adopt the FDC calibration in replacement of the hydrograph calibration. The weaker $D_{RL}$ reproducibility confirms that the absence of flow timing in FDCs would lead to poorer runoff predictions of the FDC calibration. Instead, the FDC calibration may successfully predict compact flow signatures which are less informative than FDCs (e.g., mean annual runoff and seasonal flow regime).

For ungauged or poorly gauged catchments, on the other hand, rainfall-runoff modelling against regionalised FDCs (RFDC_cal) can bring advantages. As aforementioned, a priori parameter sets derived from the outside of a catchment of interest may be more uncertain and thus less reliable than ones achieved from independently predicted flow signatures. Nevertheless, RFDC_cal was less powerful than use of parameter sets transferred from neighbouring catchments despite well-regionalised FDCs. The deficiency in RFDC_cal was likely to come not only from the absence of flow information in

FDCs, but from powerful performance of PROX_reg. Modelling conditions of this study were very suitable for the proximity-based parameter transfer based on an extensive comparative study of Parajka et al. (2013). Parajka et al. (2013) reported that parameter regionalisation generally showed higher NSE performance under humid conditions than in arid and tropical regions. They argued that PROX_reg can be competitive with or better than similarity-based and regression-based regionalisation (e.g., Oudin et al., 2008; Parajka et al., 2005). Parajka et al. (2013) also provided a relationship between

model complexity and performance, indicating that the complexity of GR4J (4 parameters) used in this study was desirable for parameter regionalisation. Given the knowledge in Parajka et al. (2013), aridity and temperature conditions of the 45 study catchments were suited to provide good predictive performance with PROX_reg. The strong performance of PROX_reg in this study suggests that functional similarity between Korean catchments may be changing gradually in space and thus found with spatial proximity. This could be confirmed by good performance of the geostatistical FDCs

regionalisation in this study. Under these conditions, it may be difficult to produce better predictions using RFDC_cal with much higher equi-finality.

### 5.2 Why the FDC calibration performs good for low flow prediction

Although we showed its weaknesses, this paper is not intended to leave negative messages on hydrological modelling against FDCs. It should be emphasised that the FDC calibration may provide advantages for applications aiming at assessing long-

30 term flow regime under projected environmental conditions (e.g., climate change impact assessment). In particular, its powerful predictability in low flows needs to be underlined. The objective function used in the parameter calibration includes the NSE, which can lead to overemphasis on high or peak flows due to squared residuals (Hrachowitz et al., 2013), albeit it is combined with the WBE. The calibration against FDCs, however, well reproduced low flows and $I_{BF}$ with no





logarithmic transformation of observed flows, and hence could be a good alternative for a low flow analysis or a long-term water resources management in both gauged and ungauged catchments.

In regard of flow variation condensed into quantile flows of a FDC, predictability of the FDC calibration may be explained. In Korean catchments under a typical monsoonal climate, low flows governed by baseflow during dry seasons have less

temporal variation than high flows generated by intermittent storm events. Thus, information loss of low flows is much smaller than high flows when a hydrograph is summarised in frequency domain. Figure 10a and b illustrate that high flows modelled by the collection of 50 parameter sets have flow timing errors and low robustness in medium to high flows in spite of fairly good agreement between observed and modelled FDCs across all flow magnitudes. The ranges of baseflow and direct runoff (i.e., main controls of low and high quantile flows) for the calibration period are shown together in Figure 10c.

It indicates that direct runoff is more significantly condensed into a FDC. Because of the flow regime with small low-flow variability of the Korean catchments, the FDC calibration could automatically incline the model parameter towards reproduction of low flows. Should considerable variability exist in baseflow (e.g., snow-fed catchments), performance of the FDC calibration may differ.

### 5.3 Flow signatures for improving calibration against FDCs

As evaluated, rainfall-runoff modelling against FDCs has strength in baselow or low flow prediction in South Korea while high flows were not well captured due to the absence of flow timing. It was confirmed by the flow signature reproducibility in Figure 9 and the low robustness of direct runoff simulations in Figure 10b. Hence, additional constraining may fill the gap in FDC calibration as discussed in Westerberg et al. (2014). Westerberg et al (2014) emphasised the necessity of further constraining to reduce predictive uncertainty despite their sophisticated modelling against FDCs. The comparative evaluation

of this study simply suggests that orthogonal (or complementary) flow signatures to a FDC should explain temporal flow variation (e.g., $D_{RL}$, falling limb density, and recession rate).

The box plots in Figure 11 show how the FDC calibration can be improved by additional constraints of the three flow signatures ($R_{QP}$, $I_{BF}$, and $D_{RL}$). For runoff predictions, we simply chose one parameter set with the best reproducibility of each signature from the collection of 50 parameter sets of the FDC calibration. As expected from the competitive

reproducibility of the FDC calibration in $R_{QP}$ and $I_{BF}$, no meaningful improvement was found from the addition of both signatures. On the contrary, the parameter sets constrained by $D_{RL}$ resulted in fairly improved performance, suggesting the need of metrics associated with temporal flow variation in the FDC calibration. A further study needs to be directed for regionalising flow metrics representing flow dynamics together with a framework to combine multiple signatures as it could fill the gap in model calibration against FDCs.

### 5.4 Limitations and future research directions

This study provides a lesson that modelling against regional FDCs may not be an attractive option where proximity-based parameter regionalisation performs greatly. In our knowledge, the topic of runoff prediction in ungauged catchments has





been rarely dealt in South Korea due to limited availability of quality streamflow data, thus this study may become a good reference for scientific community. Nonetheless, there are several limitations in our comparative evaluation. First, this study did not consider uncertainty in streamflow data. McMillan et al. (2012) reported typical ranges of relative errors in discharge data as around 10-20% for medium to high flow and 50-100% for low flows. The measurement errors and epistemic

uncertainty in input and output data may cause a disinformation effect on model calibration. Especially for the hydrograph calibration, if the model is significantly forced to compensate disinformation in high flows, calibrated parameters can be biased (Westerberg et al., 2011). We assumed that quality of the discharge data was adequate based on rigorous controls of the data distribution centre, but consideration of such errors will clarify their relative effects on the hydrograph- and FDC-based runoff modelling. Second, we used a conceptual runoff model with a fixed structure for all catchments, but it could be

a structural error source for some catchments. Blöschl et al. (2013) recommended that structuring a conceptual model needs to be considered in a realistic manner for reliable predictions. If this step was included in this study, predictive power might be better in catchments with relatively low NSE performance. Finally, though the proximity-based parameter regionalisation was powerful, other regionalisation methods such as the regional calibration and the spatial similarity parameter transfer would provide comprehensive information.

Obviously, one research direction stemming from this study is how to regionalise metrics related to flow timing and dynamics. The signature calibration inherently removes the concern in conventional parameter regionalisation approaches, but should be based on well-regionalised signatures. Candidate flow signatures that can enhance the FDC calibration would be the overall flow variability, the flow autocorrelation, the rising and falling limb densities, and the slope of fast recession curve among other metrics. Unfortunately, the task of regionalising these signatures will be challenging. Westerberg et al.

(2016) found that the metrics gauging flow dynamics could be more uncertain than one measuring flow distribution (e.g., quantile flows). A new framework beyond conventional regionalisation methods may be needed to reduce uncertainty in regional flow signatures.

## 6 Summary and conclusions

In this study, we investigated performance of the FDC calibration by comparing it with hydrograph-based methods for

gauged and ungauged catchments. We began with parameter calibration of the GR4J model against the observed hydrographs and empirical FDCs at 45 catchments in South Korea using random simulations. Predictive performance and uncertainty of each catchment were evaluated using parameter sets obtained. For evaluation for ungauged catchments, hydrographs of the 45 catchments were again predicted by treating each catchment as ungauged. In doing so, we estimated regional FDCs of the catchments using a promising geostatistical method, and calibrated model parameters against the

regional FDCs. Predictive performance of the model based on regional FDCs was evaluated in comparison to hydrographs simulated with parameters transferred from neighbouring catchments. The key findings from our comparative evaluation are summarized as follows:





(1) For gauged catchments, if the FDC calibration is employed instead of the hydrograph calibration, predictive performance of the rainfall-runoff model can be significantly degraded by loss of flow timing information. Uncertainty of the hydrographs predicted by the FDC calibration would be increased by the augmented equi-finality.

(2) The geostatistical FDC regionalization showed good performance in prediction of quantile flows despite its low TND reproducibility. The top-kriging weights interpolating TNDs had high performance to predict quantile flows. Topological proximity is likely to well explain functional similarity between catchments in South Korea. However, it is notable that considering topological proximity only can bring bias where gauging density is low.

(3) The typical proximity-based parameter transfer was of strong performance to regenerate hydrographs, and outperformed model calibration with regional FDCs. Although regional FDCs would have potential for capturing functional behaviour of ungauged catchments, the absence of flow timing would lead to less robust and less predictive performance than the proximity-based parameter transfer that shows good performance under the given modelling conditions

(4) Relative merits of the model calibration with regional FDCs were strong performance in low-flow prediction. Without logarithmic transformation of the observed flows, the parameters with the regional FDCs seem to be forced to reproduce low flows because of relatively low temporal variation in baseflow of Korean catchments.

(5) Complementary flow signatures for the FDC calibration could be metrics describing flow timing and dynamics. Additional constraining with $D_{RL}$ showed fairly improved performance with the FDC calibration. A further study for regionalising those metrics will improve the model calibration against regional FDCs.

In brief, we suggest that classical parameter regionalisation is pragmatic for predicting hydrographs in ungauged catchments in South Korea where spatial proximity well captures functional similarity between catchments. Nonetheless, we believe that further studies on regionalisation of relevant flow signatures will inherently improve runoff modelling in ungauged catchments using the FDC-based calibration. The FDC calibration still has a major advantage that it can directly identify parameters against plausible flow information of the catchment of interest unlike the parameter regionalisation.

**Acknowledgements**

This study was supported by the APEC Climate Center. The authors send special thanks to Ms. Yoe-min Jeong and Dr. Hyungil Eum for their PRISM climate data sets. Data needed to reproduce modelling results are available upon request from the authors (d.kim@apcc21.org, jachun@apcc21.org).

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





**Table 1: List of the gauged catchments and hydrological features (2007-2015)**

| ID | Name | Ar[1] | Elv[2] | $P_a$[3] | $T_a$[4] | Ard[5] | $P_s$[6] | ID | Name | Ar | Elv | $P_a$ | $T_a$ | Ard | $P_s$ |
|---|---|---|---|---|---|---|---|---|---|---|---|---|---|---|---|
| 1 | Goesan Dam | 677 | 363 | 1223 | 11.0 | .69 | 29.5 | 24 | Chunyang | 145 | 201 | 1611 | 13.2 | 58 | 12.8 |
| 2 | Namgang Dam | 2293 | 431 | 1558 | 13.8 | .61 | 5.7 | 25 | Osu | 360 | 255 | 1434 | 11.7 | 61 | 49.6 |
| 3 | Miryang Dam | 104 | 512 | 1824 | 13.3 | .50 | 20.1 | 26 | Daecheon | 816 | 198 | 1336 | 13.2 | 70 | 23.4 |
| 4 | Boryeong Dam | 162 | 244 | 1997 | 11.4 | .44 | 140.8 | 27 | Jeonju | 276 | 176 | 1312 | 12.9 | 71 | 29.5 |
| 5 | Buan Dam | 57 | 177 | 1253 | 13.7 | .76 | 39.3 | 28 | Hari | 528 | 197 | 1332 | 13.4 | 71 | 20.8 |
| 6 | Seomjingang Dam | 763 | 357 | 1487 | 11.4 | .58 | 54.7 | 29 | Bongdong | 345 | 245 | 1354 | 13.2 | 69 | 19..4 |
| 7 | Soyanggang Dam | 2783 | 634 | 1231 | 9.5 | .64 | 50.6 | 30 | Hannaedari | 284 | 126 | 1218 | 12.6 | 75 | 31.2 |
| 8 | Andong Dam | 1629 | 543 | 1330 | 10.0 | .61 | 51.5 | 31 | Suchon | 224 | 94 | 1254 | 12.4 | 72 | 42.4 |
| 9 | Yongdam Dam | 930 | 510 | 1508 | 12.6 | .60 | 22.6 | 32 | Wolpo | 1158 | 315 | 1303 | 11.3 | 66 | 30.1 |
| 10 | Imha Dam | 1976 | 388 | 1319 | 10.1 | .63 | 50.6 | 33 | Jeomchon | 615 | 371 | 1230 | 11.5 | 71 | 29.9 |
| 11 | Hoengseong Dam | 208 | 436 | 1247 | 11.1 | .68 | 28.5 | 34 | Sancheong | 1131 | 554 | 1608 | 13.8 | 59 | 14.1 |
| 12 | Habcheon Dam | 929 | 495 | 1470 | 12.9 | .62 | 17.1 | 35 | Seonsan | 988 | 298 | 1202 | 12.0 | 73 | 27.7 |
| 13 | Chungju Dam | 6705 | 608 | 1289 | 9.9 | .62 | 51.5 | 36 | Nonsan | 477 | 151 | 1309 | 13.0 | 71 | 19.4 |
| 14 | Juam Dam | 1029 | 269 | 1765 | 12.7 | .52 | 19.5 | 37 | Ugon | 134 | 39 | 1272 | 13.2 | 73 | 19.3 |
| 15 | Jangheung Dam | 192 | 198 | 1733 | 13.4 | .54 | 17.6 | 38 | Seokdong | 156 | 71 | 1268 | 12.8 | 72 | 29.5 |
| 16 | Jungranggyo | 209 | 131 | 1388 | 12.7 | .66 | 22.9 | 39 | Cheongju | 165 | 149 | 1235 | 12.3 | 73 | 24.8 |
| 17 | Munmak | 1138 | 303 | 1286 | 11.9 | .69 | 25.1 | 40 | Heodeok | 609 | 193 | 1266 | 12.4 | 71 | 23.0 |
| 18 | Yeongchun | 4775 | 996 | 1145 | 7.9 | .62 | 83.3 | 41 | Yuseong | 246 | 193 | 1253 | 12.6 | 73 | 23.0 |
| 19 | Yeongwol-1 | 1614 | 625 | 1263 | 9.7 | .62 | 51.3 | 42 | Boksu | 162 | 216 | 1267 | 12.2 | 71 | 23.6 |
| 20 | Pyeongchang | 696 | 720 | 1235 | 9.3 | .62 | 62.3 | 43 | Sangyeogyo | 495 | 255 | 1267 | 12.2 | 71 | 23.6 |
| 21 | Naerincheon | 1013 | 752 | 1231 | 9.5 | .64 | 50.6 | 44 | Gidaegyo | 361 | 250 | 1218 | 11.3 | 70 | 30.6 |
| 22 | Wontong | 300 | 707 | 1283 | 8.6 | .59 | 71.0 | 45 | Indong | 68 | 203 | 1229 | 12.0 | 72 | 24.8 |
| 23 | Hampyeong | 105 | 87 | 1327 | 13.7 | .72 | 23.7 | | | | | | | | |

[1]Draiage Area (km$^2$), [2]Mean elevation (m), [3]Mean annual precipitation (mm), [4]Mean annual temperature (℃), [5]Aridity (unitless) defined by the sum of potential evapotranspiration divided by the sum of precipitation, and [6]Mean annual snowfall (mm) defined by mean annual precipitation when mean temperatures were below 0℃. All climatological features were calculated by spatial averages of the grid data.



**Table 2: Ranges of GR4J parameters used for parameter calibration (Demirel et al., 2013)**

| Parameter | Range |
|-----------|-----------|
| X1 (mm) | 10 to 2000 |
| X2 (mm) | -8 to +6 |
| X3 (mm) | 10 to 500 |
| X4 (days) | 0.5 to 4.0 |



**Figure 1: Locations of the gauged catchments for GR4J model and FDC regionalization. Catchment numbers are labelled at the centroid of each catchment.**




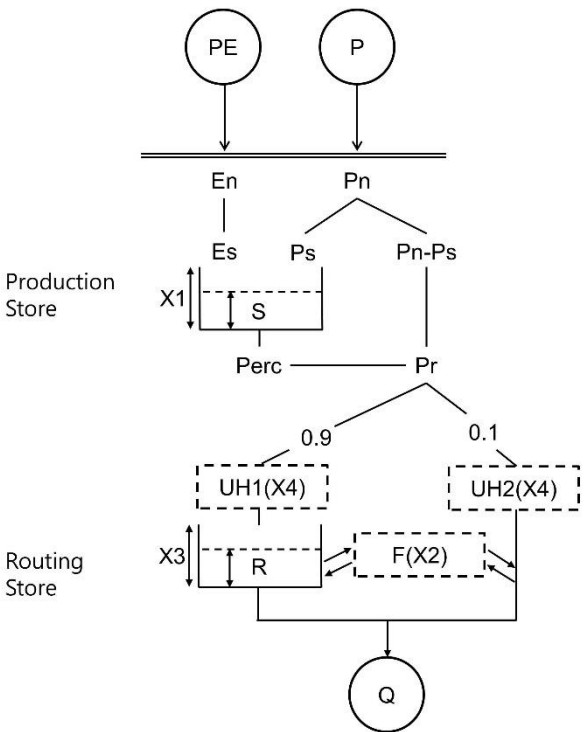

**Figure 2: The schematised structure of GR4J (X1-X4: model parameters, PE: potential evapotranspiration, P: precipitation, Q: runoff, other letters indicate variables conceptualizing internal catchment processes).**





**Figure 3: Performance comparison between the hydrograph calibration (a and c) and the FDC calibration (b and d) in terms of high flow (NSE) and low flow reproducibility (LNSE). Straight lines connect two measures for the calibration and validation periods of each catchment.**





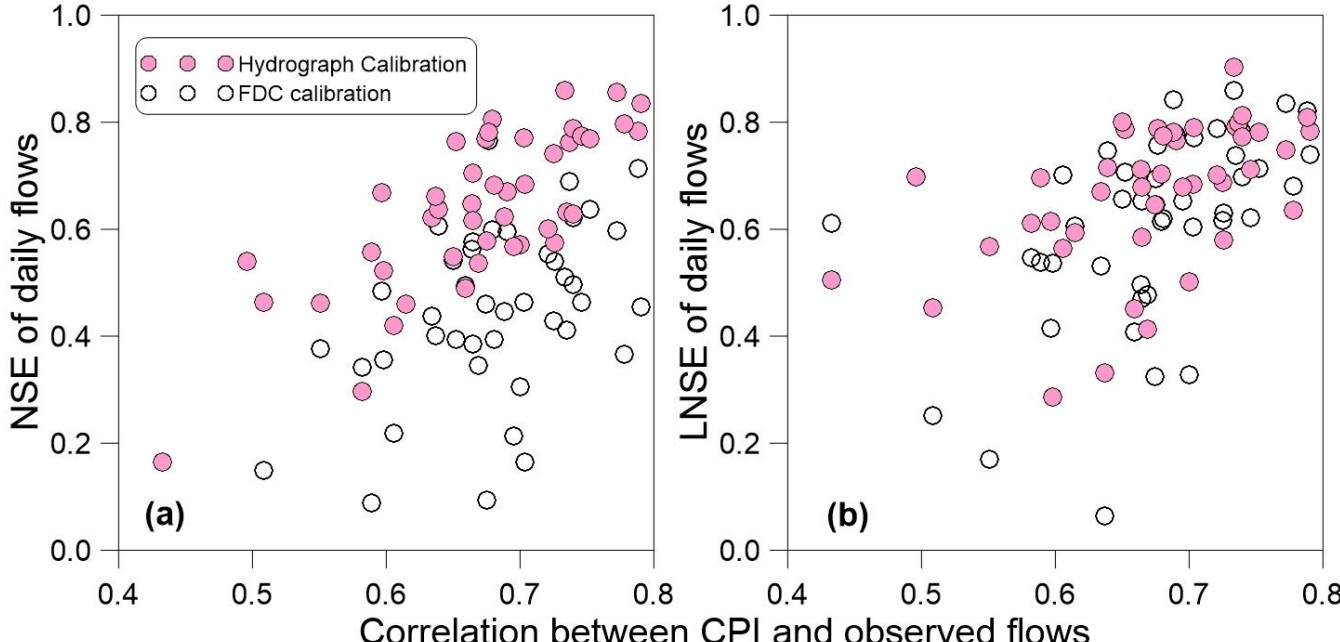

**Figure 4: The relationships between model input-output consistency and (a) high flow reproducibility (NSEs) and (b) low flow reproducibility (LNSEs)**







**Figure 5: Observed and predicted hydrographs (continuous and dashed lines) with estimated uncertainties (shaded area) at three stations with best (top), intermediate (middle), and worst (bottom) predictive performance respectively. The plot inside of each hydrograph present agreement between observed and modelled FDCs in log-log space in which its horizontal and vertical axes are for exceedance probability (range of 0-1) and runoff (same range of each hydrograph) respectively.**

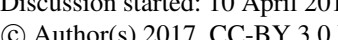



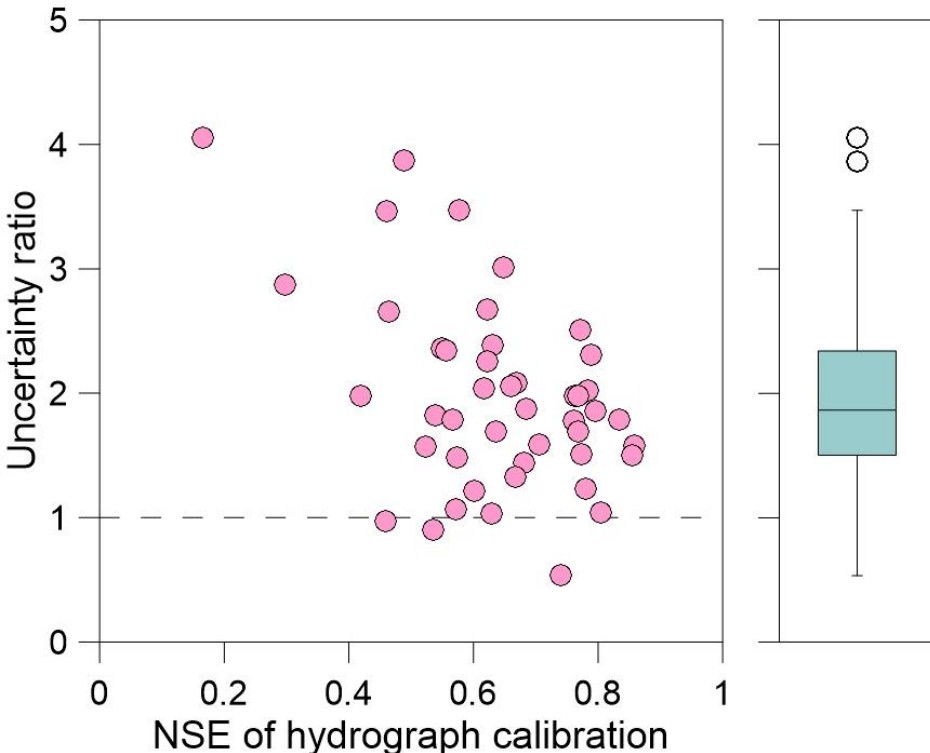

**Figure 6: 1:1 scatter plot between the uncertainty ratio and NSE of the hydrograph calibration, and the box plot of the uncertainty ratios**





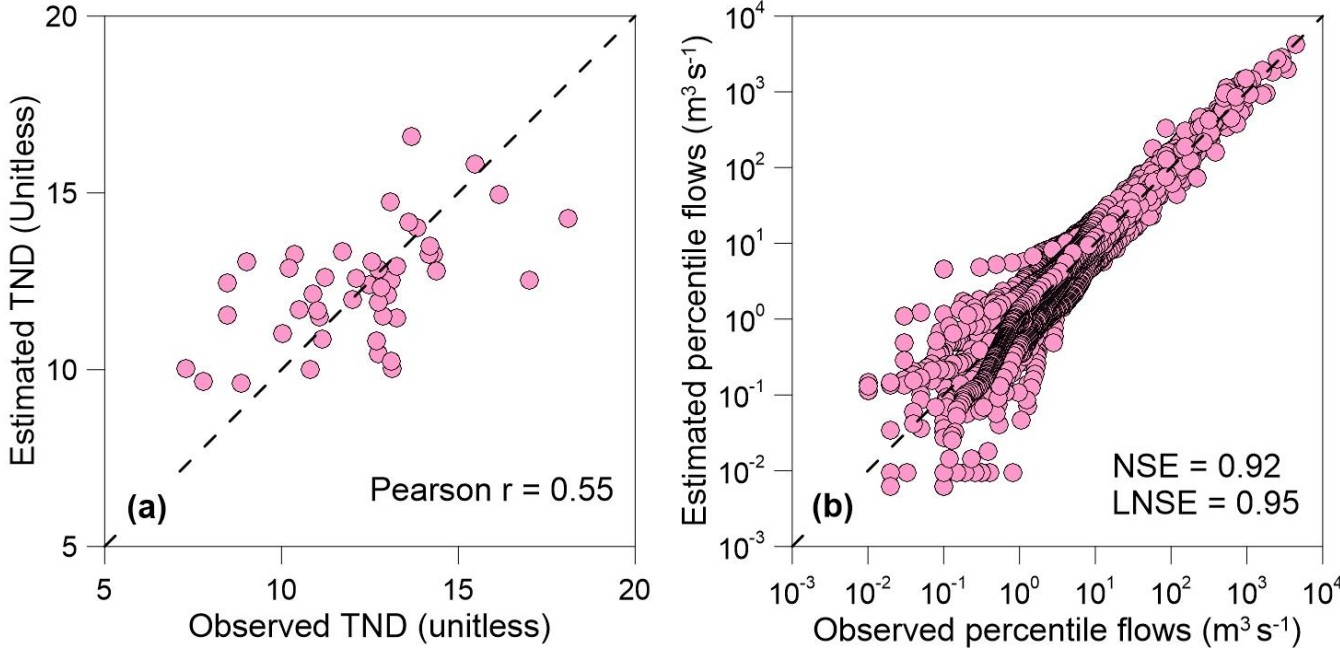

**Figure 7: (a) 1:1 scatter plots between the observed and estimated TNDs, and (b) the observed and estimated quantile flows of 45 catchments.**





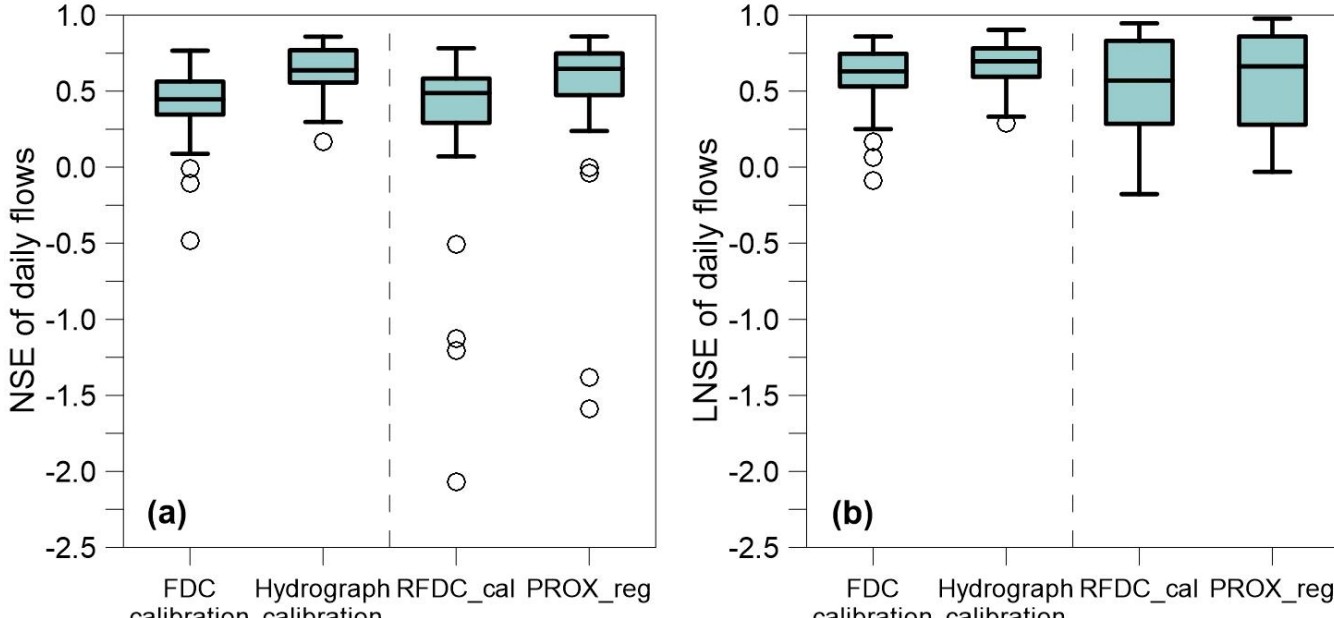

**Figure 8: (a) boxplots of NSEs (high flow reproducibility) of methods for gauged catchments (FDC and Hydrograph calibrations) and for ungauged catchments (RFDC_cal and PROX_reg), (b) boxplots of LSNEs (low flow reproducibility) gained from the same methods. The dashed lines distinguish between method for gauged and ungauged catchments.**





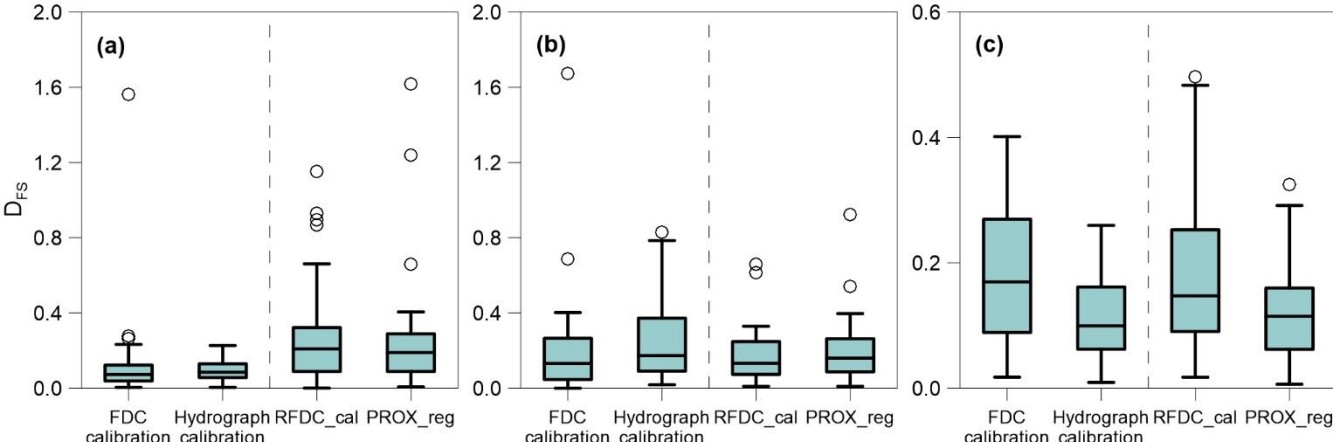

**Figure 9: Flow signature reproducibility of methods for gauged catchments (FDC and Hydrograph calibrations) and for ungauged catchments (RFDC_cal and PROX_reg) in terms of (a) $R_{QP}$, (b) $I_{BF}$, and (c) $D_{RL}$. The dashed lines distinguish between method for gauged and ungauged catchments.**





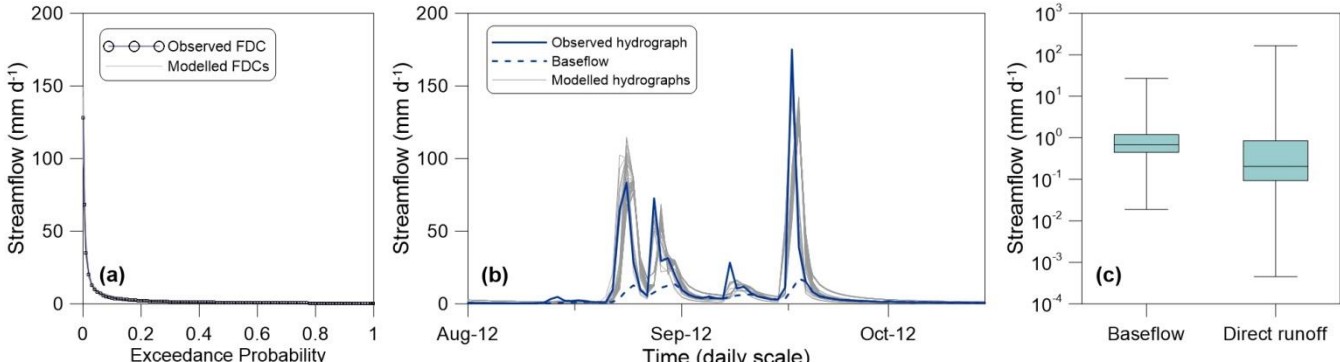

**Figure 10: (a) observed FDC and FDCs modelled by the 50 parameter sets from the FDC calibration, (b) sample observed hydrograph, and hydrograph modelled by the same 50 parameter sets, and (c) Box plots of observed baseflow and direct runoff. The whiskers indicate maximum and minimum values. All panels are for Namgang dam (catchment 2) with 0.86 and 0.51 NSEs of daily flows using the hydrograph calibration and the FDC calibration respectively.**




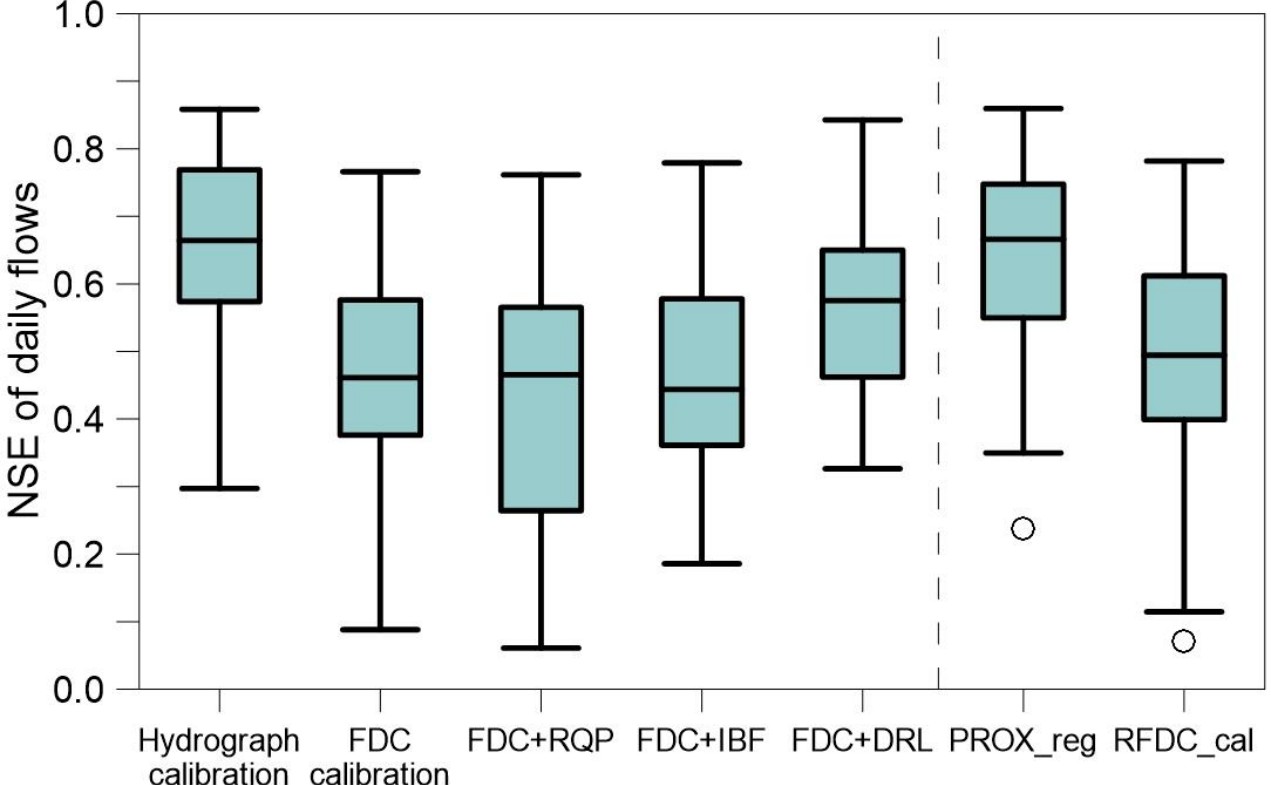

**Figure 11: Predictive performance of the FDC calibration with additional constraining using $R_{QP}$ (FDC+RQP), $I_{BF}$ (FDC+IBF), and $D_{RL}$ (FDC+DRL). The dashed line distinguishes between methods for gauged and ungauged catchments. The dashed line distinguishes between method for gauged and ungauged catchments. 39 catchments having positive NSEs for all methods were plotted.**