# Peer review of "Comparative evaluation of rainfall-runoff modelling against flow duration curves in semi-humid catchments"

_Hydrology and Earth System Sciences, 2017_

## Referee Comment (RC1) · Anonymous Referee #1 · 18 May 2017

Review summary:

This study evaluates the predictive performance of a rainfall-runoff model when it is calibrated against flow duration curve (FDC), and compares the results with those obtained with conventional hydrograph-based approaches. Authors focus on 45 gauged catchments in South Korea and derive FDCs and streamflow indices using regionalization. Their results show that even though FDC calibration yields promising performance in predicting low flows, it could generally lead to noticeably weaker performance and higher uncertainty in streamflow predictions (in comparison to hydrograph-focused calibration), potentially due to the absence of flow timing. In ungauged catchments, their results demonstrate that the proximity-based parameter regionalization (i.e., not using

FDC) performs better than the calibration against regional FDCs estimated by a geo-statistical method. I have found this study valid from the scientific and presentation quality, however, I have a number of major issues with its scientific contributions, which I am elaborating on in this review. Overall, I recommend re-submission after major revisions.

Major comments:

The first objective in this study, as stated on page 4 lines 8-10, is to evaluate predictive performance of the hydrograph calibration and the FDC calibration as well as their uncertainty for gauged catchments. I think this idea has been addressed extensively in the literature (some of which are cited in the present manuscript), and therefore, it does not need any further examination. The fact that this study finds FDC-based calibration less promising than hydrograph-based approach (as stated on page 11 lines 13-15) is not of a big surprise, e.g., due to different challenges in FDC estimation and that timing is not handled by FDC, as authors point out in the manuscript as well. Probably, what is more worth studying is how FDC can help to reduce equi-finality. As a result, I suggest that authors remove the first part of the study, or consider FDC as an additional criteria in model calibration and show how its use would improve parameter identifiability (e.g., posterior ranges) and reduce uncertainty (e.g., uncertainty ratio of hydrograph+FDC to only hydrograph).

Authors claim that FDC calibration performs promising for low flow prediction. I would argue that FDC-based approach performs only better than hydrograph-based approach, not good overall. Looking at figure 9, I see that there are several large deviations between simulated and observed BFI (up to 90%) which means that FDC-based method is not that reliable. The reason why it performs better than hydrograph-based approach is that the latter only focuses on high-flows as the Nash metric is biased on large values. So, this claim is of a sort of concern to me.

My other major issue is with how authors set the experiments related to streamflow

predictions in ungauged catchments. They first mention three classes of parameter regionalisation in lines 26-30 on page 8, but then mention that they chose the proximity-based approach due to its simplicity. I think, given than the first part of the paper can be removed according to my view, authors should focus more on this part and compare different regionalization approaches. Also, why not considering the proximity-based transfer of FDCs from donor catchments as am additional approach? Then, a potential topic for the paper can be "comparative evaluation of different regionalization approaches for model calibration in ungauged catchment".

Page 7 line 15 says that "Synthetic runoff time series were generated by GR4J for the same 45 catchments by treating each catchment as ungauged.

Introduction needs to be shorter. Objectives are stated after 6 very long paragraphs in the introduction section. Moreover, discussions sub-sections are too long. I think authors can make them briefer, but still transfer the message to readers.

Minor comments (for improving manuscript quality):

I suggest continuous line numbering in the next version of the manuscript.

Page 3, line 34: I suggest that a little explanation is provided here about the proximity-based approach. It is not clear up to this point what that approach actually is. Authors provide a brief description on page 7 line 17. Also, I suggest removing "in truth"

Also related to the description of proximity-based approach, section 3.3.2 is not fully understandable. I suggest rewording the paragraph so that the approach is explained in a clearer way. Moreover, please explain at the beginning of this section that when you talk about parameters in the proximity-based approach, you actually mean the parameters of the hydrologic model. Because one can also estimate the parameters of a parametric FDC using this approach.

Page 9 line 1: what do you mean by "synchronizing" donor catchments?

Page 4 line 3: define "orthogonal"

Please explain why Monte Carlo is used for parameter estimation, whereas SCE has been used by authors in one of the catchments. I believe that there is the possibility of quantifying uncertainty bounds using the solutions sampled by SCE.

Page 12 line 26-28: the sentence is not understandable. Please reword.

———————————————————

---

## Referee Comment (RC2) · Anonymous Referee #2 · 13 Jun 2017

The work explores the predictive performance of application of a FDC in comparison with conventional hydrograph calibration and parameter regionalisation for gauged and ungauged catchments. While the manuscript has some interesting results and discussion, it is not clear to me from the text how the work is innovative and unique to the previous studies mentioned in the literature review and discussion. For this reason I suggest major review to lift the manuscript before the work is suitable for publication in HESS.

To me the manuscript currently lacks focus in the sense that the key research gaps and innovation should stand out more clearly in the introduction and conclusion. In my

opinion the authors should focus on quality and innovation rather than applying existing techniques, and quantity of results and discussion.

Major comments:

The innovation of this work compared to previous studies is not clear to me. Could the authors please state explicitly the innovation of their work compared to previous FDC regionalisation studies and existing methods? The specific research gap/s that the work is addressing should be more prominent in the introduction, and the innovations compared to previous studies need to be more prominent in the summary and conclusions section.

Could the authors also please describe in detail how you improve on your previous 2016 submission to HESS that uses the same 45 South Korean catchments and has a similar goal: "Kim et al. A comparison between parameter regionalization and model calibration with flow duration curves for prediction in ungauged catchments". Reading the comments from the reviewers on the previous submission there are some points that have not been fully addressed in this submission.

I suggest adding either "ungauged" or "regionalisation " to the title of the manuscript to make the title more descriptive of the work undertaken in the manuscript.

Minor comments:

In the future please line number the manuscript continuously e.g. 1-999 rather than by each page, this will aid the review process.

The first paragraph of Section 3 introduces the GR4J model, and I see no logical progression to Section 3.1. I recommend an opening paragraph describing the structure of the methodology and turning your current paragraph into a new Section e.g. "3.1 Hydrological model (GR4J)". Furthermore I suggest a second section e.g. "3.2. Flow duration curve (FDC)" for consistency and to ensure reproducibility of your work.

Can you clarify in page 9, lines 4-7 your justification for applying a different objective

function for calibration (Eq. 2a, 2b, 2c) OBJ, to the functions used to evaluate predictive performance (Eq. 5) NSE and LNSE?

Page 10, Line 12 I disagree that the term NSE was used "directly" for calibration, rather I understand that you used a combination of the NSE and the WBE in OBJ. Please clarify.

Figure 3: I suggest adding headings "GR4J", and "FDC" to the top panels to ease interpretation.

Figure 4: If these are 1:1 plots then I suggest adding a 1:1 line to the panels to ease interpretation.

Figure 5. Where is the difference between the first and second column of panels described in the caption or figure? I suggest adding headings to describe the difference in a similar manner to my recommendation for Figure 3.

Could you please provide a more professional title (i.e. remove the phrase "performs good") to Subsection 5.2? e.g. "performs well", or a new title "Suitability of the FDC calibration for prediction of low flows"

In Figure 10a it is very difficult to see the difference between observed and modelled FDCs. If this result is presented then could the authors provide an inset zoom to allow the reader to see the difference between the FDCs for the highest flows?

Please proof read future submissions in greater detail, see some notes below.

Typos and clarifications:

Abstract line 11: "...Monte-Carlo framework..." is a bit vague given the complexity of your calibration (e.g. initial use of the SCE) please be more descriptive.

Page 1, Line 2: Should we not have an "and"?

Page 2, Line 9: Should "has" be replaced with "is"?

Page 2, Line 15: In the papers that you refer to in the previous sentence (i.e. Beven 2006), the term used is "equifinality" rather than "equi-finality". As this is a widely used term in the field of hydrological modelling I think that this consistency is important. Furthermore, the paper referenced (Oudin, 2008) does not refer to the term "equifinality", and so I feel that you may wish to choose a reference that better reflects the implication of the sentence.

Page 4, Line 3: Please clarify what you mean by "orthogonal" here

Page 4, Line 13: Why have you used the term "simply"? I suggest removing it.

Page 4, Line 18: "Characterized", previously you have used UK English rather than US English, e.g.

Page 4, Line 7 "regionalisation". Another e.g. Figure 1 caption "regionalization". Another Page 8, Line 25: "regionalization". Another example when you refer to Figure 2 you use "schematized", but in the

Figure 2 caption you use "schematised". Please be consistent throughout the paper.

Page 4, Line 32: typo "Mistry", should be "Ministry"

Page 7, Line 25: Please choose an alternative wording to: "and thus of consistency", e.g. "and therefore are consistent"

Page 8, line 10: "50 parameter sets" I recommend adding "...from the Monte-Carlo..." to remind the reader what you are referring to here.

Page 10, Paragraph starting with line 22. Please clarify what correlation coefficient you are referring to. I.e. Pearson correlation.

Page 16, line 15. I am not sure if the word "Obviously" is necessary here. How is this future work more "obvious" than the other limitations that you have discussed above? I suggest removing it.

[Figure]

Table 1: Typo: "Draiage"

[Figure]

---

## Author Comment (AC1) · 11 Jul 2017

We greatly appreciate your valuable efforts to review our manuscript. Following are specific responses as per comment.

Comments from Anonymous Referee #1: This study evaluates the predictive performance of a rainfall-runoff model when it is calibrated against flow duration curve (FDC), and compares the results with those obtained with conventional hydrograph-based approaches. Authors focus on 45 gauged catchments in South Korea and derive FDCs and streamflow indices using regionalization. Their results show that even though FDC calibration yields promising performance in predicting low flows, it could generally lead

to noticeably weaker performance and higher uncertainty in streamflow predictions (in comparison to hydrograph-focused calibration), potentially due to the absence of flow timing. In ungauged catchments, their results demonstrate that the proximity-based parameter regionalization (i.e., not using FDC) performs better than the calibration against regional FDCs estimated by a geostatistical method. I have found this study valid from the scientific and presentation quality, however, I have a number of major issues with its scientific contributions, which I am elaborating on in this review. Overall, I recommend re-submission after major revisions. Major comments: The first objective in this study, as stated on page 4 lines 8-10, is to evaluate predictive performance of the hydrograph calibration and the FDC calibration as well as their uncertainty for gauged catchments. I think this idea has been addressed extensively in the literature (some of which are cited in the present manuscript), and therefore, it does not need any further examination. The fact that this study finds FDC-based calibration less promising than hydrograph-based approach (as stated on page 11 lines 13-15) is not of a big surprise, e.g., due to different challenges in FDC estimation and that timing is not handled by FDC, as authors point out in the manuscript as well. Probably, what is more worth studying is how FDC can help to reduce equi-finality. As a result, I suggest that authors remove the first part of the study, or consider FDC as an additional criteria in model calibration and show how its use would improve parameter identifiability (e.g., posterior ranges) and reduce uncertainty (e.g., uncertainty ratio of hydrograph+FDC to only hydrograph).

–> We agree that low performance of the FDC calibration is not a surprise for gauged catchments with continuous hydrographs. However, we think it is necessary to show uncertainty from equifinality in the FDC calibration is double of that in the hydrograph calibration. It may provide information that doubled equifinality can produce much higher errors than transferring parameter sets.

Authors claim that FDC calibration performs promising for low flow prediction. I would argue that FDC-based approach performs only better than hydrograph-based approach, not good overall. Looking at figure 9, I see that there are several large deviations between simulated and observed BFI (up to 90%) which means that FDC-based method is not that reliable.

–> We disagree. It was difficult for us to conclude that BFI reproducibility of the FDC calibration is worse than hydrograph calibration. In figure 9(b), the median of the FDC calibration is less than hydrograph calibration. Its 3rd quartile is much smaller than that of hydrograph calibration. RFDC_cal and PROG_reg also showed similar performance to reproduce BFI.

The reason why it performs better than hydrograph-based approach is that the latter only focuses on high-flows as the Nash metric is biased on large values. So, this claim is of a sort of concern to me.

–> The FDC calibration used the same objective function in Eq. 2(a). If NSE exaggerated high-flow reproducibility in the hydrograph calibration, the FDC calibration should be in the case (i.e., high flow quantiles should be emphasized too). Nonetheless, the FDC calibration showed reproducibility comparable to the hydrograph calibration in low flows.

My other major issue is with how authors set the experiments related to streamflow predictions in ungauged catchments. They first mention three classes of parameter regionalisation in lines 26-30 on page 8, but then mention that they chose the proximity based approach due to its simplicity. I think, given than the first part of the paper can be removed according to my view, authors should focus more on this part and compare different regionalization approaches.

–> Although comparing between regionalization methods is a meaningful topic, it has been studied widely. For example, Oudin et al. (2008) and Parajka et al. (2013) provided a lesson that the high performance of proximity-based calibration. The proximity-based regionalization was attractive under modeling conditions in Korean catchments based on their comprehensive evaluations. On the other hand, it may be difficult to

find a comparison between a FDC calibration and a regionalization. This study shows that a simple parameter transfer from gauged catchments outperform a local calibration against well-predicted FDC. This lesson can be practically meaningful, because regionalization of flow signatures (e.g., FDCs) requires additional efforts.

Also, why not considering the proximity-based transfer of FDCs from donor catchments as an additional approach? Then, a potential topic for the paper can be "comparative evaluation of different regionalization approaches for model calibration in ungauged catchment".

–> This is an outside topic of this study. We can consider it for future studies. For example, we can answer a research question "Can simply transferred FDCs be comparable to empirical or regional FDCs in rainfall-runoff modeling?" It is a good suggestion, but beyond our topic to compare between a local calibration against a regional FDC and a parameter regionalization.

Page 7 line 15 says that "Synthetic runoff time series were generated by GR4J for the same 45 catchments by treating each catchment as ungauged.

–> Nothing is requested. This sentence is to explain how to evaluate runoff simulation for ungauged catchment. If necessary, we will review the sentence again.

Introduction needs to be shorter. Objectives are stated after 6 very long paragraphs in the introduction section. Moreover, discussions sub-sections are too long. I think authors can make them briefer, but still transfer the message to readers.

–> We can consider this comment in revision to make the manuscript concise to have better readability.

Minor comments (for improving manuscript quality):

I suggest continuous line numbering in the next version of the manuscript.

–> For convenience, we will add the line numbers continuously.

Page 3, line 34: I suggest that a little explanation is provided here about the proximity-based approach. It is not clear up to this point what that approach actually is. Authors provide a brief description on page 7 line 17. Also, I suggest removing "in truth"

–> We will shortly add the description in the sentence. We will remove "in truth".

Also related to the description of proximity-based approach, section 3.3.2 is not fully understandable. I suggest rewording the paragraph so that the approach is explained in a clearer way. Moreover, please explain at the beginning of this section that when you talk about parameters in the proximity-based approach, you actually mean the parameters of the hydrologic model. Because one can also estimate the parameters of a parametric FDC using this approach.

–> We will review this section again and will concisely restate the methodology with more readability.

Page 9 line 1: what do you mean by "synchronizing" donor catchments?

–> It simply means that we used same donor catchments for the regional FDC and the parameter regionalization. We will reword it.

Page 4 line 3: define "orthogonal"

–> We adopted the term of "orthogonal" from Hrachowitz et al. (2013). "orthogonal" means something that can complement FDCs. We will clearly define it, or use a more appropriate expression.

Please explain why Monte Carlo is used for parameter estimation, whereas SCE has been used by authors in one of the catchments. I believe that there is the possibility of quantifying uncertainty bounds using the solutions sampled by SCE.

–> Using SCE, it was difficult to find convergence when calibrating against FDCs because of high equifinality. Thus, we used a similar method in Weterberg et al. (2011, 2014) that proposed a FDC-based calibration. The Monte-Carlo framework was better

for us to flexibly use for all calibrations in this study.

Page 12 line 26-28: the sentence is not understandable. Please reword.

–> We will rewrite it. We just mentioned that errors in regional FDCs are not a great concern based on high performance of the geostatistical method.

References

Hrachowitz, M. et al.: A decade of Predictions in Ungauged Basins (PUB) - A review. Hydrolog. Sci. J., 58, 1198–1255. Doi:10.1080/02626667.2013.803183, 2013.

Oudin, L., Andréassian, V., Perrin, C., Michel, C., and Le Moine, N.: Spatial proximity, physical similarity, regression and ungaged catchments: a comparison between of regionalization approaches based on 913 French catchments, Water Resour. Res., 44, W03413, doi:10.1029/2007WR006240, 2008.

Parajka, J., Viglione, A., Rogger, M., Salinas, J. L., Sivapalan, M., and Blöschl, G.: Comparative assessment of predictions in ungauged basins – Part 1: Runoff-hydrograph studies, Hydrol. Earth Syst. Sci., 17, 1783-1795, doi:10.5194/hess-17-1783-2013, 2013.

Westerberg, I. K., Gong, L., Beven, K. J., Seibert, J., Semedo, A., Xu, C.-Y., and Halldin, S.: Regional water balance modelling using flow-duration curves with observational uncertainties, Hydrol. Earth Syst. Sci., 18, 2993-3013, doi:10.5194/hess-18-2993-2014, 2014.

Westerberg, I. K., Guerrero, J.-L., Younger, P. M., Beven, K. J., Seibert, J., Halladin, S., Freer, J. E., and Xu, C.-Y.: Calibration of hydrological models using flow-duration curves, Hydrol. Earth Syst. Sci., 15, 2205-2227, doi:10.5194/hess-15-2205-2011, 2011.

---

## Author Comment (AC2) · 11 Jul 2017

We greatly appreciate your valuable efforts to review our manuscript. Following are specific responses as per your constructive comment. Once again, we are thankful for your valuable time.

Anonymous Referee #2:

The work explores the predictive performance of application of a FDC in comparison with conventional hydrograph calibration and parameter regionalisation for gauged and ungauged catchments. While the manuscript has some interesting results and discussion, it is not clear to me from the text how the work is innovative and unique to the previous studies mentioned in the literature review and discussion. For this reason I suggest major review to lift the manuscript before the work is suitable for publication in HESS. To me the manuscript currently lacks focus in the sense that the key research gaps and innovation should stand out more clearly in the introduction and conclusion. In my opinion the authors should focus on quality and innovation rather than applying existing techniques, and quantity of results and discussion.

–> Although we applied existing techniques to predict runoff and FDCs, we believe the comparative evaluation in this study is meaningful to select an appropriate approach for ungauged catchments. There are a plethora of approaches to predict streamflow in ungauged catchments (e.g., rainfall-runoff modeling with parameter regionalization, model calibration against flow signatures, and direct FDC regionalization among many others). Which one is better is a practical and important question for modelers. It is a heavy burden for a modeler to apply all existing methods for ungauged catchments. This study provides a lesson that model calibration against a well-predicted FDC may not be comparable to a simple parameter transfer from neighboring gauged catchments. Thus, we can highlight that the loss of flow timing can significantly influence efficiency of rainfall-runoff modeling even in the case of ungauged catchments. It is no surprise in the fact that the FDC calibration with no flow timing information cannot provide a better model performance for gauged catchments. However, for ungauged catchments, we cannot assure if a parameter regionalization outperforms the calibration against a predicted FDC. This study implies that calibration against a well-predicted FDC may be less attractive than a simple parameter transfer. Despite the absence of flow timing information, FDC is regarded as comprehensive flow information reflecting catchment behaviors. Thus, one can hypothesize that behavioral parameter sets directly obtained from an empirical (or predicted) FDC may be better than a priori parameter sets. This study shows that a priori parameter sets gained from surrounding gauged catchments may be better. This is novelty of our comparative study. We believe a comparative study can provide a practical lesson as did Zhang et al. (2015),

although existing techniques are applied only.

Major comments: The innovation of this work compared to previous studies is not clear to me. Could the authors please state explicitly the innovation of their work compared to previous FDC regionalisation studies and existing methods? The specific research gap/s that the work is addressing should be more prominent in the introduction, and the innovations compared to previous studies need to be more prominent in the summary and conclusions section.

–> It is possible to improve the introduction to more clearly highlight the research question. Several FDC regionalization studies emphasized that a FDC can comprehensively reflect catchment behaviors. Thus, one may assume that direct calibration against a regional FDC is a solution to overcoming drawbacks in parameter regionalization. Our study argues that it is not true in South Korea. We can improve the introduction with the given references.

Could the authors also please describe in detail how you improve on your previous 2016 submission to HESS that uses the same 45 South Korean catchments and has a similar goal: "Kim et al. A comparison between parameter regionalization and model calibration with flow duration curves for prediction in ungauged catchments". Reading the comments from the reviewers on the previous submission there are some points that have not been fully addressed in this submission.

–> Here, we briefly summarize how we considered the comment given by the previous review process. We believe the comprehensive comments were considered in the revision generally. For example, actual constraining with flow signatures, and replacing the objective function, evaluating low and high flows are main revisions that considered the comments. If necessary, we will recheck the comments again.

–> The referee 1 mainly argued that our study had limited contribution to prediction in ungauged basins because of existing FDC methods for runoff prediction. However, the objective of our study was not to provide a new FDC-based runoff prediction, but a

comparative evaluation between existing methods. Hence, we disagreed. The referee 1 also argued that it is no surprise with low performance of the FDC calibration. However, we cannot assure it in case of ungauged catchments. We disagreed. The small number of gauged catchments was pointed out; however, we cannot do anything to improve it because it is a given condition. 45 is not a great number, but some parameter regionalization studies used even smaller samples. The reviewer 1 argued that the objective function of NSE is not practical because of its emphasis on high flows. We replaced the objective function with one proposed by Zhang et al. (2015). And, we considered all catchments for regionalization instead of only using high performance catchments. Other minor comments were considered as well.

–> The referee 2 recommended to soften conclusions that PROXreg is better than the other. If the revised version still needs it, we will tone down again. Use of multiple criteria was recommended as well, thus we used NSE and LNSE together in revision. Some minor suggestions for title, tables, and context were given together. We added new figures and tables. The manuscript is retitled. Because the definition of "orthogonal" was missed in the revision, we will add it.

–> The referee 3 provided very constructive comments, asking first "why not parameter regionalization gained from observed FDCs?" We did not consider this comment because it is beyond the objective of this paper. We did not intend to propose a new parameter regionalization method and its performance evaluation. The primary research question of this study is that "Do parameters directly identified by predicted FDCs outperform a conventional parameter regionalization?" Regionalization of parameters gained against observed FDCs is expected to have low performance due to the loss of flow timing. Regionalizing those parameters may lead to higher uncertainty for ungauged catchments. The referee 3 also suggested including uncertainty evaluation for both approaches for ungauged catchments. Although it indirectly shows uncertainty for ungauged catchments, the uncertainty evaluation added in the revision provides a lesson that uncertainty of the FDC calibration would be two-times of that in

the hydrograph calibration for gauged catchments. Referee3 also argued that there is no evidence that the rising limb density can complement the FDC. Hence, we provided actual calibration results conditioned by the rising limb density. With some minor comments, it was asked to provide more specific examples using flow signatures in runoff modeling. So, we improved the introduction with more literatures.

I suggest adding either "ungauged" or "regionalisation " to the title of the manuscript to make the title more descriptive of the work undertaken in the manuscript.

–> We agree. We will imply it in the title.

Minor comments: In the future please line number the manuscript continuously e.g. 1-999 rather than by each page, this will aid the review process.

–> For convenience, we will add line number continuously.

The first paragraph of Section 3 introduces the GR4J model, and I see no logical progression to Section 3.1. I recommend an opening paragraph describing the structure of the methodology and turning your current paragraph into a new Section e.g. "3.1 Hydrological model (GR4J)". Furthermore I suggest a second section e.g. "3.2. Flow duration curve (FDC)" for consistency and to ensure reproducibility of your work.

–> We can consider this comment to improve readability.

Can you clarify in page 9, lines 4-7 your justification for applying a different objective function for calibration (Eq. 2a, 2b, 2c) OBJ, to the functions used to evaluate predictive performance (Eq. 5) NSE and LNSE?

–> We will clearly show the objective function and the evaluation criteria to prevent confusion between them. As known, NSE and LNSE are metrics evaluating reproducibility of high and low flows respectively.

Page 10, Line 12 I disagree that the term NSE was used "directly" for calibration, rather I understand that you used a combination of the NSE and the WBE in OBJ. Please

clarify.

–> We will remove the term "directly". Since NSE is still used in the objective function, optimization can be toward high flows. WBE is to reduce bias, not to regenerate low flows. No metrics regarding low flows are included in the objective function.

Figure 3: I suggest adding headings "GR4J", and "FDC" to the top panels to ease interpretation.

–> We will add the headings in the figures to give prompt indications. However, all simulations were from GR4J in this study. We will use other appropriate headings.

Figure 4: If these are 1:1 plots then I suggest adding a 1:1 line to the panels to ease interpretation.

–> They are not 1:1 plots. They display the relationship between input-output consistency and model performance.

Figure 5. Where is the difference between the first and second column of panels described in the caption or figure? I suggest adding headings to describe the difference in a similar manner to my recommendation for Figure 3.

–> Left and right panels are presented in same scales. We will add headings and tick labels to provide prompt indication.

Could you please provide a more professional title (i.e. remove the phrase "performs good") to Subsection 5.2? e.g. "performs well", or a new title "Suitability of the FDC calibration for prediction of low flows"

–> We will consider "Suitability of the FDC calibration for prediction of low flows" as a new title of the section. Thanks for the good suggestion.

In Figure 10a it is very difficult to see the difference between observed and modelled FDCs. If this result is presented then could the authors provide an inset zoom to allow the reader to see the difference between the FDCs for the highest flows?

–> This figure indicates higher variance loss in direct flow than in baseflow when using the FDC calibration. We will improve readability.

Please proof read future submissions in greater detail, see some notes below. Typos and clarifications: Abstract line 11: ". . .Monte-Carlo framework. . ." is a bit vague given the complexity of your calibration (e.g. initial use of the SCE) please be more descriptive.

–> We will provide a clearer explanation about the methodology in the abstract.

Page 1, Line 2: Should we not have an "and"?

–> For convenience, we will add it later.

Page 2, Line 9: Should "has" be replaced with "is"?

–> We consider either "gaining" or "has increasing".

Page 2, Line 15: In the papers that you refer to in the previous sentence (i.e. Beven 2006), the term used is "equifinality" rather than "equi-finality". As this is a widely used term in the field of hydrological modelling I think that this consistency is important. Furthermore, the paper referenced (Oudin, 2008) does not refer to the term "equifinality", and so I feel that you may wish to choose a reference that better reflects the implication of the sentence.

–> We will use "equifinality" consistently. Oudin et al. (2008) did not use the term "equifinality" literally; however, they pointed out that "most models have been shown to have no unique set of parameters to define the best model fit to the flow response of a catchment" (in paragraph 3). In the context, we could find equifinality is an important uncertainty source when extrapolating parameters to ungauged catchments. Thus, we cited it.

Page 4, Line 3: Please clarify what you mean by "orthogonal" here

–> "orthogonal" here means something that can complement FDC. We used the term in

Hrachowitz et al. (2013). We will clearly define it, or use a more appropriate expression.

Page 4, Line 13: Why have you used the term "simply"? I suggest removing it.

–> We will remove it.

Page 4, Line 18: "Characterized", previously you have used UK English rather than US English, e.g. Page 4, Line 7 "regionalisation". Another e.g. Figure 1 caption "regionalization". Another Page 8, Line 25: "regionalization". Another example when you refer to Figure 2 you use "schematized", but in the Figure 2 caption you use "schematised". Please be consistent throughout the paper.

–> We will have consistency in English. We will globally review the expressions.

Page 4, Line 32: typo "Mistry", should be "Ministry"

–> We will check typos globally.

Page 7, Line 25: Please choose an alternative wording to: "and thus of consistency", e.g. "and therefore are consistent"

–> "and therefore are consistent" is better. We will revise it.

Page 8, line 10: "50 parameter sets" I recommend adding ". . .from the Monte-Carlo. . ." to remind the reader what you are referring to here.

–> Maybe it is in page 9. We will add it as recommended.

Page 10, Paragraph starting with line 22. Please clarify what correlation coefficient you are referring to. I.e. Pearson correlation.

–> It is the Pearson correlation coefficient. We will clearly show it.

Page 16, line 15. I am not sure if the word "Obviously" is necessary here. How is this future work more "obvious" than the other limitations that you have discussed above? I suggest removing it.

–> We agree. We will remove it as suggested.

Table 1: Typo: "Draiage"

–> We will correct typos globally.

References

Hrachowitz, M. et al.: A decade of Predictions in Ungauged Basins (PUB) - A review. Hydrolog. Sci. J., 58, 1198–1255. Doi:10.1080/02626667.2013.803183, 2013.

Oudin, L., Andréassian, V., Perrin, C., Michel, C., and Le Moine, N.: Spatial proximity, physical similarity, regression and ungaged catchments: a comparison between of regionalization approaches based on 913 French catchments, Water Resour. Res., 44, W03413, doi:10.1029/2007WR006240, 2008.

Zhang, Y., Vaze, J., Chiew, F. H. S., and Li, M.: Comparing flow duration curve and rainfall-runoff modelling for predicting daily runoff in ungauged catchments, J. Hydrol., 525, 72-86, 2015.

---

## Author Response (AR1)

Dear Dr. Fabrizio Fenicia

First, let us thank for your efforts in handling our manuscript. We greatly appreciate the constructive comments from you and the anonymous referees. We believe all the comments were helpful to improve the quality of our work.

In this revision, to improve the clarity of this study, we highlighted that the scientific meaning of the model calibration against regional flow duration curves (RFDC_cal), and clearly stated the objective of this study to compare RFDC_cal with a classical parameter regionalization. We introduced strengths of RFDC_cal in line 37-72. Then, we addressed potential questions that can arise when applying RFDC_cal in practice in line 73-84. We emphasized that RFDC_cal has barely compared with conventional parameter regionalization schemes in line 85-87. If RFDC_cal has poorer predictability than the proximity-based parameter transfer (PROX_reg), RFDC_cal would not be pragmatic. The main research question of this study is whether RFDC_cal outperform PROX_reg. We addressed this question by applying two methods to 45 Korean catchments in the jackknife cross validation mode.

As shown in the previous version of our manuscript, RFDC_cal was likely to have weaker predictability than PROX_reg due to the absence of flow timing information in regional FDCs. And, we argued that flow signatures in temporal dimensions should supplement RFDC_cal. In the revision, we attempted one more parameter regionalization that transfers the parameters gained against observed FDCs to ungauged catchments. This approach cannot transfer flow timing information through the model parameters from gauged to ungauged catchments (we referred this approach to as FPROX_reg), because the behavioral parameters were gained against flow magnitudes only. We found that PROX_reg significantly outperformed FPROX_reg via a paired t-test between them. This implies that PROX_reg could transfer flow timing information to ungauged catchments, while it is impossible when using RFDC_cal. In section 4.4 (from line 348), you can find the results of several paired t-tests between modeling approaches applied in this study. We believe they provide clearer indications about performance of RFDC_cal.

In addition, as an alternative method of RFDC_cal in data-rich regions, we suggested use of regional hydrographs (e.g., Viglione et al., 2013) to preserve flow amount and timing information together. And, we emphasized that preserving all flow information inherent in hydrographs would be a key for rainfall-runoff modeling against flow metrics that condense the hydrographs. You may find this context in section 5. To make the manuscript more concise, we combined the discussion and the conclusion sections.

We believe this revision can provide clear lessons and readability. Following are our responses to specific comments from the referees. Again, we thank for all of your editing efforts.

Sincerely,

Jong Ahn Chun
Corresponding author

Response to comments from reviewer 1:

Major comments: The first objective in this study, as stated on page 4 lines 8-10, is to evaluate predictive performance of the hydrograph calibration and the FDC calibration as well as their uncertainty for gauged catchments. I think this idea has been addressed extensively in the literature (some of which are cited in the present manuscript), and therefore, it does not need any further examination. The fact that this study finds FDC-based calibration less promising than hydrograph-based approach (as stated on page 11 lines 13-15) is not of a big surprise, e.g., due to different challenges in FDC estimation and that timing is not handled by FDC, as authors point out in the manuscript as well. Probably, what is more worth studying is how FDC can help to reduce equi-finality. As a result, I suggest that authors remove the first part of the study, or consider FDC as an additional criteria in model calibration and show how its use would improve parameter identifiability (e.g., posterior ranges) and reduce uncertainty (e.g., uncertainty ratio of hydrograph+FDC to only hydrograph).

➔ *We globally revised the manuscript to provide clearer lessons from this study.*
*We agreed that it was not a surprise that the FDC calibration has more equifinality than the hydrograph calibration. Therefore, we focused on comparing RFDC_cal and PROX_reg for ungauged catchments (i.e. we removed the comparative assessment for gauged catchments in the previous version).*
*We did not consider the second option to use FDCs as an additional criterion, because it is already proposed by Pfannerstill et al. (2014).Instead, in this revision, we added one more regionalization approach that transfers parameters gained from observed FDCs to ungauged (FPROX_reg) in order to check whether PROX_reg transfers flow timing information for ungauged catchments. FPROX_reg uses parameters gained from flow magnitudes only, thus it cannot transfer flow timing information to ungauged catchments. A paired t-test showed that the performance difference between PROX_reg and FPROX_reg was significant (i.e., parameters gained from flow magnitudes only may cause predictability losses).*
*Through several paired t-tests, we found a clearer indication that PROX_reg is better than RFDC_cal for the Korean catchments. We believe that this revision can provide you clear indications.*

Authors claim that FDC calibration performs promising for low flow prediction. I would argue that FDC-based approach performs only better than hydrograph-based approach, not good overall. Looking at figure 9, I see that there are several large deviations between simulated and observed BFI (up to 90%) which means that FDC-based method is not that reliable.

➔ *In revision, we withdrew this argument. RFDC_cal could provide better predictability in low flows than high flows due to smaller variability in base flow; however, it was unlikely that RFDC_cal outperformed PROX_reg in low flows. However, it is unclear that PROX_reg outperform RFDC_cal in reproducing BFI as addressed in Q6 in Table 3.*

My other major issue is with how authors set the experiments related to streamflow predictions in ungauged catchments. They first mention three classes of parameter regionalisation in lines 26-30 on

page 8, but then mention that they chose the proximity based approach due to its simplicity. I think, given than the first part of the paper can be removed according to my view, authors should focus more on this part and compare different regionalization approaches.

➔ *In section 3.5, we addressed why the proximity-based parameter regionalization was chosen. Modeling conditions in this study were suitable to use PROX_reg. Other regionalization such as similarity-based or regression-based regionalization can be applied too, but our focus was comparing RFDC_cal with the simplest parameter regionalization.*

Also, why not considering the proximity-based transfer of FDCs from donor catchments as an additional approach? Then, a potential topic for the paper can be "comparative evaluation of different regionalization approaches for model calibration in ungauged catchment".

➔ *The geostatistical method applied in this study is a proximity-based transfer (or interpolation) of empirical FDCs. We already transferred observed FDCs to ungauged catchment using the top-kriging weights. And, it showed promising performance for predicting FDCs in ungauged catchments as addressed in section 4.1. The focus of this study is comparison between RFDC_cal and PROX_reg for rainfall-runoff modeling in ungauged catchments.*

Page 7 line 15 says that "Synthetic runoff time series were generated by GR4J for the same 45 catchments by treating each catchment as ungauged.

➔ *Nothing was requested. We globally reviewed the manuscript and used the term "LOOCV mode" to distinguish between approaches for gauged and ungauged catchments.*

Introduction needs to be shorter. Objectives are stated after 6 very long paragraphs in the introduction section. Moreover, discussions sub-sections are too long. I think authors can make them briefer, but still transfer the message to readers.

➔ *In revision, we highlighted the scientific meaning of RFDC_cal in comparison to PROX_reg. The main objective of this study is a comparative assessment of RFDC_cal.*

Minor comments (for improving manuscript quality):

I suggest continuous line numbering in the next version of the manuscript.

➔ *For convenience, we used continuous line numbers in the revised manuscript.*

Page 3, line 34: I suggest that a little explanation is provided here about the proximity-based approach. It is not clear up to this point what that approach actually is. Authors provide a brief description on page 7 line 17. Also, I suggest removing "in truth"

➔ *We globally revised the manuscript, and PROX_reg was addressed in section 3.5. We removed the term "in truth".*

Also related to the description of proximity-based approach, section 3.3.2 is not fully understandable. I suggest rewording the paragraph so that the approach is explained in a clearer way. Moreover, please explain at the beginning of this section that when you talk about parameters in the proximity-based approach, you actually mean the parameters of the hydrologic model. Because one can also estimate the parameters of a parametric FDC using this approach.

➔ *PROX_reg is now addressed in section 3.5. From line 227, we explained how we transferred behavioral parameters from gauged to ungauged catchments.*

Page 9 line 1: what do you mean by "synchronizing" donor catchments?

➔ *It means that we used same donor catchments for the regional FDC and the parameter regionalization. It was for consistency between PROX_reg and RFDC_cal as explained in line 228.*

Page 4 line 3: define "orthogonal"

➔ *In revision, we did not use the term "orthogonal".*

Please explain why Monte Carlo is used for parameter estimation, whereas SCE has been used by authors in one of the catchments. I believe that there is the possibility of quantifying uncertainty bounds using the solutions sampled by SCE.

➔ *The Monte-Carlo framework was good for us to gauge equifinality across all catchments under the same sampling size and the acceptance rate, though there are other methods for individual catchments. This approach was good to evaluate equifinality under changing input-output consistency across the 45 catchments. It is explained in line 169-175.*

Page 12 line 26-28: the sentence is not understandable. Please reword.

➔ *In revision, we did not use this sentence.*

Response to comments from reviewer 2:

The work explores the predictive performance of application of a FDC in comparison with conventional hydrograph calibration and parameter regionalisation for gauged and ungauged catchments. While the manuscript has some interesting results and discussion, it is not clear to me from the text how the work is innovative and unique to the previous studies mentioned in the literature review and discussion. For this reason I suggest major review to lift the manuscript before the work is suitable for publication in HESS. To me the manuscript currently lacks focus in the sense that the key research gaps and innovation should stand out more clearly in the introduction and conclusion. In my opinion the authors should focus on quality and innovation rather than applying existing techniques, and quantity of results and discussion.

➔ To improve the clarity of this study, we addressed strengths of RFDC_cal in comparison to the classical parameter regionalization in line 37-72. Then, in line 73-84, we addressed potential questions when applying RFDC_cal in practice. If RFDC_cal has poorer predictability than the proximity-based parameter transfer (PROX_reg), RFDC_cal would not be pragmatic because regional FDC may require expensive efforts. The main research question of this study is whether RFDC_cal outperform PROX_reg for ungauged catchments. We believe the new introduction shows objectives of this study more clearly. In addition, we added the section of paired t-tests for checking our hypotheses. It was emphasized that the flow timing information embedded in parameters gained against observed hydrographs affects predictability for ungauged catchments.

Major comments: The innovation of this work compared to previous studies is not clear to me. Could the authors please state explicitly the innovation of their work compared to previous FDC regionalisation studies and existing methods? The specific research gap/s that the work is addressing should be more prominent in the introduction, and the innovations compared to previous studies need to be more prominent in the summary and conclusions section.

➔ *As answered above, the new introduction is now focused on evaluating RFDC_cal in comparison to PROX_reg, which has been barely addressed in previous similar studies. We added paired t-tests between modelling approaches applied in this study. And, we argued that flow timing information can play an important role in prediction even in ungauged catchments. You can find this context throughout the revised manuscript. We also provide a suggestion that regional hydrographs, instead of regional FDCs, would be better to preserve flow timing information for calibration of rainfall-runoff models in ungauged catchments.*

Could the authors also please describe in detail how you improve on your previous 2016 submission to HESS that uses the same 45 South Korean catchments and has a similar goal: "Kim et al. A comparison between parameter regionalization and model calibration with flow duration curves for prediction in ungauged catchments". Reading the comments from the reviewers on the previous submission there are some points that have not been fully addressed in this submission.

➔ *Here, we briefly summarize how we considered the comments given by the previous review process. We believe the comprehensive comments were considered in the revision generally. For example, actual constraining with flow signatures, and replacing the objective function, evaluating low and high flows were considered in the manuscript.*

➔ *The referee 1 mainly argued that our study had limited contribution to prediction in ungauged basins because of existing FDC methods for runoff prediction. However, the objective of our study was not to provide a new FDC-based runoff prediction, but a comparative evaluation between existing methods. Hence, we disagreed. The referee 1 also argued that it is no surprise with low performance of the FDC calibration. However, we cannot assure it in the case of ungauged catchments, thus we disagreed. The small number of gauged catchments was pointed out; however, 45 is not a large number, but some parameter regionalization studies used even smaller samples. The reviewer 1 argued that the objective function of NSE is not practical because of its emphasis on high flows. We replaced the objective function with one proposed by Zhang et al. (2015) that considers NES and WBE together. And, we considered all catchments for regionalization instead of only using high performance catchments. Other minor comments were considered as well.*

➔ *The referee 2 recommended us to soften conclusions that PROX_reg is better than the other. Nevertheless, in the revision, it was necessary to highlight that RFDC_cal is not as good as PROX_reg, because we received clearer indications that flow timing information in gauged catchments plays an important role in prediction in ungauged catchments too. Use of multiple criteria was recommended as well, thus we used NSE and LNSE together in revision. Some minor suggestions for title, tables, and context were given together. We added new figures and tables. And, the manuscript is retitled.*

➔ *The referee 3 provided constructive comments, asking first "why not parameter regionalization gained from observed FDCs?" We did consider this comment to check whether parameters gained against hydrographs can outperform those from FDCs in ungauged catchments. As mentioned, the former significantly outperformed the latter, implying that flow timing information for ungauged catchments might be contained in the parameters from observed hydrographs. The referee3 also suggested including uncertainty evaluation for both approaches for ungauged catchments. The equifinality evaluation using the Monte-Carlo simulations provides a lesson that uncertainty of the FDC calibration would be much larger than in the hydrograph calibration, though this evaluation was not a direct uncertainty comparison between RFDC_cal and PROX_reg. Referee3 also argued that there is no evidence that the rising limb density can supplement the FDC. Hence, we provided actual calibration results conditioned by the rising limb density. This could lend support to the hypothesis. With some minor comments, it was asked to provide more specific examples using flow signatures in runoff modeling. So, we improved the introduction with more literatures about use of FDCs in model calibrations.*

I suggest adding either "ungauged" or "regionalisation " to the title of the manuscript to make the title more descriptive of the work undertaken in the manuscript.

➔ *We agree. We retitled the manuscript as "A comparative assessment of rainfall-runoff modelling against regional flow duration curves for ungauged catchments".*

Minor comments: In the future please line number the manuscript continuously e.g. 1-999 rather than by each page, this will aid the review process.

➔ *Now we used continuous line numbers.*

The first paragraph of Section 3 introduces the GR4J model, and I see no logical progression to Section 3.1. I recommend an opening paragraph describing the structure of the methodology and turning your current paragraph into a new Section e.g. "3.1 Hydrological model (GR4J)". Furthermore I suggest a second section e.g. "3.2. Flow duration curve (FDC)" for consistency and to ensure reproducibility of your work.

➔ *We considered this comment to improve readability of the methodology section.*

Can you clarify in page 9, lines 4-7 your justification for applying a different objective function for calibration (Eq. 2a, 2b, 2c) OBJ, to the functions used to evaluate predictive performance (Eq. 5) NSE and LNSE?

➔ *The objective function was to consider high-flow reproducibility and long-term water balance in model calibration. NSE and LNSE were to evaluate model predictability in high and low flows. They are addressed in line 156 and 238, respectively.*

Page 10, Line 12 I disagree that the term NSE was used "directly" for calibration, rather I understand that you used a combination of the NSE and the WBE in OBJ. Please clarify.

➔ *We provide new results and discussion sections. This sentence was removed.*

Figure 3: I suggest adding headings "GR4J", and "FDC" to the top panels to ease interpretation.

➔ *Now, Figure 5 compares between RFDC_cal and PROX_reg. GR4J and FDC do not distinguish the two approaches for ungauged catchments.*

Figure 4: If these are 1:1 plots then I suggest adding a 1:1 line to the panels to ease interpretation.

➔ *They were not 1:1 plots. They display the relationship between input-output consistency and model performance. Now it is combined in Figure 3(b) only for the hydrograph calibration.*

Figure 5. Where is the difference between the first and second column of panels described in the caption or figure? I suggest adding headings to describe the difference in a similar manner to my recommendation for Figure 3.

➔ *Instead, we provided Figure 6 to emphasize the equifinality in FDC_cal.*

Could you please provide a more professional title (i.e. remove the phrase "performs good") to Subsection 5.2? e.g. "performs well", or a new title "Suitability of the FDC calibration for prediction of low flows"

➔ *Now we mainly focused on comparing RFDC_cal and PROX_reg rather than the performance of the FDC regionalization. Accordingly, we revised all headings.*

In Figure 10a it is very difficult to see the difference between observed and modelled FDCs. If this result is presented then could the authors provide an inset zoom to allow the reader to see the difference between the FDCs for the highest flows?

➔ *We did not use this figure in revision.*

Please proof read future submissions in greater detail, see some notes below. Typos and clarifications: Abstract line 11: ". . .Monte-Carlo framework. . ." is a bit vague given the complexity of your calibration (e.g. initial use of the SCE) please be more descriptive.

➔ *We rewrote the abstract.*

Page 1, Line 2: Should we not have an "and"?

➔ *The given form is unlikely to use "and" between author names.*

Page 2, Line 9: Should "has" be replaced with "is"?

➔ *We restructured the introduction.*

Page 2, Line 15: In the papers that you refer to in the previous sentence (i.e. Beven 2006), the term used is "equifinality" rather than "equi-finality". As this is a widely used term in the field of hydrological modelling I think that this consistency is important. Furthermore, the paper referenced (Oudin, 2008) does not refer to the term "equifinality", and so I feel that you may wish to choose a reference that better reflects the implication of the sentence.

➔ *We used "equifinality" in the revision. Oudin et al. (2008) did not use the term "equifinality" literally; however, they pointed out that "most models have been shown to have no unique set of parameters to define the best model fit to the flow response of a catchment" (in paragraph 3). In the context, we could find equifinality is an important uncertainty source when extrapolating parameters to ungauged catchments. Thus, we cited it.*

Page 4, Line 3: Please clarify what you mean by "orthogonal" here

➔ *In revision, we did not use the term "orthogonal"*

Page 4, Line 13: Why have you used the term "simply"? I suggest removing it.

➔ *We removed it.*

Page 4, Line 18: "Characterized", previously you have used UK English rather than US English, e.g. Page 4, Line 7 "regionalisation". Another e.g. Figure 1 caption "regionalization". Another Page 8, Line 25: "regionalization". Another example when you refer to Figure 2 you use "schematized", but in the Figure 2 caption you use "schematised". Please be consistent throughout the paper.

➔ *We globally reviewed the expressions.*

Page 4, Line 32: typo "Mistry", should be "Ministry"

➔ *We corrected it.*

Page 7, Line 25: Please choose an alternative wording to: "and thus of consistency", e.g. "and therefore are consistent"

➔ *The context in this sentence is now moved to section 3.5 in line 228.*

Page 8, line 10: "50 parameter sets" I recommend adding ". . .from the Monte-Carlo. . ." to remind the reader what you are referring to here.

➔ *We added it in line 316 where it is necessary.*

Page 10, Paragraph starting with line 22. Please clarify what correlation coefficient you are referring to. -->I.e. Pearson correlation.

➔ *In revision, we clearly stated "Pearson" correlation coefficient where it is necessary.*

Page 16, line 15. I am not sure if the word "Obviously" is necessary here. How is this future work more "obvious" than the other limitations that you have discussed above? I suggest removing it.

➔ *In revision, it was removed.*

Table 1: Typo: "Draiage"

➔ *In revision, it was removed.*

References

Oudin, L., Andréassian, V., Perrin, C., Michel, C., and Le Moine, N.: Spatial proximity, physical similarity, regression and ungaged catchments: a comparison between of regionalization approaches based on 913 French catchments, Water Resour. Res., 44, W03413, doi:10.1029/2007WR006240, 2008.

Pfannerstill, M., Guse, B., and Fohrer N.: Smart low flow signature metrics for an improved overall performance evaluation of hydrological models, J. Hydrol., 510, 447-458, 2014.

Viglione, A., Parajka, J., Rogger, M., Salinas, J. L., Laaha, G., Sivapalan, M., and Blöschl, G.: Comparative assessment of predictions in ungauged basins – Part 3: Runoff signatures in Austria, Hydrol. Earth Syst. Sci., 17, 2263-2279, doi: 10.5194/hess-17-2263-2013, 2013.

Zhang, Y., Vaze, J., Chiew, F. H. S., and Li, M.: Comparing flow duration curve and rainfall-runoff modelling for predicting daily runoff in ungauged catchments, J. Hydrol., 525, 72-86, 2015.

**A comparative assessment of rainfall-runoff modelling against regional flow duration curves for ungauged catchments**

Daeha Kim[1], Ilwon Jung[2], Jong Ahn Chun[1]

[1] APEC Climate Center, Busan, 48058, South Korea
[2] Korea Infrastructure Safety & Technology Corporation, Jinju, Gyeongsangnam-do, 52852, South Korea

*Correspondence to*: Jong Ahn Chun (jachun @apcc21.org)

**Abstract.** Rainfall-runoff modelling has long been a special subject in hydrological sciences, but identifying behavioural parameters in ungauged catchments is still challenging. In this study, we comparatively evaluated performance of the local calibration of a rainfall-runoff model against regional flow duration curves (FDC), which is  a seemingly alternative method of classical parameter regionalisation  signature for ungauged catchments. We used a parsimonious rainfall-runoff model over 45 Korean catchments under semi-humid climate. The calibration against regional FDCs was compared with the simple proximity-based parameter regionalisation. Results show that transferring behavioural parameters from gauged to ungauged catchments significantly outperformed the local calibration against regional FDCs due to the absence of flow timing information in the regional FDCs. The behavioural parameters gained from observed hydrographs were likely to contain intangible flow timing information affecting predictability in ungauged catchments. Additional constraining with the  rising limb density appreciably improved the FDC calibrations, implying that flow signatures in temporal dimensions would supplement the FDCs. As an alternative approach in data-rich regions, we suggest calibrating a rainfall-runoff model against regionalised hydrographs to preserve flow timing information. We also suggest use of flow signatures that can supplement hydrographs for calibrating rainfall-runoff models in gauged and ungauged catchments.

**1 Introduction**

~~The runoff hydrograph, a time series of streamflow, is the basis for practical resource management tasks such as water resource allocations, designing infrastructures, flood and drought forecasting, environmental impact assessment (Westerberg et al., 2014; Parajka et al., 2013). It is essential information for investigating physical controls of catchment functional behaviours because a hydrograph aggregates processes interacting within a catchment. Prediction of the runoff hydrograph has long been an important subject in hydrological sciences and is gaining increasing attention with growing concerns about environmental changes (Blöschl et al., 2013). Runoff prediction in ungauged sites has already been a special topic in hydrological sciences, e.g., a decade-long project, Prediction in Ungauged Basins (PUB) by the International Association of Hydrological Sciences (see http://iahs.info/pub/biennia.php). However, predicting hydrograph is a still challenging task due to poor data availability and unknown knowledge of complex catchment responses (Zhang et al., 2015; Blöschl et al., 2013).~~

A standard method to predict daily streamflow is to employ a rainfall-runoff model that conceptualises catchment functional behaviours, and simulate synthetic hydrographs from atmospheric drivers (Wagener and Wheater, 2006; Blöschl et al., 2013). A prerequisite of this conceptual modelling approach is parameter identification to enable the rainfall-runoff model to imitate actual catchment behaviours. Conventionally, behavioural parameters are estimated via model calibration against observed hydrographs (referred to as the hydrograph calibration hereafter). The hydrograph calibration provides convenience to attain reproducibility of the predictand (i.e., streamflow time series), which is commonly used as a performance measure in rainfall-runoff modelling studies. Because the degree of belief in hydrological models is normally measured by how they can reproduce observations (Westerberg et al., 2011), use of the hydrograph calibration has a long tradition in runoff modelling (Hrachowiz et al., 2013).

The hydrograph calibration, however, can be challenged by epistemic errors in input and output data, sensitivity to calibration criteria, and inability under no or poor data availability (Westerberg et al, 2011; Zhang et al., 2008). Importantly, it is difficult to know whether the parameters optimised toward maximising hydrograph reproducibility are unique to represent actual catchment behaviours, since multiple parameter sets possibly show similar predictive performance (Beven, 2006, 1993). This low uniqueness of the optimal parameter set, namely the equifinality problem in conceptual hydrological modelling, can become a significant uncertainty source particularly when extrapolating the optimal parameters to ungauged catchments (Oudin et al., 2008).

To overcome or circumvent those disadvantages, distinctive flow signatures (i.e., metrics or auxiliary data representing catchment behaviours) in lieu of observed hydrographs  can be used to identify model parameters

~~hydrographs alone. Hingray et al. (2010), for instance, calibrated a runoff model with specific flow signatures relevant to its parameters such as snow accumulation and ablation, recession curves, and rising limb, and subsequently found enhanced performance in hourly runoff prediction in Alpine catchments. Yadav et al. (2007) used spatially extrapolated flow metrics for parameter identification, and found major streamflow indices related to catchment functional behaviours. Euser et al. (2013) proposed a framework for structuring a flexible perceptual model with multiple hydrograph signatures, and evaluated model plausibility. Other examples include use of remotely sensed geomorphological metrics (Fang et al., 2010), isotope concentrations (Son and Sivapalan, 2007), the baseflow index (Bulygina et al., 2009), the spectral density of streamflow observations (Montanari and Toth, 2007; Winsemius et al., 2009), and long term hydrograph descriptors (Shamir et al., 2005).~~

 . The flow duration curve (FDC) has received particular attention in the signature-based model calibrations as a single criterion  (e.g., Westerberg et al., 2014. 2011; . ~~The FDC, the relationship between the frequency and flow magnitudes, provides a summary of temporal streamflow variations at the outlet of a catchment (Vogel and Fennessey (1994). It has been useful for numerous hydrological applications. Vogel and Fennessey (1995) exemplified potential uses of FDCs in hydrological studies including wetland inundation mapping, lake sedimentation studies, instream flow assessment, hydropower feasibility analysis, contaminant and waste management, water resources allocation, and flood frequency analysis. FDCs has been extensively used for runoff prediction (Zhang et al., 2015; Kim and Kaluarachchi, 2014; Smkhtin and Masse, 2000), land use change assessment (Zhao et al., 2012), design of power plants (Liucci et al, 2014), water quality evaluation (Morrison and Bonta, 2008), and catchment classification (Sawicz et al., 2011) among many variations. Along with those applications, FDCs or metrics from FDCs (e.g., the slope of FDCs) were often used as a single calibration criterion(e.g., Westerberg et al., 2011, 2014;Son and Sivapalan, 2007; Yadav et al., 2007) for identifying behavioural parameters. The rationale behind the model calibration against FDCs is that the catchment functional behaviours can be captured byFDCs (Vogel and Fennessey, 1995; Yokoo and Sivapalan, 2011). This hypothesis also made it possible to apply runoff models to FDC prediction (Zhang et al., 2014; Yokoo and Sivapalan, 2011) or investigation of physical controls of FDCs (e.g.,~~the FDC (e.g., Cheng et al., 2012; Ye et al., 2012; Yokoo and Sivaplan, 2011; Bottor et al., 2007). With only few physical parameters, the shape of the period-of-record FDC could be analytically expressed (Botter et al., 2008). Based on this strong relationship between catchment physical properties and the FDC, one may hypothesise that model calibration against the FDC (referred to as the FDC calibration hereafter) can provide parameters that can sufficiently capture actual catchment behaviours. Sugawara (1979) is the first attempt at the FDC calibration, emphasising its advantage to reduce negative effects of epistemic errors in rainfall-runoff data. Westerberg et al. (2011) also highlighted that the FDC

calibration may provide robust predictions to moderate disinformation such as the presence of event flows under inconsistency between inputs and outputs.

If it allows rainfall-runoff models to sufficiently capture functional behaviours of catchments, the  FDC calibration  would have an especial value in comparison to the parameter regionalisation for prediction in ungauged catchment. The parameter regionalisation, which transfers or extrapolates behavioural parameters from gauged to ungauged catchments (e.g., Kim and Kaluarachchi, 2008; Oudin et al., 2008; Parajka et al., 2007; Wagener and Wheater, 2006; Dunn and Lilly, 2011), conveniently provides a priori estimates of behavioural parameters and thus became a popular approach to parameter identification in ungauged catchments (see a comprehensive review in Parajka et al., 2013).  However, it has a critical concern that regionalised parameters are highly dependent on model calibrations at gauged sites  that may have substantial equifinality problems. Under no flow information in ungauged catchments, it is impossible to know whether regionalised parameters are behavioural. Thus, regionalised parameters might be insufficiently reliable and highly uncertain (Bárdossy, 2007; Oudin et al., 2008; Zhang et al., 2008).

On the other hand, the calibration against regional FDCs (referred to as RFDC_cal hereafter) may reduce the primary concern in the classical parameter regionalisation  scheme. The regional models predicting FDCs at ungauged sites have showed strong performance, for instance, via regression analyses between quantile flows and catchment properties (e.g., Shu and Ouarda, 2012; Mohammoud, 2008; Smakhtin et al., 1997), geostatistical interpolation of quantile flows (e.g., Pugliese et al., 2014; Westerberg et al., 2014), and regionalisation of theoretical probability distributions (e.g., Atieh et al., 2017; Sadegh et al., 2016) among many variations. The parameters obtained from RFDC_cal are deemed behavioural, because a distinctive flow signature of the target ungauged catchment directly identifies them; however, predicted  FDCs should be reliable in this case. A FDC is a compact representation of runoff variability at all time scales from inter-annual to event-scale, embedding various aspects of multiple flow signatures (Blöschl et al., 2013).  Based on this strength, several studies already  showed promising predictive performance using RFDC_cal for ungauged catchments (e.g., Westerberg et al., 2014    ; Yu and Yang, 2000).

Nevertheless, practical questions arise when using RFDC_cal for  ungauged catchments. First, the FDC is simplified information with flow magnitudes only; hence, the FDC calibration could worsen the  equifinality problem relative to the hydrograph calibration. Due to no flow  timing information in reginal FDCs, one may cast  uncertainty in predicted FDCs possibly introduced by the regionalisation models (Westerberg et al., 2011; Yu et al., 2002).  RFDC_cal may be undesirable when a simple parameter regionalisation.  can provide better performance, because regionalising observed FDCs may require expensive efforts. Several comparative studies on parameter regionalisation (e.g., Parajka et al.,  2013; Oudin et al., 2008) suggested that the simple proximity-based parameter transfer can be competitive in many regions . Second, there  may be additional flow signatures to improve predictive performance of the FDC calibration. Additional constraining can lead to better predictive performance.  of the RFDC (Westerberg et al., 2014); however, it is still an open question which flow signatures can supplement the FDC calibration.

 As discussed, RFDC_cal seems promising for prediction in ungauged catchments. However, to our knowledge, RFDC_cal has never been evaluated in a comparative manner with classical parameter regionalisation except Zhang et al. (2015), which assessed its performance in part. Therefore, this study aimed to evaluate predictive performance of RFDC_cal in comparison to a conventional parameter regionalisation. We focused on the absence of flow timing in the FDC and its impacts on rainfall-runoff modelling ~~in comparison with the conventional approaches, the hydrograph calibration and the parameter regionalisation for gauged and ungauged catchments respectively. To answer the questions given, we (1) evaluated predictive performance of the hydrograph calibration and the FDC calibration with their uncertainty for gauged catchment, (2) assessed the calibration against regional FDCs in comparison with the proximity-based parameter regionalisation for ungauged catchments, and (3) gauged ability of the FDC calibration to reproduce typical flow signatures.from lumped atmospheric forcing for 45 unregulatedcatchmentsimplyparameter identificationuncertainty assessment. The following section presents~~measure equifinality in the hydrograph and the FDC calibrations.

**2 The study area and data Description of the study area and data**

The study area is 45-gauged45 catchments located across South Korea with no or negligible human-made alterations (e.g., river diversion and dam operations) ininfluences on flow variations were selected for this study (Figure 1). South Korea is characterizedcharacterised as a temperate and semi-humid climate with rainy summer seasons. The North Pacific high-pressure brings monsoon rainfall with high temperatures induring summer seasons, while dry and cold weathers prevail in winter seasons due to the Siberian high-pressure. Typical ranges of annual precipitation are 1200-1500 and 1000-1800 mm in the northern and the southern areas respectively (Rhee and Cho, 2016). Annual mean temperatures in South Korea range between 10 and 15 ℃ (Korea Meteorological Administration, 2011). Approximately, 60-70 percent of precipitation falls in summer seasons from June to September (Bae et al., 2008). Streamflow usually peaks in the middle of summer seasons because of heavy rainfall or typhoons, and hence information of catchment responsebehaviours is largely concentrated on summer-season hydrographs. Snow accumulation and ablation are observedoccurring at high elevations, but their effects have minor influences on temporal flow variations are minor due to the limitedrelatively small amount of winter precipitation (Bae et al., 2008). Annual temperatures range between 10 and 15 ℃ (Korea Meteorological Administration, 2011).

The study catchments shown in Figure 1 were selected based on availability of streamflow data. Although long streamflow data are available at a few river gauging stations, highHigh-quality daily streamflow data across the South Korea have been produced since establishment of the Hydrological Survey CenterCentre in 2007 (Jung et al., 2010). We), though river stages have been monitored for an extensive length at a few gauging stations. Thus, we collected streamflow data at 29 river gauging stations from 2007 to 2015 together with inflow data of 16 multi-purpose dams for the same data period from the Water Resources Management Information System operated by the MistryMinistry of Land, Infrastructure, and Transport of the Korean government (available at http://www.wamis.go.kr/). The selectedmean annual flow of the study catchments are listed in Table was 739 mm yr$^{-1}$ with their climatological featuresa standard deviation of 185 mm yr$^{-1}$ during 2007-2015. As the climaticIn addition, as atmospheric forcing inputs for rainfall runoff modelling, we used gridded , we collected daily precipitation, and maximum and minimum temperatures for 2005-2015 at 3-km grid resolution produced by spatial interpolationinterpolations between 60 stations of the automated surface observing system (ASOS) maintained by the Korea Meteorological Administration. Jung and Eum (2015) combinedThe ASOS data were interpolated by the Parameter-elevation Regression on Independent Slope Model (PRISM; Daly et al., 2008) with), and overestimated pixels of the PRISM grid data were smoothed by the inverse distance method for . Jung and Eum (2015) found that this combined method improved the spatial interpolation, and found improved performance for producing grid of precipitation and the temperatures in South Korea. The annual mean precipitation and temperature datasets across South Korea. For simulating streamflow at outlets of the of the study catchments, we collected the grid climatic data from 2005 to 2015. Annual precipitation and mean temperature in each catchment range vary within ranges of 1145–1997 mm yr$^{-1}$ and 8.0–13.8 ℃

 during 2007-2015. Hydro-climatological features of the 45 catchments are summarised in Table 1.

**3 Methodology**

**3.1 Hydrological model (GR4J)**

A parsimonious rainfall-runoff model, GR4J (Perrin et al., 2003), was adopted to simulate daily hydrographs of the 45 catchments for 2007-2015. GR4J conceptualises functional catchment response to rainfall with four free parameters that regulate the water balance and water transfer functions. Figure 2 schematises the structure of GR4J. The four parameters (X1 to X4) conceptualises soil water storage, groundwater exchange, routing storage, and the base time of unit hydrograph respectively.  Since its parsimonious and efficient structure allows robust calibration and reliable regionalisation of the parameters, GR4J has been frequently used for modelling daily hydrographs with various purposes under diverse climatic conditions (Zhang et al., 2015). The computation details and discussion are found in Perrin et al. (2003). The potential evapotranspiration (PE in Figure 2) was estimated by the temperature-based model proposed by Oudin et al. (2005)  for lumped rainfall-runoff modelling.

**3.2 Preliminary data processing**

Before rainfall-runoff modelling , we preliminarily processed the grid climatic data to convert precipitation data to liquid water forcing  (i.e., rainfall and snowmelt depths) using a physics-based snowmelt model proposed by Walter et al. (2005). The preliminary snowmelt modelling was mainly for reducing systematic errors  from no snow component in GR4J, which may affect model performance in catchments at relatively high elevations. We chose this preliminary processing to avoid adding more parameters (e.g., the temperature index  ) to the existing structure of GR4J. In the case of GR4J, one additional parameter implies 25% complexity increase in terms of the number of parameters, and thus can worsen the equifinality. The snowmelt model uses the same inputs of GR4J to simulate point-scale snow accumulation and ablation processes (i.e., no additional inputs are required). The snowmelt model is a physics-based model but uses empirical methods to estimate its parameters  for the energy balance simulation. As outputs, it produces the liquid water depths and the snow water equivalent

outsputs. After the snowmelt modelling, . For lumped inputs to GR4J, we took spatially averaged pixel values of the liquid water depths and the maximum and minimum temperatures within the boundary of each catchment as lumped inputs to GR4J.

Besides, After the snowmelt modelling, consistency between the spatially averaged liquid water depths and the observed hydrographsflows (i.e., input-output consistency) was checked using the current precipitation index (CPI; Smakhtin and Masse, 2000) defined as:

$$I_t = I_{t-1} \cdot K + R_t \tag{1}$$

where $I_t$ is the CPI (mm) at day t, K is a decay coefficient (0.85 d$^{-1}$), and $R_t$ is the liquid water depth (mm d$^{-1}$) at day t that forces the catchment (i.e., rainfall or snowmelt). CPI mimics temporal variations inof typical streamflow data by converting intermittent rainfallprecipitation data to a continuous time series with an assumption of the linear reservoir. The consistency between modelThe input and output was checked for each catchmentconsistency can be evaluated using correlation between CPI and observed streamflow as in Westerberg et al. (2014) and Kim and Kaluarachchi (2014). The Pearson correlation coefficients between CPI and streamflow data of the 45 catchments had an average of 0.67 with a range of 0.43-0.79, and no outliers were found in the box plot of the correlation coefficients. Hence, we hypothesisedassumed that acceptable consistency existed between climatic forcing and observed hydrographs for parameter calibrationwas acceptable.

**3.2 Rainfall-runoff modelling for3 The hydrograph calibration in gauged catchments**

To search behavioural parameter sets of GR4J using observed runoff time seriesagainst the streamflow observations (i.e., the hydrograph calibration), we used the Monte-Carlo random sampling was used withinwith the parameter ranges given by Demirel et al. (2013). The objective function in Zhang et al. (2015) was chosen as the calibration criterion that considers togetherto consider the Nash--Sutcliffe Efficiency (NSE) and the Water Balance Error (WBE) between observed and modelled hydrographs astogether:

$$OBJ = (1 - NSE) + 5|\ln(1 + WBE)|^{2.5} \tag{2a}$$

$$NSE = 1 - \frac{\sum_{i=1}^{N}(Q_{obs,i} - Q_{sim,i})^2}{\sum_{i=1}^{N}(Q_{obs,i} - \overline{Q_{obs}})^2} \tag{2b}$$

$$WBE = \frac{\sum_{i=1}^{N}(Q_{obs,i} - Q_{sim,i})}{\sum_{i=1}^{N} Q_{obs,i}} \tag{2c}$$

where $Q_{obs}$ and $Q_{sim}$ are the observed and simulated flows respectively, $\overline{Q_{obs}}$ is the arithmetic mean of $Q_{obs}$, and N is the total number of flow observations. The best parameter sets for each study catchment was obtained from minimisation of the OBJ using the Monte-Carlo simulations described below.

To determine a sufficient runs for the random simulations, we calibrated GR4J parameters using the shuffled complex evolution (SCE) algorithm (Duan et al., 1992) for one catchment with highmoderate input-output consistency. Then, the total

number of random simulations was iteratively determined by adjusting the number of runs until the minimum OBJ of the random simulations became adequately close to the OBJ value from the SCE algorithm. We found that approximately 20,000 runs could provide the minimum OBJ value equivalent to that from the SCE algorithm. Subsequently, GR4J was calibrated by 20,000 runs of the Monte-Carlo simulations for all 45 catchments, and the parameter sets with the minimum OBJ values were taken for runoff predictions. In addition, we sorted the 20,000 parameter sets in terms of corresponding OBJ values in ascending order , and first 50 sets (0.25% of the total samples) were taken to measure the degree of equifinality. We measured the equifinality simply by the prediction area between 2.5% and 97.5% boundaries of runoff simulations given by the collected 50 parameter sets. This prediction area was later compared to that from the FDC calibration under the same Monte-Carlo framework. Note that we estimated the prediction area to comparatively evaluate the degree of equifinality between the hydrograph and the FDC calibrations under the same sampling size and the same acceptance rate for all the catchments. For more sophisticated and reliable uncertainty estimation, other methods are available such as the Generalised Likelihood Uncertainty Estimation (GLUE; Beven and Bingley, 1992) and the Differential Evolution Adaptive Metropolis (DREAM; Vrugt and Ter Braak, 2011).

For the hydrograph calibration, the 9-year streamflow data were divided into two parts for calibration (2011-2015) and for validity check (2007-2010), respectively. A two-year warm-up period was used for initialising all runoff simulations in this study.

~~The FDC calibration was also conducted by the same Monte-Carlo sampling but for minimisation of OBJ between observed and modelled quantile flows. We used quantile flows at 103 exceedance probabilities (p of 0.001, 0.005, 99 points between 0.01 and 0.99 at an interval of 0.01, 0.995, and 0.999) to evaluate agreement between observed and simulated FDCs. As did in the hydrograph calibration, the best parameter set was found by 20,000 random simulations and 50 behavioural parameter sets were taken.~~

**4 Model calibration against the regional FDC for ungauged catchments**
* * *
Each catchment was treated ungauged for the comparative evaluation of RFDC_cal in the leave-one-out cross-validation (LOOCV) mode. For regionalising empirical FDCs, the geostatistical method recently proposed by Pugliese et al. (2014) was used . Pugliese et al. (2014) employed the top-kriging method (Skøien et al., 2006) to spatially interpolate the total negative deviation (TND), which is defined as the area between the mean annual

flow and below-average flows in a normalised FDC. The top-kriging weights that interpolate TND values were taken as weights to estimate flow quantiles of ungauged catchments from empirical FDCs of surrounding gauged catchments.  The FDC of an ungauged catchment in Pugliese et al. (2014) is estimated from normalised FDCs of surrounding gauged catchments as:

$$\widehat{\Phi}(w_0, p) = \widehat{\phi}(w_0, p) \cdot \overline{Q}(w_0) \tag{3a}$$

$$\widehat{\phi}(w_0, p) = \sum_{i=1}^{n} \lambda_i \cdot \phi_i(w_i, p), \quad p \in (0,1) \tag{3b}$$

where $\widehat{\Phi}(w_0, p)$ is the estimated quantile flow (m$^3$ s$^{-1}$) at an exceedance probability p (unitless) for an ungauged catchment $w_0$, $\widehat{\phi}(w_0, p)$ is the estimated normalised quantile flow (unitless), $\overline{Q}(w_0)$ is the annual mean streamflow (m$^3$ s$^{-1}$) of the ungauged catchment, and $\phi_i(w_i, p)$ and $\lambda_i$ are normalised quantile flows (unitless) and corresponding top-kriging weights (unitless) of gauged catchment $w_i$, respectively. The unknown mean annual flow of an ungauged catchment, $\overline{Q}(w_0)$, can be estimated with a rescaled mean annual precipitation defined as:

$$MAP^* = 3.171 \times 10^{-5} \cdot MAP \cdot A \tag{4}$$

where MAP* is the rescaled mean annual precipitation (m$^3$ s$^{-1}$), MAP is mean annual precipitation (mm yr$^{-1}$) and A is the area (km$^2$) of the ungauged catchment, and the constant 3.171×10$^{-5}$ converts the unit of MAP$^*$ from mm yr$^{-1}$ km$^2$ to m$^3$ s$^{-1}$.

A distinct advantage of the geostatistical method is its ability to estimate the entire flow quantiles in a FDC with a single set of top-kriging weights. Since a parametric regional FDC (e.g., Yu et al., 2002; Mohamoud, 2008) is obtained from independent models for each flow quantile in many cases, for instance, by multiple regressions between selected quantile flows and catchment properties, fundamental characteristics in a FDC continuum would be entirely or partly lost. The geostatistical method, on the other hand, treats all flow quantiles as a single object; thereby, features in a FDC continuum can be preserved. It showed promising performance to reproduce empirical FDCs only using topological proximity between catchments. More details on the geostatistical method are found in Pugliese et al. (2014).

For regionalising empirical FDCs of the 45 catchments, we followed the same procedure of Pugliese et al. (2014). We obtained top-kriging weights ($\lambda_i$) by the geostatistical interpolation of TND values from observed FDCs for the calibration period (2011-2015). Then, the top-kriging weights were used to interpolate empirical flow quantiles. The number of neighbours for the TND interpolation was iteratively determined as five at which additional neighbouring TNDs are unlikely to bring better agreement between the estimated and observed TNDs. In other words, normalised flow quantiles of five catchments surrounding the target ungauged catchment were interpolated with the top-kriging weights. Then, MAP$^*$ of  the target ungauged

catchment was multiplied. We predicted flow quantiles at 103 exceedance probabilities. Against the (p of 0.001, 0.005, 99 points between 0.01 and 0.99 at an interval of 0.01, 0.995, and 0.999) for rainfall-runoff modelling against regional FDCs, parameters of GR4J were directly calibrated for each catchment. The parameters were identified in the same manner of 20,000 runs of the Monte-Carlo simulations, but towards minimisation of the OBJ value between regional and modelled FDCs. (i.e., RFDC_cal).

For runoff prediction in ungauged catchments, the GR4J parameters were identified by the same Monte-Carlo sampling but toward minimisation of OBJ value between the regional and the modelled flow quantiles at the 103 exceedance probabilities. The best parameter set, which provided the minimum OBJ value, was taken as the best behavioural set of RFDC_cal for each catchment.

**3.2.25 Proximity-based parameter regionalizationregionalisation for ungauged catchments**

As a counterpart of the calibration against regional FDCs, we usedWe selected the proximity-based parameter transfer for prediction in ungauged catchments.(referred to as PROX_reg hereafter) to comparatively evaluate predictive performance of RFDC_cal. The parameter regionalisation can be classified intohas three typicalclassical categories: (a) proximity-based parameter transfer (i.e., PROX_reg; e.g., Oudin et al., 2008); (b) similarity-based parameter transfer (e.g., McIntyre et al., 2005); and (c) regression between parameters and physical properties of gauged catchments (e.g., Kim and Kaluarachchi, 2008). Based on itsA comprehensive review on the parameter regionalisation in Parajka et al. (2013) reported that PROX_reg has competitive performance under humid climate with low-complexity models relative to the other categories. Based on modelling conditions in this study (semi-humid climate and simplicity (Oudin et al., 2008; Parajka et al., 20134 parameters), we chose the proximity-based parameter regionalisationPROX_reg to evaluate RFDC_cal.

For prediction in ungauged catchment, five donor catchments chosen for the FDC regionalisation were again used for transferring their parameter sets to each catchment of interest. To be consistent between two proximity-based approaches, we synchronised donor catchments. The five runoff simulations were averaged for representing modelled hydrographs for each catchment.

To predict runoff at the 45 catchments in the LOOCV mode, we transferred the behavioural parameter sets obtained from the hydrograph calibration of the five donor catchments used for the FDC regionalisation. In other words, we used the same donor catchments for FDC regionalisation and PROX_reg. This allows us to have consistency in transferring hydrological information from gauged to ungauged catchments between RFDC_cal and PROX_reg. Using the best behavioural parameter sets of the five donor catchments, we generated five runoff time series and took the arithmetic averages of them to represent runoff predictions by PROX_reg.

**3.3 Evaluation of 6 Performance evaluation**

We used multiple performance metrics to evaluate predictive performance and uncertainty Two performance measures were used for evaluating model predictiveof all modelling approaches applied in this study. Predictive performance. One is NSE

 of each modelling approach was graphically evaluated using box plots of the performance metrics of the 45 catchments. In addition, we performed several paired t-tests to check the statistical significance of performance differences between  the modelling approaches. Following is the description of the performance metrics.

To measure high- and low-flow reproducibility, we chose two traditional performance metrics, (1) the NSE between observed and predicted flows (Eq. 2b) and  NSE of log-transformed flows (LNSE) respectively. LNSE is calculated as:

$$\text{LNSE} = 1 - \frac{\sum_{i=1}^{N}\left(\ln(Q_{obs,i}) - \ln(Q_{sim,i})\right)^2}{\sum_{i=1}^{N}\left(\ln(Q_{obs,i}) - \overline{\ln(Q_{obs})}\right)^2}$$
(5)

Though NSE and LNSE are frequently used for performance evaluation, they may be sensitive to errors in ~~predicted flows was quantified by the area between the lower and upper bounds of simulated hydrographs. We took a ratio of uncertainty of the FDC calibration to that of the hydrograph calibration for each catchment and defined it as the uncertainty ratio. Note that this assessment was not to estimate absolute uncertainty but to measure relative uncertainty gained by replacing a hydrograph with a FDC for model calibration.~~

flow observations (Westerberg et al., 2011). Hence, we additionally selected three typical flow metrics that embed dynamic flow variation in a compact manner; the runoff ratio ($R_{QP}$), the baseflow index ($I_{BF}$), and the rising limb density ($D_{RL}$). $R_{QP}$, $I_{BF}$, and $D_{RL}$ are proxies of aridity and water holding capacity, contribution of the baseflow to flow variations, and flashness of catchment behaviours, respectively. They are defined as the ratio of runoff to precipitation, the ratio of  baseflow to total runoff, and the inverse of average time to peak (d$^{-1}$) as:

$$R_{QP} = \frac{\overline{Q}}{\overline{P}}$$
(6a)

$$I_{BF} = \sum_{t=1}^{T} \frac{Q_{B,t}}{Q_t}$$
(6b)

$$D_{RL} = \frac{N_{RL}}{T_R}$$
(6c)

where $\overline{Q}$ and $\overline{P}$ are average flow and precipitation for a given period (mm d$^{-1}$), $Q_t$ and $Q_{B,t}$ (m d$^{-1}$) is the  streamflow and the base flow at time t respectively, $N_{RL}$ is the number of rising limb, and $T_R$ is the total amount of time when the hydrograph is rising (days). $Q_{B,t}$ can be calculated by subtracting direct flow $Q_{D,t}$ from $Q_t$ as:

$$Q_{D,t} = c \cdot \text{}Q_{D,t-1} + 0.5 \cdot (1+c) \cdot (Q_t - Q_{t-1})$$
(7a)

$$Q_{B,t} = Q_t - Q_{D,t} \quad\quad\quad\quad\quad\quad\quad\quad\quad\quad\quad\quad\quad\quad\quad\quad\quad\quad\quad\quad\quad\quad\quad (7b)$$

where  the filter parameter , which was set to 0.925 (Brooks et al., 2011; Eckhardt  2007).

Flow signature reproducibility of RFDC_cal and PROX_reg were evaluated by the relative absolute bias between modelled and observed signatures as:

$$D_{FS} = \frac{|FS_{sim} - FS_{obs}|}{FS_{obs}} \quad\quad\quad\quad\quad\quad\quad\quad\quad\quad\quad\quad\quad\quad\quad\quad\quad (8)$$

where $D_{FS}$ is the relative absolute bias, $FS_{sim}$ is a flow signature of the modelled flows, and $FS_{obs}$ is that of the observed flows.

**4 Results**

**4.1  Hydrograph calibration and FDC regionalisation in gauged catchments**

~~The box plots in Figure 3 comparatively show distributions of NSE and LNSE values between observed and modelled flows. It was clearly indicated that the hydrograph outperformed the FDC calibration in prediction of high flows. The NSEs of the hydrograph calibration were generally greater than those of the FDC calibration for the both calibration and validation periods. The FDC calibration was of much wider NSE ranges than the hydrograph calibration and thus greater uncertainty in high flow prediction. The prediction results tend to have greater medians of NSEs for the calibration periods than the validation period. Because the term NSE is directly used for calibration, the parameter identification could be slightly inclined towards reproduction of high flows for the calibration period. The significantly increasing NSE ranges from the calibration to validation periods in Figure 3b may imply that the FDC calibration has weaker temporal parameter transferability from one period to another. In low-flow prediction, the FDC calibration showed slightly weaker performance than the hydrograph calibration. Although the LNSE medians of the FDC calibration were comparable to those of the hydrograph calibration, LNSEs of the FDC calibration also showed wider ranges than the hydrograph calibration. The FDC calibration was still likely to yield significant uncertainty in low flow predictions when parameters were temporally transferred. Unlike the NSE comparison, the median LNSE values did not decrease from the calibration to the validation periods for the both hydrograph and the FDC calibrations. This would imply that the behavioural parameter sets have more temporal consistency in low flows than high flows.~~

~~Figure 4 illustrates 1:1 scatter plots between the performance measures and correlation between CPI and observed hydrographs, indicating that consistency between model input and output meaningfully affects predictive performance of rainfall-runoff models. The performance measures were generally in positive relationships with correlation between CPI and observed hydrographs. Adequate input-output consistency seems to be a prerequisite of parameter identification to attain good high-flow predictability especially for the hydrograph calibration. For having 0.6 or more NSE, the correlation~~

coefficient between CPI and observed flows should be greater than 0.6 approximately. On the other hand, predictability of low flows was achieved with relatively low input out consistency. LNSEs less than 0.4 were rarely observed than NSEs for the both hydrograph and FDC calibrations. Interestingly, the FDC calibration appears to have better predictability in low flows despite the use of NSE for parameter calibration, which is a sensitive measure to high flow reproducibility. It implies that the FDC calibration has some deficiency to capture catchment response to storm events even with adequate model input-output consistency whereas it performs well for long-term low flow or baseflow predictions.

Shortly, the FDC calibration could lead to relatively low predictive power with increased uncertainty when adopted as an alternative of the hydrograph calibration. Low predictability in high flows can be a particular concern of the FDC calibration. The simplification of flow information appears to exacerbate the equi-finality in parameter identification. This weakness of the FDC calibration was confirmed by the uncertainty bounds of modelled hydrographs in Figure 5. The collection of 50 parameter sets from the FDC calibration showed less robust simulations than the hydrograph calibration for the three catchments even though their FDCs were fairly well reproduced by the FDC calibration. For the 45 catchments, the mean NSE between observed and modelled FDCs was 0.95 when using the FDC calibration. In other words, parameters reproducing observed FDCs generally were less unique to represent catchment functional behaviours than ones reproducing observed hydrographs. The equi-finality in the FDC calibration is likely to get worse with decreasing performance of the hydrograph calibration (Figure 6). On average, uncertainty of predicted hydrographs was doubled for the 45 catchments when the FDC calibration substitutes for the hydrograph calibration. The prediction results from the 45 gauged catchments, hence, suggest that parameter identification with compact information of FDCs could yield weaker performance and less parameter identifiability than the hydrograph calibration.

**4.2 Geostatistical FDC regionalisation**

Figure 7a illustrates the 1:1 scatter plot between observed and estimated TNDs of the 45 catchments. The correlation coefficient between empirical and estimated TNDs was Figure 3a displays results of the parameter identification against the observed hydrographs (i.e., the hydrograph calibration). The 45 catchments had the mean NSE and LNSE of 0.66 and 0.65 between the simulated and observed flows for the calibration period, respectively. The average NSE reduction from the calibration to the validation periods was 0.06 with a standard deviation of 0.10. The temporal transfer of the calibrated parameters did not decrease the mean LNSE value, while a wider LNSE range indicates that uncertainty of low-flow predictions may increase when temporally transferring the calibrated parameters.

The predictive performance was closely related to the input-output consistency (Figure 3b), which was measured by the Pearson correlation coefficient between the CPI and the observed flows. A low input-output consistency implies that the rainfall-runoff data may include significant epistemic errors such as minimal flow responses to heavy rainfall or excessive response to tiny rainfalls. If the model calibration compensates disinformation from such errors, the parameters would be forced to have biases. Figure 3b shows that consistency in input-output data is a critical factor affecting parameter identification and thus performance. Perhaps, screening catchments with low input-output consistency may provide better

predictions in ungauged catchments. However, we did not consider it in the LOOCV for RFDC_cal and PROX_reg, since variation in input-output consistency would be a common situation. Rather, reducing the number of gauged catchments lowers spatial proximity and thus can cause biases for ungauged catchments too. Overall, 27 catchments and 33 catchments showed NSE and LNSE values greater than 0.6. We assumed the hydrograph calibration under the Monte-Carlo framework, which was assisted by the SCE optimisation, was able to acceptably identify the behavioural parameters under given data quality.

Besides, Figure 4 illustrates the 1:1 scatter plot between the observed and predicted flow quantiles of all the catchments, indicating high applicability of the top-kriging FDC regionalisation. The overall NSE and LNSE values between the observed and regionalised flow quantiles show good applicability of the geostatistical method. The NSE and LNSE values for individual catchments have averages of 0.83 and 0.91 with standard deviations of 0.25 and 0.11, respectively, implying that low-flow predictions were slightly better. The performance of the geostatistical method was relatively poor at locations where gauging density is low. 0.56 (equivalent to 0.30 NSE). The relatively poor prediction of TNDs was likely from the use of annual precipitation for normalising quantile flows. In the original study of the geostatistical method (Pugliese et al., 2014), the TND prediction became poorer (NSE was decreased from 0.81 to 0.60) when using the rescaled annual precipitation instead of observed mean annual flow. Uncertainty introduced by estimation of mean annual flows might influence predictive power of the geostatistical TND interpolation. Another likely reason is that TND is a complex signature of streamflow regime; yet, it could be descriptive in terms of functional similarity between catchments (Pugliese et al., 2016). It may be difficult to completely capture spatial variation of TNDs with topological proximity only. However, Pugliese et al. (2016) also argued that poor prediction of TND did not automatically result in poor quantile flow predictions. Their comparative study achieved successful FDC predictions for 182 catchments in the United States (0.95 of median NSE) using the top-kriging weights of TNDs in spite of low TND predictability. Though it is an outside scope of this study, a further study needs to be directed towards effects of TND prediction on the FDC regionalisation. Because it is still unclear whether or not descriptors from FDCs well predict flow quantiles, top-kriging weights of various flow signatures need to be tested for improving the geostatistical FDC prediction as well.

The high performance in FDC prediction with poor TND prediction was replicated in this study. Overall NSE and LNSE values between observed and predicted quantile flows of the 45 catchments suggest good applicability of the geostatistical method to the study catchments (Fig 7b). The averages of individual NSEs and LNSEs for each catchment were 0.83 and 0.91 with standard deviations of 0.25 and 0.11 respectively. The higher LNSEs imply that performance of the geostatistical method is better for low flows. This might be because the top-kriging weights interpolating TNDs were obtained from below average flows only. No information of above-average flows reflected in TNDs might incline the FDC regionalisation towards low flow predictions. Low predictive power of the regional FDC model was found at locations with low gauging density. Catchments 4, 10, 35, and 36, which recorded 0.6 or less NSEs, were are limitedly hatched with no hatching catchments and/or limited adjacent to the other catchments; nonetheless, LNSEs of those catchments were still greater than 0.7. This result wasis consistent with a finding of Pugliese et al. (2016) that performance of the geostatistical method was

highly sensitive to river gauging density. Transferring  flow quantiles from remote catchments may not sufficiently capture functional similarity  between donor and receiver catchments. In spite of the minor shortcomings, the geostatistical FDC regionalisation was deemed acceptable based on the high NSE and LNSE of flow quantiles. Topological proximity was generally a good predictor of flow quantiles for the study catchments.

**4.2 Comparing hydrograph predictability between RFDC_cal and PROX_reg**

Figure 5 compares the box plots of NSE and LNSE values between RFDC_cal and PROX_reg. PROX_reg generally outperforms RFDC_cal in predicting both high and low flows, suggesting that transferring parameters identified by observed hydrographs would be a better choice than a local calibration against predicted FDCs. The differences between NSE values of PROX_reg and RFDC_cal have an average of 0.22 with a standard deviation of 0.34. Only 8 catchments showed higher NSEs with RFDC_cal. These higher NSE values of PROX_reg imply that PROX_reg is preferable when high-flow predictability is needed such as flood analyses. In the case of LNSE, PROX_reg still had a higher median than RFDC_cal (0.53 and 0.62 for RFDC_cal and PROX_reg respectively). In 25 catchments, PROX_reg provided LNSE values greater than those of RFDC_cal.

The low performance of RFDC_cal was also found in the comparative assessment of Zhang et al. (2015), which evaluated RFDC_cal for 228 Australian catchments using the same GR4J model. Zhang et al. (2015) found that RFDC_cal was inferior to PROX_reg in the Australian catchments, because the FDC calibration poorly reproduced temporal flow variations relative to the hydrograph calibration. This study confirms the difficulty to capture dynamic catchment behaviours with FDCs containing no flow timing information.

A major weakness of RFDC_cal is the absence of flow timing information in the parameter calibration process. Unlike RFDC_cal, PROX_reg did not discard the flow timing information. The regionalised parameters may be able to implicitly transfer the flow timing information from gauged to ungauged catchments (this hypothesis will be discussed later in Section 4.4). Figure 6 illustrates how the absence of flow timing negatively influences on predictive performance. For this comparison, the parameters were recalibrated against the observed FDCs (not regional FDCs) under the same Monte Carlo method to discard errors introduced by the FDC regionalisation (i.e., equivalent to calibrations against perfectly regionalised FDCs). The parameters identified by the observed hydrograph (Figure 6a) brought a good predictability in both high and low flows, resulting in an excellent performance to reproduce the FDC. On the other hand, an excellent FDC reproducibility does not guarantee a good predictability in high flows (Figure 6b). This indicates that reproducing FDCs with rainfall-runoff models would be less sufficient than the hydrograph calibration to capture functional catchment responses.

In addition, Figure 6 shows that the prediction area of the 50 behavioural parameters from the Monte-Carlo simulations (indicated by the grey areas and the blue arrows) became much larger when using the FDC calibration instead of the hydrograph calibration. We calculated the ratio of the prediction area of the FDC calibration to that of the hydrograph calibration, and defined it as the equifinality ratio. It quantifies the degree of equifinality augmented by replacing the

hydrograph calibration with the FDC calibration. Figure 7 displays the scatter plot between the equifinality ratio and the input-output consistency. The equifinality augmented by the loss of flow timing is likely to increase as the input-output consistency decreases. The average of the equifinality ratios was 1.96, implying that potential equifinality inherent in RFDC_cal could be substantial. This may suggest that the equifinality problem embedded in RFDC_cal could be more significant than that in PROX_reg.

**4.3 Comparing flow-signature predictability between RFDC_cal and PROX_reg**

Figure 8 summarises performance of RFDC_cal and PROX_reg to regenerate three flow signatures of $R_{QP}$, $I_{BF}$, and $D_{RL}$. RFDC_cal is competitive in reproducing the averaged-based signatures $R_{QP}$ and $I_{BF}$, while it showed relatively a weak ability to regenerate the event-based signature $D_{RL}$. $R_{QP}$ and $I_{BF}$ are flow metrics based on averages of long-term flow and precipitation in which no flow timing information is involved. Especially, RFDC_cal showed strong performance in reproducing $I_{BF}$ relative to PROX_reg. This result can be explained by considering that baseflow has less temporal variations than direct runoff in the Korean catchments under typical monsoonal climate. High seasonality of monsoonal precipitation makes high temporal variations in direct runoff during June to September, while relatively steady baseflow is dominant during dry seasons (October to May). In Catchment 2 whose flow variation is displayed in Figure 6, for example, the coefficient of variance (CV) of direct runoff was 5.86 for 2007-2015, which is approximately 3.5 times as high as that CV of baseflow.

On the other hand, RFDC_cal was poorer to reproduce $D_{RL}$ than PROX_reg. This highlights the weakness of RFDC_cal in which only flow magnitudes were used for identifying model parameters. PROX_reg showed better performance to predict $D_{RL}$ than RFDC_cal. Flow timing information gained from the observed hydrographs might be preserved, even after behavioural parameters were transferred to ungauged catchments. Overall, PROX_reg seems to be better than RFDC_cal to predict the three flow signatures together.

The box plots in Figure 9 provide an indication that $D_{RL}$ is likely to supplement the FDC calibration and thus improve RFDC_cal. From the collection of 50 behavioural parameter sets given by the FDC calibration, we chose the parameter set providing the lowest bias for each flow signature as the best behavioural sets, and simulated runoff again for all catchments. The high-flow predictability was fairly improved by additional constraining with $D_{RL}$, suggesting that flow metrics associated with flow timing makes up for the weakness of the FDC calibration. Additional constraining with $R_{QP}$ and $I_{BF}$ did not bring appreciable improvement in the FDC calibration. However, PROX_reg was still better than the additional constraining with $D_{RL}$, indicating that a further study is needed for better constraining rainfall-runoff models using FDCs together with additional flow metrics.

**4.4 Paired t-tests between the modelling approaches**

For comparative evaluation in this study, we produced several runoff prediction sets using multiple rainfall-modelling approaches. First, we calibrated GR4J against the observed hydrographs (referred to as Q_cal), and transferred the

behavioural parameters to ungauged catchments in the LOOCV mode (PROX_reg). We constrained GR4J with the regional FDCs (RFDC_cal). To evaluate equifinality, we recalibrated the GR4J parameters against the observed FDCs (referred to as FDC_cal). Additionally, we constrained the model with observed FDCs plus the flow signatures, and significant performance improvement was found with $D_{RI}$ (referred to as FDC+$D_{RI}$_cal). A paired t-test using the performance metrics (NSE, LNSE, or $D_{FS}$) between these modelling approaches can answer various questions beyond the graphical evaluations with box plots. For paired t-tests, we added one more case of transferring parameters gained from FDC_cal to ungauged catchments (referred to as FPROX_reg). FPROX_reg transfers behavioural parameters with no flow timing information from gauged to ungauged catchments. The mean NSE of FPROX_reg was 0.44 with a standard deviation of 0.49.

A primary hypothesis of this study was that RFDC_cal could outperform PROX_reg. This question can be addressed by NSE differences between RFDC_cal and PROX_reg. The mean NSE difference between them was -0.22 and the standard error was 0.051, providing an evaluation that the NSE differences were less than zero at a 95% confidence level. The paired t-test did not lend support the hypothesis (i.e., PROX_reg outperformed RFDC_cal significantly). However, we could assume that $D_{RI}$ could improve predictive performance of FDC_cal. The mean NSE difference between FDC+DRI_cal and FDC_cal was 0.12 and the standard error was 0.025, confirming the significance at a 95% confidence level.

Likewise, we tested several questions relevant to rainfall-runoff modelling in ungauged catchments using different combinations. One interesting question would be "Did the behavioural parameters from Q_cal contain flow timing information for ungauged catchments?" We addressed this question by comparing between PROX_reg and FPROX_reg with a hypothesis that predictability in ungauged catchments would decrease if the regionalised parameters were gained only from flow magnitudes. FPROX_reg uses FDC_cal for searching behavioural parameters at gauged catchments; thereby, it cannot transfer flow timing information to ungauged catchments through the behavioural parameters. The mean NSE difference between PROX_reg and FPROX_reg was 0.10, and the standard error was 0.031. The NSE differences were greater than zero significantly. The behavioural parameters from Q_cal were likely to have flow timing information affecting predictability in ungauged catchments. In Table 3, we summarised the results of paired t-tests for scientific questions that may arise from this study. They could be beneficial information for rainfall-runoff modelling in ungauged catchments.

**5 Discussion and conclusions**

**5.1 RFDC_cal for rainfall-runoff modelling in ungauged catchments**

The use of regional FDCs as a single calibration criterion appears to be a good choice for searching behavioural parameters in ungauged sites. As discussed earlier, the FDC is a compact representation of runoff variability at all time scales, and thus able to embed multiple hydrological features in catchment dynamics (Blöschl et al., 2013). A pilot study of Yokoo and Sivapalan (2011) discovered that the upper part of an FDC is controlled by interaction between extreme rainfall and fast runoff, while the lower part is governed by baseflow recession behaviour during dry periods. The middle part connecting the upper and the lower parts is related to the mean within year flow variations, which is controlled by interactions between

water availability, energy, and water storage (Yager et al., 2012; Yokoo and Sivapalan, 2011). It is well-documented that hydro-climatological processes within a catchment are reflected in the FDC (e.g., Cheng et al., 2012; Ye et al., 2012; Coopersmith et al., 2012; Yaeger et al., 2012; Botter et al., 2008), and therefore the model parameters identified solely by a regional FDC are expected to provide reliable predictions in ungauged catchments (e.g., Westerberg et al., 2014; Yu and Yang, 2000).

The comparative evaluation in this study, however, provides another expected lesson that the FDC calibration is good to reproduce the FDC itself, but it insufficiently captures functional responses of catchments due to the absence of flow timing information. A hydrograph is the most complete flow signature embedding numerous processes interacting within a catchment (Blöschl et al., 2013), being more informative than an FDC. Since any simplification of a hydrograph, including the FDC, should lose some amount of flow information, it is no surprise that the FDC calibration worsens the equifinality. This study emphasises that the absence of flow timing in RFDC_cal may cause larger prediction errors than regionalised parameters gained against observed hydrographs. The paired t-test between PROX_reg and FPROX_reg highlights that regionalised parameters gained from observed hydrographs were likely to contain intangible flow timing information even for ungauged catchments. The flow timing information implicitly transferred to ungauged catchment is a major gap between PROX_reg and RFDC_cal. The errors introduced by the FDC regionalisation were not significant due to high performance of the geostatistical method in this study.

Because the hydrograph calibration can compensate the errors in input-output data, one may convert the hydrograph into the FDC to avoid effects of disinformation on rainfall-runoff modelling. However, in this case, valuable flow timing information should be paid in trade-off. For RFDC_cal in this study, we began with converting the observed hydrographs into the flow quantiles to regionalise them; thus, the flow timing information was initially lost. As shown, the performance of RFDC_cal was generally lower than that of PROX_reg. Therefore, when condensing observed hydrographs into flow signatures, preserving all available flow information in the hydrograph would be a key for a successful rainfall-runoff modelling. This study shows only using regionalised FDCs could lead to less reliable rainfall-runoff modelling in ungauged catchments than regionalised parameters. An FDC is unlikely to preserve all flow information in a hydrograph necessary for rainfall-runoff modelling.

**5.2 Suggestions for improving RFDC_cal**

Westerberg et al. (2014) suggested the necessity of further constraining to reduce predictive uncertainty in RFDC_cal. This study found that RFDC_cal could provide comparable performance to regenerate the flow signatures within which flow magnitudes are only involved (i.e., $R_{OP}$ and $I_{BF}$). To supplement regional FDCs, flow signatures associated with flow timing seems to be essential. Figure 9 shows potential of additional constraining with $D_{RL}$, and Q2 in Table 3 confirms it. Other flow signatures in temporal dimensions such as the high- and the low-flow event durations in Westerberg and McMillan (2015) can be candidates to improve RFDC_cal. However, uncertainty in those flow signatures will be a challenge to build regional models for ungauged catchments (Westerberg et al., 2016).

An alternative method of RFDC_cal is to directly regionalise hydrographs to ungauged catchments (e.g., Viglione et al., 2013). In data-rich regions, topological proximity could better capture spatial variation of daily flows than rainfall-runoff modelling with regionalised parameters (Viglione et al., 2013). Although a dynamic model may be required for regionalising observed daily flows at an expensive computational cost, flow timing information would be contained in regionalised hydrographs. The parameter identification against the regional hydrographs may become a better approach than RFDC_cal and/or other signature-based calibrations.

**5.3 Limitations and future research directions**

There are caveats in our comparative evaluation. First, uncertainty in input-output data was not considered in our assessment. McMillan et al. (2012) reported typical ranges of relative errors in discharge data as 10-20% for medium to high flow and 50-100% for low flows. We assumed that quality of the discharge data was adequate. However, other methods objectively considering uncertainty could better estimate model performance and the equifinality (e.g., Westerberg et al., 2011, 2014

~~The box plots in Figure 8 present predictive performance of the calibration against regional FDCs (referred to as RFDC_cal hereafter) in comparison with the proximity-based parameter regionalisation (referred to as PROX_reg hereafter). The performance measures between observed and modelled hydrographs were computed for the entire period of streamflow data (2007-2011). Distributions of NSEs clearly showed that PROX_reg outperforms the FDC calibration in prediction of high flows (Figure 8a), indicating that a priori parameter sets from neighbouring catchments should perform even better than local calibrations against observed FDCs. The average difference between NSEs of PROX_reg and RFDC_cal was 0.18 with a standard deviation of 0.25. RFDC_cal outperformed PROX_reg only for 8 out of the 45 catchments. LNSEs with PROX_reg were still of a slightly higher median than RFDC_cal. Although RFDC_cal appears to have comparable predictability in low flows, 31 out of 45 catchments were having greater LNSEs with PROX_reg. The results for ungauged catchments bring the same intuition as the case for gauged catchment that a priori parameter sets obtained from nearby gauged catchments seem to be more desirable than local parameter identification against regional FDCs.~~

). Second, we used a conceptual runoff model with a fixed structure for all the catchments. Uncertainty from the model structure would vary across the study catchments; nevertheless, the structural uncertainty was not measured here. Our comparative assessment was based on the basic premise that modelling conditions should be fixed for all study catchments. Finally, though the proximity-based parameter regionalisation was good for the Korean catchments, comparison between RFDC_cal and other regionalisation methods, such as the regional calibration and the similarity-based parameter transfer, may provide beneficial information for rainfall-runoff modelling in ungauged catchments. Comparative assessment between RFDC_cal and other parameter regionalisation using more sample catchments under diverse climates will provide more meaningful lessons.

We can no longer hypothesise that the parameters gained against regionalised FDCs would perform sufficiently, because an FDC contains less information than a hydrograph (i.e., the absence of flow timing). For improving RFDC_cal, we suggested

to supplement RFDC_cal with flow signatures in temporal dimensions. Then, a question should be addressed on how to make flow signatures more informative than (or equally informative to) hydrographs. It may be impossible only using flow signatures originated from hydrographs (e.g., mean annual flow, baseflow index, recession rates, FDCs, etc.). Combinations of those signatures are unlikely more informative than their origins (i.e., hydrographs), though it depends on how much disinformation is present in the observed flows. Future research topics may include finding new signatures that supplement hydrographs, and how to combine them with existing flow signatures for rainfall-runoff modelling in ungauged catchments.

**5.4 Conclusions**

While the rainfall-runoff modelling against regional FDCs appeared a good approach for prediction in ungauged catchments, this study highlights its weakness in the absence of flow timing information, which may cause poorer predictive performance than the simple proximity-based parameter regionalisation. The following conclusions are worth emphasising:

(1) For ungauged catchments in South Korea where spatial proximity well captured functional similarity between gauged catchments, the model calibration against regional FDCs is unlikely to outperform the conventional proximity-based parameter transfer for daily runoff prediction. The absence of flow timing information in regional FDCs seems to cause a substantial equifinality problem in the parameter identification process and thus lower predictability.

(2) The model parameters gained from observed hydrographs would contain flow timing information even for ungauged catchments. This intangible flow timing information should be discarded if one calibrates a rainfall-runoff model against regional FDCs. This information loss may reduce predictability in ungauged catchments significantly.

(3) To improve the calibration against regional FDCs, flow metrics in temporal dimensions, such as the rising limb density, need to be included as additional constraints. As an alternative approach, if river gauging density is high, regionalised hydrographs preserving flow timing information can be used for local calibrations at ungauged catchments.

(4) For better predictions in ungauged catchments, it is necessary to find new flow signatures that can supplement the observed hydrographs. How to combining them with existing information will be a future research topic for rainfall-runoff modelling in ungauged catchments.

[revised manuscript text omitted]

Blöschl, G., Sivapalan, M., Wagener, T., Viglione, A., and Savenije,. H.: Runoff Prediction in Ungauged Basins, Simthesis

20 across Processes, Places, and Scales. Cambridge University Press, New York, USA, 2013.

Botter, G., Porporato, A., Rodriguez-Iturbe, I. and Rinaldo, A.: Basin-scale soil moisture dynamics and the probabilistic characterization of carrier hydrologic flows: Slow, leaching-prone components of the hydrologic response, Water Resour. Res., 43,

25 W02417, doi:10.1029/2006WR005043, 2007.

Botter, G., Zanardo, S., Porporato, A., Rodriguez-Iturbe, I., and Rinaldo, A.: Ecohydrological model of flow duration curves and annual minima, Water Resour. Res., 44, W08418, doi:10.1029/2008WR006814., 2008.

Brooks, P. D., Troch P. A., Durcik, M., Gallo, E., and Schlegel, M.: Quantifying regional-scale ecosystem response to

30 changes in precipitation: Not all rain is created equal, Water Resour. Res., 47, W00J08, doi:10.1029/2010WR009762, 2011.

Cheng, L., Yaeger, M., Viglione, A., Coopersmith, E., Ye, S., and Sivapalan, M.:

analysisExploring the physical controls of regional patterns of flow duration curves – Part 1: Insights from statistical analyses, Hydrol. Earth Syst. Sci., 13, 893-90416, 4435-4446, doi:10.5194/hess-16-4435-2012, 2012.

[revised manuscript text omitted]

and Sivapalan, M.: Improving model structure and reducing parameter uncertainty in conceptual water balance models through the use of auxiliary data, Water Resour. Res., 43, W01415, doi:10.1029/2006WR005032, 2007.

Tian, Y., Xu, Y. P., and Zhang, X. J.: Assessing of climate change impacts on river high flow through comparative use of GR4J, HBV, and Xinanjiang models. Water Reour. Manage., 27, 2871-2888, doi:01.1007/s11269-013-0321-4, 2013.

van Werkhoven, K., Wagener, T., Reed, P., and Tang, Y.: Sensitivity-guided reduction of parametric dimensionality for multi-objective calibration of watershed models, Adv. Water Resour., 32, 1154-1169, doi:10.1016/j.advwatres.2009.03.002, 2009.

Vogel, R. M., and Fennessey, N. M.: Flow duration curves I: A new interpretation and confidence intervals, J. Water Resour. Plann. Manage., 120, 485-504. 1994.

Vogel, R. M., and Fennessey, N. M.: Flow duration curves II: A review of applications in water resources planning, Water Resour. Bull., 31, 1029-1039, 1995.

Wagener, T. and Wheater, H. S.: Parameter estimation and regionalization for continuous rainfall-runoff models including uncertainty, J. Hydrol., 320, 132-154, 2006.

Walter, M. T., Brooks, E. S., McCool, D. K., King, L. G., Molnau, M., and Boll, J.: Process-based snowmelt modeling: does it require more input data than temperature-index modeling?, J. Hydrol., 300, 65-75, doi: 10.1016/j.jhydrol.2004.05.002, 2005.

Westerberg, I. K. and McMillan, H. K.: Uncertainty in hydrological signatures, Hydrol. Earth Syst. Sci., 19, 3951-3968, doi: 10.5194/hess-19-3951-2015, 2015.

Westerberg, I. K., Gong, L., Beven, K. J., Seibert, J., Semedo, A., Xu, C.-Y., and Halldin, S.: Regional water balance modelling using flow-duration curves with observational uncertainties, Hydrol. Earth Syst. Sci., 18, 2993-3013, doi:10.5194/hess-18-2993-2014, 2014.

Westerberg, I. K., Guerrero, J.-L., Younger, P. M., Beven, K. J., Seibert, J., Halladin, S., Freer, J. E., and Xu, C.-Y.: Calibration of hydrological models using flow-duration curves, Hydrol. Earth Syst. Sci., 15, 2205-2227, doi:10.5194/hess-15-2205-2011, 2011.

Westerberg, I. K., Wagener, T., Coxon, G., McMillan, H. K., Castellarin, A., Montanari, A., and Freer, J.: Uncertainty in hydrological signatures for gauged and ungauged catchments, Water Resour. Res., 52, 1847-1865, doi:10.1002/2015WR017635., 2016

Winsemius, H. C., Schaefli, B., Montanari, A., and Savenije, H. H. G.: On the calibration of hydrological models in ungauged basins: A framework for integrating hard and soft hydrological information, Water Resour. Res., 45, W12422, doi:10.1029/2009wr007706, 2009.

Yadav, M., Wagener, T., and Gupta, H.: Regionalization of constraints on expected watershed response behavior for improved predictions in ungauged basins, Adv. Water Resour., 30, 1756-1774, 2007.

Yaeger, M., Coopersmith, E., Ye, S., Cheng, L., Viglione, A., and Sivapalan, M.: Exploring the physical controls of regional patterns of flow duration curves – Part 4: A synthesis of empirical analysis, process modeling and catchment classification, Hydrol. Earth Syst. Sci., 16, 4483-4498, doi: 10.5194/hess-16-4483-2012, 2012.

[revised manuscript text omitted]

[1]Ratio of potential ET to total precipitation, [2]Percentage of snowfall to total precipitation. Climatological features were calculated using spatial averages of the grid data, while the flow metrics were from the daily hydrographs for 2007-2015 as explained in Section 3.6.

**Table 2: Ranges of GR4J parameters used for parameter calibration (Demirel et al., 2013)**

| Parameter | Range |
|---|---|
| X1 (mm) | 10 to 2000 |
| X2 (mm) | -8 to +6 |
| X3 (mm) | 10 to 500 |
| X4 (days) | 0.5 to 4.0 |

**Table 3: Results of the paired t-tests for potential questions on rainfall-runoff modelling in ungauged catchments**

| Questions | Corresponding pair | [1]PM | [2]ΔPM | [3]std. err. | Answer |
|---|---|---|---|---|---|
| Q1. Did RFDC_cal outperform PROX_reg? | RFDC_cal – PROX_reg | NSE | -0.22 | 0.051 | No[*] |
| Q2. Did $D_{RI}$ improve FDC_cal? | FDC+DRL_cal – FDC_cal | NSE | 0.12 | 0.025 | Yes[*] |
| Q3. Did parameters from Q_cal contain flow timing information for ungauged catchments? | PROX_reg – FPROX_reg | NSE | 0.10 | 0.031 | Yes[*] |
| Q4. Did absence of flow timing affect model efficiency? | Q_cal – FDC_cal | NSE | 0.23 | 0.026 | Yes[*] |
| Q5. Did PROX_reg outperform RFDC_cal in predicting low flows? | PROX_reg – RFDC_cal | LNSE | 0.09 | 0.031 | Yes[*] |
| Q6. Did PROX_reg outperform RFDC_cal in reproducing $I_{BF}$? | PROX_reg – RFDC_cal | $D_{FS}(I_{BF})$ | 0.06 | 0.028 | No |
| Q7. Did errors in regional FDCs affect RFDC_cal significantly? | RFDC_cal – FDC_cal | NSE | -0.09 | 0.069 | No |

[1]Performance metric used for t-test, [2]Mean PM difference between the corresponding pair, [3]Standard error of ΔPM. [*]ΔPM is significantly different from zero. The significance was evaluated at 95% confidence levels.

[Figure]

**Figure 1: Locations of the study catchments in South Korea. The numbers are labelled at the outlet of each catchment.**

[Figure]

**Figure 2: The schematised structure of GR4J (X1-X4: model parameters, PE: potential evapotranspiration, P: precipitation, Q: runoff, other letters indicate variables conceptualising internal catchment processes).**

[Figure]

[Figure]

[Figure]

Figure 3: (a) box plots of high flow (NSE) and low flow (LNSE) reproducibility of the behavioural parameters obtained from the hydrograph calibration at the 45 catchments, (b) the relationship between the input-output consistency and the model performance. The straight lines in the box plots connect the performance metrics for the calibration (2011-2015) and the validation periods (2007-2010) in each catchment.

[Figure]

Figure 4: The relationships between model the behavioural parameters obtained from the hydrograph calibration at the 45 catchments, (b) the relationship between the input-output consistency and (a) high flow reproducibility (NSEs) and (b) low flow reproducibility (LNSEs)

[Figure]

Figure 4: 1:1 scatter plot between the empirical flow quantiles and the flow quantiles predicted by the top-kriging FDC regionalisation method.

[Figure]

[Figure]

the model

Figure 5: Observed and predicted hydrographs (continuous and dashed lines) with estimated uncertainties (shaded area) at three stations with best (top), intermediate (middle), and worst (bottom) predictive performance respectively. The plot inside of each hydrograph present agreement between observed and modelled FDCs in log-log space in which its horizontal and vertical axes are for exceedance probability (range of 0-1) and runoff (same range of each hydrograph) respectively. . The straight lines in the box plots connect the performance metrics for the calibration (2011-2015) and the validation periods (2007-2010) in each catchment.

[Figure]

**Figure 5: Box plots of NSE and LNSE values between the observed and the predicted hydrographs by RFDC_cal and PROX_reg for the 45 catchments under the cross validation mode.**

[Figure]

[Figure]

**Figure 6: The observed and predicted hydrographs, the prediction areas, and the observed and predicted FDCs given by (a) the hydrograph calibration and (b) the FDC calibration for the Catchment 2.**

[Figure]

[Figure]

Figure 7: The input-output consistency vs. equifinality increased by replacing the hydrograph calibration with the FDC calibration. The equifinality ratio is defined as the ratio between the prediction areas of the 50 behavioural parameters gained from the FDC calibration and the hydrograph calibration.

[Figure]

 **FDC and Hydrograph calibrations) and for ungauged catchments (RFDC_cal and PROX_reg), (b) boxplots of LSNEs (low flow reproducibility) gained from the same methods. The dashed lines distinguish between** regionalisation **method** .

[Figure]

Figure 8: Flow signature reproducibility comparison between RFDC_cal and PROX_reg in terms of $R_{QP}$ (a), $I_{BF}$ (b), and $D_{RL}$ (c).

[Figure]

[Figure]

Figure 9: Predictive performance of the FDC calibrations additionally conditioned by $R_{QP}$ (FDC+RQP), $I_{BF}$ (FDC+IBF), and $D_{RL}$ (FDC+DRL) in comparison to the other modelling approaches. Q_cal and FDC_cal refer to the hydrograph and the FDC calibration in gauged catchments respectively. 38 catchments with positive NSEs for all the modelling approaches were used in the box-plots.

[Figure]

Figure 10: (a) observed FDC and FDCs modelled by the 50 parameter sets from the FDC calibration, (b) sample observed hydrograph, and hydrograph modelled by the same 50 parameter sets, and (c) Box plots of observed baseflow and direct runoff. The whiskers indicate maximum and minimum values. All panels are for Namgang dam (catchment 2) with 0.86 and 0.51 NSEs of daily flows using the hydrograph calibration and the FDC calibration respectively.

[Figure]

Figure 11: 8: Flow signature reproducibility comparison between RFDC_cal and PROX_reg in terms of ROF (a), IBF (b), and DRL (c).

---

## Referee Report (RR1)

The authors have done an excellent job at improving the focus of the paper, the introduction and generally across the manuscript. I have a couple of suggestions for further improvements to the introduction that are at the discretion of the author, and some minor comments across the manuscript that will need to be addressed prior to publication.

**Optional suggestion 1:** I recommend using quotation marks the first time that you use the terms referring to the key techniques used in the paper.

> Line 25: (referred to as the "hydrograph calibration" hereafter)
>
> Line 47: (referred to as the "FDC calibration" hereafter)
>
> Line 62: (referred to as "RFDC_cal" hereafter)
>
> This makes it clear to the reader the general term that you will be using to describe these techniques throughout the rest of the manuscript. If they get lost in the methodology and results then they can easily find where you define these terms in the introduction.

**Optional suggestion 2:** Add a table/figure summarising (in columns) the key methods (e.g. "hydrograph calibration", "FDC calibration", "RFDC_cal") the key references for these methods, the key strengths, the key weaknesses and gaps in literature. This will give the reader a bit of a road-map of the introduction and make it easier to see the research gap that you are trying to address.

**Minor comments:**

1. Line 136: "and thus can worsen the equifinality" without providing evidence or a reference I suggest removing this statement.
2. Line 177: For completeness I suggest also mentioning the BATEA uncertainty estimation methodology of Kuczera et al (2006).
3. Line 154: I suggest being a bit more descriptive about your methods than using the term "Monte-Carlo random sampling" or to reference a particular methodology that you used for implementing this broad technique. This is important to ensure the reproducibility of the work.
4. Line 333: "Catchment 2" is a bit vague now that you have removed the previous version of the manuscript "Table 1" that listed the catchments. I recommend using the catchment name here i.e. "Namgang Dam" and also in the Figure 6 caption, to again ensure the reproducibility of the work.
5. Line 374: I recommend referring to Table 3 much earlier in this section (at the beginning if possible) so that the reader knows what you are referring to.

**A few typos:**

1. Line 75: "reginal" to "regional"
2. Line 94: (Optional) My preference is to not start a section/paragraph with a numerical number.
3. Line 271: "rainfalls" to "rainfall" to be consistent with earlier in the sentence.
4. Line 461: "combining" to "combine"

---

## Author Response (AR2)

Dear Dr. Fabrizio Fenicia

We greatly appreciate the positive feedbacks from the reviewers and your valuable editing efforts. In the second round revision, we incorporated most of the referees' comments.

We believe that all comments from you and the referees were very constructive to improve our manuscript. Specific information on how each comment was incorporated into the revised manuscript is given below. Once again, we thank for all of your editing efforts.

Sincerely,

Jong Ahn Chun
Corresponding author

Response to comments from reviewer 1:

A. Lines 38-41: one approach to using FDC signatures in model calibration is direct use of multiple signatures (e.g., Yilmaz et al., 2008; and Shafii and Tolson, 2015), which is missing in the list of practices in this section.
→ *We shortly mentioned this approach in line 37, as "distinctive flow signatures can be used in lieu of hydrograph." And, we cited the given references at the sentence.*

B. Regarding hypothesis testing results: when the null hypothesis cannot be rejected, it does not mean that it is accepted. Rather, we should say there is no evidence that it is rejected. Therefore, in Table 3, the word 'No' has to be replaced with 'No Evidence'. The corresponding text needs to change too, which will be a slight modification.
→ *We agree to this comment and used "Unlikely" for Q6 and Q7 in Table 3. For Q1, since we found a statistical significance that PROX_reg outperformed RFDC_cal, we used the term "No" to differentiate from "Unlikely"*

C. It is not clear how additional constraints (i.e., signatures) are utilized. For example, looking at Figure 9, how is IBF used in FDC+IBF case? Has there been a threshold for accepting/rejecting models? Some explanation is needed in the paper.
→ *We explained this in line 342. We chose the parameter set providing the minimum biases for each flow signature from the collection of 50 parameters gained from the FDC calibration. This simple selection could bring the improved predictability.*

D. In the limitation section, I think authors should mention that all catchments have a long record of data, and conclusions may not be expandable to poorly gauged catchments. I am wondering if hydrograph is still more appropriate than FDC if the number of available data points is not high.
→ *We addressed this issue in line 431. We agree that a same comparison between PROX_reg and RFDC_cal may bring different conclusions in a poor river gauging network.*

Response to comments from reviewer 2:

Optional suggestion 1: I recommend using quotation marks the first time that you use the terms referring to the key techniques used in the paper.
Line 25: (referred to as the "hydrograph calibration" hereafter)
Line 47: (referred to as the "FDC calibration" hereafter)
Line 62: (referred to as "RFDC_cal" hereafter)
This makes it clear to the reader the general term that you will be using to describe these techniques throughout the rest of the manuscript. If they get lost in the methodology and results then they can easily find where you define these terms in the introduction.

→ *We agree. We used quotations marks for the terms and for "PROX_reg" as requested. They are in lines 25, 47, 62, and 220 in the revised manuscript, respectively.*

Optional suggestion 2: Add a table/figure summarising (in columns) the key methods (e.g."hydrograph calibration", "FDC calibration", "RFDC_cal",) the key references for these methods, the key strengths, the key weaknesses and gaps in literature. This will give the reader a bit of a road-map of the introduction and make it easier to see the research gap that you are trying to address.

→ *We agree that adding a table with reviews on the modelling approaches is a good way to promote readers' awareness of research gaps. However, the hydrograph calibration is a general approach applied in many rainfall-runoff studies. And, FDCs or metrics from FDCs have been frequently used for parameter identification. Condensing many studies into a table with key information would require a long time. Since we believe that the literatures introducing the objective of this study seem to be sufficient, we want to consider this comment for our further study on rainfall-runoff modelling with regional flow signatures.*

Minor comments:
1. Line 136: "and thus can worsen the equifinality" without providing evidence or a reference I suggest removing this statement.
→ *We removed the statement as recommended (line 136).*

2. Line 177: For completeness I suggest also mentioning the BATEA uncertainty estimation methodology of Kuczera et al (2006).
→ *We cited the reference suggested (line 176).*

3. Line 154: I suggest being a bit more descriptive about your methods than using the term "Monte-Carlo random sampling" or to reference a particular methodology that you used for implementing this broad technique. This is important to ensure the reproducibility of the work.
→ *In section 3.3, we introduced the objective function first (line 153), and described our calibration method (line 162-167).*

4. Line 333: "Catchment 2" is a bit vague now that you have removed the previous version of the manuscript "Table 1" that listed the catchments. I recommend using the catchment name here i.e. "Namgang Dam" and also in the Figure 6 caption, to again ensure the reproducibility of the work.
→ *We corrected as recommended (line 333).*

5. Line 374: I recommend referring to Table 3 much earlier in this section (at the beginning if possible) so that the reader knows what you are referring to

→ *In line 367 (the second sentence of the paragraph), we referred to Table 3 for better readability.*

A few typos:

1. Line 75: "reginal" to "regional"

→ *We corrected it (line 75).*

2. Line 94: (Optional) My preference is to not start a section/paragraph with a numerical number.

→ *Now we begin with "For this study, we selected 45 catchments…" (line 94).*

3. Line 271: "rainfalls" to "rainfall" to be consistent with earlier in the sentence.

→ *We corrected it (line 271).*

4. Line 461: "combining" to "combine"

→ *We corrected it (line 463).*

**A comparative assessment of rainfall-runoff modelling against regional flow duration curves for ungauged catchments**

Daeha Kim[1], Ilwon Jung[2], Jong Ahn Chun[1]

[1]APEC Climate Center, Busan, 48058, South Korea
[2]Korea Infrastructure Safety & Technology Corporation, Jinju, Gyeongsangnam-do, 52852, South Korea

*Correspondence to*: Jong Ahn Chun (jachun @apcc21.org)

**Abstract.** Rainfall-runoff modelling has long been a special subject in hydrological sciences, but identifying behavioural parameters in ungauged catchments is still challenging. In this study, we comparatively evaluated performance of the local calibration of a rainfall-runoff model against regional flow duration curves (FDC), which is a seemingly alternative method of classical parameter regionalisation for ungauged catchments. We used a parsimonious rainfall-runoff model over 45 Korean catchments under semi-humid climate. The calibration against regional FDCs was compared with the simple proximity-based parameter regionalisation. Results show that transferring behavioural parameters from gauged to ungauged catchments significantly outperformed the local calibration against regional FDCs due to the absence of flow timing information in the regional FDCs. The behavioural parameters gained from observed hydrographs were likely to contain intangible flow timing information affecting predictability in ungauged catchments. Additional constraining with the rising limb density appreciably improved the FDC calibrations, implying that flow signatures in temporal dimensions would supplement the FDCs. As an alternative approach in data-rich regions, we suggest calibrating a rainfall-runoff model against regionalised hydrographs to preserve flow timing information. We also suggest use of flow signatures that can supplement hydrographs for calibrating rainfall-runoff models in gauged and ungauged catchments.

**1 Introduction**

A standard method to predict daily streamflow is to employ a rainfall-runoff model that conceptualises catchment functional behaviours, and simulate synthetic hydrographs from atmospheric drivers (Wagener and Wheater, 2006; Blöschl et al., 2013). A prerequisite of this conceptual modelling approach is parameter identification to enable the rainfall-runoff model to imitate actual catchment behaviours. Conventionally, behavioural parameters are estimated via model calibration against observed hydrographs (referred to as the "hydrograph calibration" hereafter). The hydrograph calibration provides convenience to attain reproducibility of the predictand (i.e., streamflow time series), which is commonly used as a performance measure in rainfall-runoff modelling studies. Because the degree of belief in hydrological models is normally measured by how they can

reproduce observations (Westerberg et al., 2011), use of the hydrograph calibration has a long tradition in runoff modelling (Hrachowicz et al., 2013).

The hydrograph calibration, however, can be challenged by epistemic errors in input and output data, sensitivity to calibration criteria, and inability under no or poor data availability (Westerberg et al, 2011; Zhang et al., 2008). Importantly, it is difficult to know whether the parameters optimised toward maximising hydrograph reproducibility are unique to represent actual catchment behaviours, since multiple parameter sets possibly show similar predictive performance (Beven, 2006, 1993). This low uniqueness of the optimal parameter set, namely the equifinality problem in conceptual hydrological modelling, can become a significant uncertainty source particularly when extrapolating the optimal parameters to ungauged catchments (Oudin et al., 2008).

To overcome or circumvent those disadvantages, distinctive flow signatures (i.e., metrics or auxiliary data representing catchment behaviours) in lieu of observed hydrographs can be used to identify model parameters (e.g., Yilmaz et al., 2008; Shafii and Tolson, 2015). The flow duration curve (FDC) has received particular attention in the signature-based model calibrations as a single criterion (e.g., Westerberg et al., 2014, 2011; Yu and Yang, 2000; Sugawara, 1979) or one of calibration constraints (e.g., Pfannerstill et al., 2014; Kavetski et al., 2011; Hingray et al., 2010; Blazkova and Beven, 2009; Yadav et al., 2007). The FDC, the relationship between flow magnitude and its frequency, provides a summary of temporal streamflow variations in a probabilistic domain (Vogel and Fennessey, 1994). Many FDC-related studies have found that climatological and geophysical characteristics within a catchment determine the shape of the FDC (e.g., Cheng et al., 2012; Ye et al., 2012; Yokoo and Sivaplan, 2011; Bottor et al., 2007). With only few physical parameters, the shape of the period-of-record FDC could be analytically expressed (Botter et al., 2008). Based on this strong relationship between catchment physical properties and the FDC, one may hypothesise that model calibration against the FDC (referred to as the "FDC calibration" hereafter) can provide parameters that can sufficiently capture actual catchment behaviours. Sugawara (1979) is the first attempt at the FDC calibration, emphasising its advantage to reduce negative effects of epistemic errors in rainfall-runoff data. Westerberg et al. (2011) also highlighted that the FDC calibration may provide robust predictions to moderate disinformation such as the presence of event flows under inconsistency between inputs and outputs.

If it allows rainfall-runoff models to sufficiently capture functional behaviours of catchments, the FDC calibration would have an especial value in comparison to the parameter regionalisation for prediction in ungauged catchment. The parameter regionalisation, which transfers or extrapolates behavioural parameters from gauged to ungauged catchments (e.g., Kim and Kaluarachchi, 2008; Oudin et al., 2008; Parajka et al., 2007; Wagener and Wheater, 2006; Dunn and Lilly, 2011), conveniently provides a priori estimates of behavioural parameters and thus became a popular approach to parameter identification in ungauged catchments (see a comprehensive review in Parajka et al., 2013). However, it has a critical concern that regionalised parameters are highly dependent on model calibrations at gauged sites that may have substantial equifinality problems. Under no flow information in ungauged catchments, it is impossible to know whether regionalised parameters are behavioural. Thus, regionalised parameters might be insufficiently reliable and highly uncertain (Bárdossy, 2007; Oudin et al., 2008; Zhang et al., 2008).

On the other hand, the calibration against regional FDCs (referred to as "RFDC_cal" hereafter) may reduce the primary concern in the classical parameter regionalisation scheme. The regional models predicting FDCs at ungauged sites have showed strong performance, for instance, via regression analyses between quantile flows and catchment properties (e.g., Shu and Ouarda, 2012; Mohammoud, 2008; Smakhtin et al., 1997), geostatistical interpolation of quantile flows (e.g., Pugliese et al., 2014; Westerberg et al., 2014), and regionalisation of theoretical probability distributions (e.g., Atieh et al., 2017; Sadegh et al., 2016) among many variations. The parameters obtained from RFDC_cal are deemed behavioural, because a distinctive flow signature of the target ungauged catchment directly identifies them; however, predicted FDCs should be reliable in this case. A FDC is a compact representation of runoff variability at all time scales from inter-annual to event-scale, embedding various aspects of multiple flow signatures (Blöschl et al., 2013). Based on this strength, several studies already showed promising predictive performance using RFDC_cal for ungauged catchments (e.g., Westerberg et al., 2014; Yu and Yang, 2000).

Nevertheless, practical questions arise when using RFDC_cal for ungauged catchments. First, the FDC is simplified information with flow magnitudes only; hence, the FDC calibration could worsen the equifinality problem relative to the hydrograph calibration. Due to no flow timing information in regional FDCs, one may cast a concern that parameters obtained from RFDC_cal may provide poorer predictive performance than regionalised parameters gained from the hydrograph calibration. Indeed, there is additional uncertainty in predicted FDCs possibly introduced by the regionalisation models (Westerberg et al., 2011; Yu et al., 2002). RFDC_cal may be undesirable when a simple parameter regionalisation can provide better performance, because regionalising observed FDCs may require expensive efforts. Several comparative studies on parameter regionalisation (e.g., Parajka et al., 2013; Oudin et al., 2008) suggested that the simple proximity-based parameter transfer can be competitive in many regions. Second, there may be additional flow signatures to improve predictive performance of the FDC calibration. Additional constraining can lead to better predictive performance of the RFDC (Westerberg et al., 2014); however, it is still an open question which flow signatures can supplement the FDC calibration.

As discussed, RFDC_cal seems promising for prediction in ungauged catchments. However, to our knowledge, RFDC_cal has never been evaluated in a comparative manner with classical parameter regionalisation except Zhang et al. (2015), which assessed its performance in part. Therefore, this study aimed to evaluate predictive performance of RFDC_cal in comparison to a conventional parameter regionalisation. We focused on the absence of flow timing in the FDC and its impacts on rainfall-runoff modelling. In this work, a parsimonious 4-parameter conceptual model was used to simulate daily hydrographs for 45 catchments in South Korea. To predict FDCs in ungauged catchments, a geostatistical regional model was adopted here. The Monte-Carlo sampling was used to identify model parameters and measure equifinality in the hydrograph and the FDC calibrations.

**2 Description of the study area and data**

For this study, we selected 45 catchments located across South Korea with no or negligible human-made influences on flow variations were selected for this study (Figure 1). South Korea is characterised as a temperate and semi-humid climate with rainy summer seasons. The North Pacific high-pressure brings monsoon rainfall with high temperatures during summer seasons, while dry and cold weathers prevail in winter seasons due to the Siberian high-pressure. Typical ranges of annual precipitation are 1200-1500 and 1000-1800 mm in the northern and the southern areas respectively (Rhee and Cho, 2016). Annual mean temperatures in South Korea range between 10 and 15 °C (Korea Meteorological Administration, 2011). Approximately, 60-70 percent of precipitation falls in summer seasons from June to September (Bae et al., 2008). Streamflow usually peaks in the middle of summer seasons because of heavy rainfall or typhoons, and hence information of catchment behaviours is largely concentrated on summer-season hydrographs. Snow accumulation and ablation occurring at high elevations have minor influences on flow variations due to relatively small amount of winter precipitation (Bae et al., 2008).

The study catchments were selected based on availability of streamflow data. High-quality daily streamflow data across South Korea have been produced since establishment of the Hydrological Survey Centre in 2007 (Jung et al., 2010), though river stages have been monitored for an extensive length at a few gauging stations. Thus, we collected streamflow data at 29 river gauging stations from 2007 to 2015 together with inflow data of 16 multi-purpose dams for the same data period from the Water Resources Management Information System operated by the Ministry of Land, Infrastructure, and Transport of the Korean government (available at http://www.wamis.go.kr/). The mean annual flow of the study catchments was 739 mm yr$^{-1}$ with a standard deviation of 185 mm yr$^{-1}$ during 2007-2015.

In addition, as atmospheric forcing inputs, we collected daily precipitation and maximum and minimum temperatures for 2005-2015 at 3-km grid resolution produced by spatial interpolations between 60 stations of the automated surface observing system (ASOS) maintained by the Korea Meteorological Administration. The ASOS data were interpolated by the Parameter-elevation Regression on Independent Slope Model (PRISM; Daly et al., 2008), and overestimated pixels of the PRISM grid data were smoothed by the inverse distance method. Jung and Eum (2015) found that this combined method improved the spatial interpolation of precipitation and the temperatures in South Korea. The annual mean precipitation and temperature of the study catchments vary within ranges of 1145–1997 mm yr$^{-1}$ and 8.0–13.8 °C during 2007-2015. Hydro-climatological features of the 45 catchments are summarised in Table 1.

**3 Methodology**

**3.1 Hydrological model (GR4J)**

A parsimonious rainfall-runoff model, GR4J (Perrin et al., 2003), was adopted to simulate daily hydrographs of the 45 catchments for 2007-2015. GR4J conceptualises functional catchment response to rainfall with four free parameters that

125 regulate the water balance and water transfer functions. Figure 2 schematises the structure of GR4J. The four parameters (X1 to X4) conceptualises soil water storage, groundwater exchange, routing storage, and the base time of unit hydrograph respectively. Since its parsimonious and efficient structure allows robust calibration and reliable regionalisation of the parameters, GR4J has been frequently used for modelling daily hydrographs with various purposes under diverse climatic conditions (Zhang et al., 2015). The computation details and discussion are found in Perrin et al. (2003). The potential

130 evapotranspiration (PE in Figure 2) was estimated by the temperature-based model proposed by Oudin et al. (2005) for lumped rainfall-runoff modelling.

**3.2 Preliminary data processing**

Before rainfall-runoff modelling, we preliminarily processed the grid climatic data to convert precipitation data to liquid water forcing (i.e., rainfall and snowmelt depths) using a physics-based snowmelt model proposed by Walter et al. (2005).

135 The preliminary snowmelt modelling was mainly for reducing systematic errors from no snow component in GR4J, which may affect model performance in catchments at relatively high elevations. We chose this preliminary processing to avoid adding more parameters (e.g., the temperature index) to the existing structure of GR4J. In the case of GR4J, one additional parameter implies 25% complexity increase in terms of the number of parameters, and thus can worsen the equifinality. The snowmelt model uses the same inputs of GR4J to simulate point-scale snow accumulation and ablation processes (i.e., no

140 additional inputs are required). The snowmelt model is a physics-based model but uses empirical methods to estimate its parameters for the energy balance simulation. As outputs, it produces the liquid water depths and the snow water equivalent. For lumped inputs to GR4J, we took spatially averaged pixel values of the liquid water depths and the maximum and minimum temperatures within the boundary of each catchment.

After the snowmelt modelling, consistency between the liquid water depths and the observed flows (i.e., input-output

145 consistency) was checked using the current precipitation index (CPI; Smakhtin and Masse, 2000) defined as:

$$I_t = I_{t-1} \cdot K + R_t \tag{1}$$

where $I_t$ is the CPI (mm) at day t, K is a decay coefficient (0.85 $d^{-1}$), and $R_t$ is the liquid water depth (mm $d^{-1}$) at day t. CPI mimics temporal variations of typical streamflow data by converting intermittent precipitation data to a continuous time series with an assumption of the linear reservoir. The input-output consistency can be evaluated using correlation between

150 CPI and observed streamflow as in Westerberg et al. (2014) and Kim and Kaluarachchi (2014). The Pearson correlation coefficients between CPI and streamflow data of the 45 catchments had an average of 0.67 with a range of 0.43-0.79, and no outliers were found in the box plot of the correlation coefficients. Hence, we assumed that consistency between climatic forcing and observed hydrographs was acceptable.

**3.3 The hydrograph calibration in gauged catchments**

To search behavioural parameter sets of GR4J against the streamflow observations (i.e., the hydrograph calibration), we used the objective function in Zhang et al. (2015) as the calibration criterion to consider the Nash-Sutcliffe Efficiency (NSE) and the Water Balance Error together:

$$\text{OBJ} = (1 - \text{NSE}) + 5|\ln(1 + \text{WBE})|^{2.5} \tag{2a}$$

$$\text{NSE} = 1 - \frac{\sum_{i=1}^{N}(Q_{obs,i} - Q_{sim,i})^2}{\sum_{i=1}^{N}(Q_{obs,i} - \overline{Q_{obs}})^2} \tag{2b}$$

$$\text{WBE} = \frac{\sum_{i=1}^{N}(Q_{obs,i} - Q_{sim,i})}{\sum_{i=1}^{N} Q_{obs,i}} \tag{2c}$$

where $Q_{obs}$ and $Q_{sim}$ are the observed and simulated flows respectively, $\overline{Q_{obs}}$ is the arithmetic mean of $Q_{obs}$, and N is the total number of flow observations. The best parameter sets for each study catchment was obtained from minimisation of the OBJ using the Monte-Carlo simulations described below.

To determine sufficient runs for the random simulations, we calibrated GR4J parameters using the shuffled complex evolution (SCE) algorithm (Duan et al., 1992) for one catchment with moderate input-output consistency with the parameter range given by Demirel et al. (2013). Then, the total number of random simulations was iteratively determined by adjusting the number of runs until the minimum OBJ of the random simulations became adequately close to the OBJ value from the SCE algorithm. We found that approximately 20,000 runs could provide the minimum OBJ value equivalent to that from the SCE algorithm. Subsequently, GR4J was calibrated by 20,000 runs of the Monte-Carlo simulations for all 45 catchments, and the parameter sets with the minimum OBJ values were taken for runoff predictions. In addition, we sorted the 20,000 parameter sets in terms of corresponding OBJ values in ascending order, and first 50 sets (0.25% of the total samples) were taken to measure the degree of equifinality. We measured the equifinality simply by the prediction area between 2.5% and 97.5% boundaries of runoff simulations given by the collected 50 parameter sets. This prediction area was later compared to that from the FDC calibration under the same Monte-Carlo framework. Note that we estimated the prediction area to comparatively evaluate the degree of equifinality between the hydrograph and the FDC calibrations under the same sampling size and the same acceptance rate for all the catchments. For more sophisticated and reliable uncertainty estimation, other methods are available such as the Generalised Likelihood Uncertainty Estimation (GLUE; Beven and Bingley, 1992), the Bayesian Total Error Analysis (BATEA; Kavetski et al., 2006), and the Differential Evolution Adaptive Metropolis (DREAM; Vrugt and Ter Braak, 2011).

For the hydrograph calibration, the 9-year streamflow data were divided into two parts for calibration (2011-2015) and for validity check (2007-2010), respectively. A two-year warm-up period was used for initialising all runoff simulations in this study.

**3.4 Model calibration against the regional FDC for ungauged catchments**

Each catchment was treated ungauged for the comparative evaluation of RFDC_cal in the leave-one-out cross-validation (LOOCV) mode. For regionalising empirical FDCs, the geostatistical method recently proposed by Pugliese et al. (2014) was used. Pugliese et al. (2014) employed the top-kriging method (Skøien et al., 2006) to spatially interpolate the total negative deviation (TND), which is defined as the area between the mean annual flow and below-average flows in a normalised FDC. The top-kriging weights that interpolate TND values were taken as weights to estimate flow quantiles of ungauged catchments from empirical FDCs of surrounding gauged catchments. The FDC of an ungauged catchment in Pugliese et al. (2014) is estimated from normalised FDCs of surrounding gauged catchments as:

$$\widehat{\Phi}(w_0, p) = \widehat{\phi}(w_0, p) \cdot \overline{Q}(w_0) \tag{3a}$$

$$\widehat{\phi}(w_0, p) = \sum_{i=1}^{n} \lambda_i \cdot \phi_i(w_i, p), \quad p\epsilon(0,1) \tag{3b}$$

where $\widehat{\Phi}(w_0, p)$ is the estimated quantile flow (m$^3$ s$^{-1}$) at an exceedance probability p (unitless) for an ungauged catchment $w_0$, $\widehat{\phi}(w_0, p)$ is the estimated normalised quantile flow (unitless), $\overline{Q}(w_0)$ is the annual mean streamflow (m$^3$ s$^{-1}$) of the ungauged catchment, and $\phi_i(w_i, p)$ and $\lambda_i$ are normalised quantile flows (unitless) and corresponding top-kriging weights (unitless) of gauged catchment $w_i$, respectively. The unknown mean annual flow of an ungauged catchment, $\overline{Q}(w_0)$, can be estimated with a rescaled mean annual precipitation defined as:

$$MAP^* = 3.171 \times 10^{-5} \cdot MAP \cdot A \tag{4}$$

where MAP* is the rescaled mean annual precipitation (m$^3$ s$^{-1}$), MAP is mean annual precipitation (mm yr$^{-1}$) and A is the area (km$^2$) of the ungauged catchment, and the constant 3.171×10$^{-5}$ converts the unit of MAP$^*$ from mm yr$^{-1}$ km$^2$ to m$^3$ s$^{-1}$.

A distinct advantage of the geostatistical method is its ability to estimate the entire flow quantiles in a FDC with a single set of top-kriging weights. Since a parametric regional FDC (e.g., Yu et al., 2002; Mohamoud, 2008) is obtained from independent models for each flow quantile in many cases, for instance, by multiple regressions between selected quantile flows and catchment properties, fundamental characteristics in a FDC continuum would be entirely or partly lost. The geostatistical method, on the other hand, treats all flow quantiles as a single object; thereby, features in a FDC continuum can be preserved. It showed promising performance to reproduce empirical FDCs only using topological proximity between catchments. More details on the geostatistical method are found in Pugliese et al. (2014).

For regionalising empirical FDCs of the 45 catchments, we followed the same procedure of Pugliese et al. (2014). We obtained top-kriging weights ($\lambda_i$) by the geostatistical interpolation of TND values from observed FDCs for the calibration period (2011-2015). Then, the top-kriging weights were used to interpolate empirical flow quantiles. The number of

neighbours for the TND interpolation was iteratively determined as five at which additional neighbouring TNDs are unlikely to bring better agreement between the estimated and observed TNDs. In other words, normalised flow quantiles of five catchments surrounding the target ungauged catchment were interpolated with the top-kriging weights. Then, MAP$^*$ of the target ungauged catchment was multiplied. We predicted flow quantiles at 103 exceedance probabilities (p of 0.001, 0.005, 99 points between 0.01 and 0.99 at an interval of 0.01, 0.995, and 0.999) for rainfall-runoff modelling against regional FDCs (i.e., RFDC_cal).

For runoff prediction in ungauged catchments, the GR4J parameters were identified by the same Monte-Carlo sampling but toward minimisation of OBJ value between the regional and the modelled flow quantiles at the 103 exceedance probabilities. The best parameter set, which provided the minimum OBJ value, was taken as the best behavioural set of RFDC_cal for each catchment.

**3.5 Proximity-based parameter regionalisation for ungauged catchments**

We selected the proximity-based parameter transfer (referred to as "PROX_reg" hereafter) to comparatively evaluate predictive performance of RFDC_cal. The parameter regionalisation has three classical categories: (a) proximity-based parameter transfer (i.e., PROX_reg; e.g., Oudin et al., 2008); (b) similarity-based parameter transfer (e.g., McIntyre et al., 2005); and (c) regression between parameters and physical properties of gauged catchments (e.g., Kim and Kaluarachchi, 2008). A comprehensive review on the parameter regionalisation in Parajka et al. (2013) reported that PROX_reg has competitive performance under humid climate with low-complexity models relative to the other categories. Based on modelling conditions in this study (semi-humid climate and 4 parameters), we chose PROX_reg to evaluate RFDC_cal.

To predict runoff at the 45 catchments in the LOOCV mode, we transferred the behavioural parameter sets obtained from the hydrograph calibration of the five donor catchments used for the FDC regionalisation. In other words, we used the same donor catchments for FDC regionalisation and PROX_reg. This allows us to have consistency in transferring hydrological information from gauged to ungauged catchments between RFDC_cal and PROX_reg. Using the best behavioural parameter sets of the five donor catchments, we generated five runoff time series and took the arithmetic averages of them to represent runoff predictions by PROX_reg.

**3.6 Performance evaluation**

We used multiple performance metrics to evaluate predictive performance of all modelling approaches applied in this study. Predictive performance of each modelling approach was graphically evaluated using box plots of the performance metrics of the 45 catchments. In addition, we performed several paired t-tests to check the statistical significance of performance differences between the modelling approaches. Following is the description of the performance metrics.

To measure high- and low-flow reproducibility, we chose two traditional performance metrics, (1) the NSE between observed and predicted flows (Eq. 2b) and (2) the NSE of log-transformed flows (LNSE) respectively. LNSE is calculated as:

$$\text{LNSE} = 1 - \frac{\sum_{i=1}^{N}\left(\ln(Q_{obs,i}) - \ln(Q_{sim,i})\right)^2}{\sum_{i=1}^{N}\left(\ln(Q_{obs,i}) - \overline{\ln(Q_{obs})}\right)^2} \tag{5}$$

Though NSE and LNSE are frequently used for performance evaluation, they may be sensitive to errors in flow observations (Westerberg et al., 2011). Hence, we additionally selected three typical flow metrics that embed dynamic flow variation in a compact manner; the runoff ratio ($R_{QP}$), the baseflow index ($I_{BF}$), and the rising limb density ($D_{RL}$). $R_{QP}$, $I_{BF}$, and $D_{RL}$ are proxies of aridity and water holding capacity, contribution of the baseflow to flow variations, and flashness of catchment behaviours, respectively. They are defined as the ratio of runoff to precipitation, the ratio of baseflow to total runoff, and the inverse of average time to peak ($d^{-1}$) as:

$$R_{QP} = \frac{\overline{Q}}{\overline{P}} \tag{6a}$$

$$I_{BF} = \sum_{t=1}^{T} \frac{Q_{B,t}}{Q_t} \tag{6b}$$

$$D_{RL} = \frac{N_{RL}}{T_R} \tag{6c}$$

where $\overline{Q}$ and $\overline{P}$ are average flow and precipitation for a given period (mm $d^{-1}$), $Q_t$ and $Q_{B,t}$ (m $d^{-1}$) is the streamflow and the base flow at time t respectively, $N_{RL}$ is the number of rising limb, and $T_R$ is the total amount of time when the hydrograph is rising (days). $Q_{B,t}$ can be calculated by subtracting direct flow $Q_{D,t}$ from $Q_t$ as:

$$Q_{D,t} = c \cdot Q_{D,t-1} + 0.5 \cdot (1 + c) \cdot (Q_t - Q_{t-1}) \tag{7a}$$

$$Q_{B,t} = Q_t - Q_{D,t} \tag{7b}$$

where c is the filter parameter, which was set to 0.925 (Brooks et al., 2011; Eckhardt, 2007).

Flow signature reproducibility of RFDC_cal and PROX_reg were evaluated by the relative absolute bias between modelled and observed signatures as:

$$D_{FS} = \frac{|FS_{sim} - FS_{obs}|}{FS_{obs}} \tag{8}$$

where $D_{FS}$ is the relative absolute bias, $FS_{sim}$ is a flow signature of the modelled flows, and $FS_{obs}$ is that of the observed flows.

**4 Results**

**4.1 Hydrograph calibration and FDC regionalisation in gauged catchments**

Figure 3a displays results of the parameter identification against the observed hydrographs (i.e., the hydrograph calibration). The 45 catchments had the mean NSE and LNSE of 0.66 and 0.65 between the simulated and observed flows for the calibration period, respectively. The average NSE reduction from the calibration to the validation periods was 0.06 with a standard deviation of 0.10. The temporal transfer of the calibrated parameters did not decrease the mean LNSE value, while

a wider LNSE range indicates that uncertainty of low-flow predictions may increase when temporally transferring the calibrated parameters.

The predictive performance was closely related to the input-output consistency (Figure 3b), which was measured by the Pearson correlation coefficient between the CPI and the observed flows. A low input-output consistency implies that the rainfall-runoff data may include significant epistemic errors such as minimal flow responses to heavy rainfall or excessive response to tiny rainfalls. If the model calibration compensates disinformation from such errors, the parameters would be forced to have biases. Figure 3b shows that consistency in input-output data is a critical factor affecting parameter identification and thus performance. Perhaps, screening catchments with low input-output consistency may provide better predictions in ungauged catchments. However, we did not consider it in the LOOCV for RFDC_cal and PROX_reg, since variation in input-output consistency would be a common situation. Rather, reducing the number of gauged catchments lowers spatial proximity and thus can cause biases for ungauged catchments too. Overall, 27 catchments and 33 catchments showed NSE and LNSE values greater than 0.6. We assumed the hydrograph calibration under the Monte-Carlo framework, which was assisted by the SCE optimisation, was able to acceptably identify the behavioural parameters under given data quality.

Besides, Figure 4 illustrates the 1:1 scatter plot between the observed and predicted flow quantiles of all the catchments, indicating high applicability of the top-kriging FDC regionalisation. The overall NSE and LNSE values between the observed and regionalised flow quantiles show good applicability of the geostatistical method. The NSE and LNSE values for individual catchments have averages of 0.83 and 0.91 with standard deviations of 0.25 and 0.11, respectively, implying that low-flow predictions were slightly better. The performance of the geostatistical method was relatively poor at locations where gauging density is low. Catchments 4, 10, 35, and 36, which recorded 0.6 or less NSEs are limitedly hatched with or adjacent to the other catchments; nonetheless, LNSEs of those catchments were still greater than 0.7. This result is consistent with a finding of Pugliese et al. (2016) that performance of the geostatistical method was sensitive to river gauging density. Transferring flow quantiles from remote catchments may not sufficiently capture functional similarity between donor and receiver catchments. In spite of the minor shortcomings, the geostatistical FDC regionalisation was deemed acceptable based on the high NSE and LNSE of flow quantiles. Topological proximity was generally a good predictor of flow quantiles for the study catchments.

**4.2 Comparing hydrograph predictability between RFDC_cal and PROX_reg**

Figure 5 compares the box plots of NSE and LNSE values between RFDC_cal and PROX_reg. PROX_reg generally outperforms RFDC_cal in predicting both high and low flows, suggesting that transferring parameters identified by observed hydrographs would be a better choice than a local calibration against predicted FDCs. The differences between NSE values of PROX_reg and RFDC_cal have an average of 0.22 with a standard deviation of 0.34. Only 8 catchments showed higher NSEs with RFDC_cal. These higher NSE values of PROX_reg imply that PROX_reg is preferable when high-flow predictability is needed such as flood analyses. In the case of LNSE, PROX_reg still had a higher median than RFDC_cal

(0.53 and 0.62 for RFDC_cal and PROX_reg respectively). In 25 catchments, PROX_reg provided LNSE values greater than those of RFDC_cal.

The low performance of RFDC_cal was also found in the comparative assessment of Zhang et al. (2015), which evaluated RFDC_cal for 228 Australian catchments using the same GR4J model. Zhang et al. (2015) found that RFDC_cal was inferior to PROX_reg in the Australian catchments, because the FDC calibration poorly reproduced temporal flow variations relative to the hydrograph calibration. This study confirms the difficulty to capture dynamic catchment behaviours with FDCs containing no flow timing information.

A major weakness of RFDC_cal is the absence of flow timing information in the parameter calibration process. Unlike RFDC_cal, PROX_reg did not discard the flow timing information. The regionalised parameters may be able to implicitly transfer the flow timing information from gauged to ungauged catchments (this hypothesis will be discussed later in Section 4.4). Figure 6 illustrates how the absence of flow timing negatively influences on predictive performance. For this comparison, the parameters were recalibrated against the observed FDCs (not regional FDCs) under the same Monte Carlo method to discard errors introduced by the FDC regionalisation (i.e., equivalent to calibrations against perfectly regionalised FDCs). The parameters identified by the observed hydrograph (Figure 6a) brought a good predictability in both high and low flows, resulting in an excellent performance to reproduce the FDC. On the other hand, an excellent FDC reproducibility does not guarantee a good predictability in high flows (Figure 6b). This indicates that reproducing FDCs with rainfall-runoff models would be less sufficient than the hydrograph calibration to capture functional catchment responses.

In addition, Figure 6 shows that the prediction area of the 50 behavioural parameters from the Monte-Carlo simulations (indicated by the grey areas and the blue arrows) became much larger when using the FDC calibration instead of the hydrograph calibration. We calculated the ratio of the prediction area of the FDC calibration to that of the hydrograph calibration, and defined it as the equifinality ratio. It quantifies the degree of equifinality augmented by replacing the hydrograph calibration with the FDC calibration. Figure 7 displays the scatter plot between the equifinality ratio and the input-output consistency. The equifinality augmented by the loss of flow timing is likely to increase as the input-output consistency decreases. The average of the equifinality ratios was 1.96, implying that potential equifinality inherent in RFDC_cal could be substantial. This may suggest that the equifinality problem embedded in RFDC_cal could be more significant than that in PROX_reg.

**4.3 Comparing flow-signature predictability between RFDC_cal and PROX_reg**

Figure 8 summarises performance of RFDC_cal and PROX_reg to regenerate three flow signatures of $R_{QP}$, $I_{BF}$, and $D_{RL}$. RFDC_cal is competitive in reproducing the averaged-based signatures $R_{QP}$ and $I_{BF}$, while it showed relatively a weak ability to regenerate the event-based signature $D_{RL}$. $R_{QP}$ and $I_{BF}$ are flow metrics based on averages of long-term flow and precipitation in which no flow timing information is involved. Especially, RFDC_cal showed strong performance in reproducing $I_{BF}$ relative to PROX_reg. This result can be explained by considering that baseflow has less temporal variations than direct runoff in the Korean catchments under typical monsoonal climate. High seasonality of monsoonal precipitation

makes high temporal variations in direct runoff during June to September, while relatively steady baseflow is dominant during dry seasons (October to May). In Namgang Dam  whose flow variation is displayed in Figure 6, for example, the coefficient of variance (CV) of direct runoff was 5.86 for 2007-2015, which is approximately 3.5 times as high as that CV of baseflow.

On the other hand, RFDC_cal was poorer to reproduce $D_{RL}$ than PROX_reg. This highlights the weakness of RFDC_cal in which only flow magnitudes were used for identifying model parameters. PROX_reg showed better performance to predict $D_{RL}$ than RFDC_cal. Flow timing information gained from the observed hydrographs might be preserved, even after behavioural parameters were transferred to ungauged catchments. Overall, PROX_reg seems to be better than RFDC_cal to predict the three flow signatures together.

The box plots in Figure 9 provide an indication that $D_{RL}$ is likely to supplement the FDC calibration and thus improve RFDC_cal. From the collection of 50 behavioural parameter sets given by the FDC calibration, we chose the parameter set providing the lowest bias for each flow signature as the best behavioural sets, and simulated runoff again for all catchments. The high-flow predictability was fairly improved by additional constraining with $D_{RL}$, suggesting that flow metrics associated with flow timing make up for the weakness of the FDC calibration. Additional constraining with $R_{QP}$ and $I_{BF}$ did not bring appreciable improvement in the FDC calibration. However, PROX_reg was still better than the additional constraining with $D_{RL}$, indicating that a further study is needed for better constraining rainfall-runoff models using FDCs together with additional flow metrics.

**4.4 Paired t-tests between the modelling approaches**

For comparative evaluation in this study, we produced several runoff prediction sets using multiple rainfall-modelling approaches. First, we calibrated GR4J against the observed hydrographs (referred to as Q_cal), and transferred the behavioural parameters to ungauged catchments in the LOOCV mode (PROX_reg). We constrained GR4J with the regional FDCs (RFDC_cal). To evaluate equifinality, we recalibrated the GR4J parameters against the observed FDCs (referred to as "FDC_cal"). Additionally, we constrained the model with observed FDCs plus the flow signatures, and significant performance improvement was found with $D_{RL}$ (referred to as FDC+$D_{RL}$_cal). A paired t-test using the performance metrics (NSE, LNSE, or $D_{FS}$) between these modelling approaches can answer various questions beyond the graphical evaluations with box plots. For paired t-tests, we added one more case of transferring parameters gained from FDC_cal to ungauged catchments (referred to as FPROX_reg). FPROX_reg transfers behavioural parameters with no flow timing information from gauged to ungauged catchments. The mean NSE of FPROX_reg was 0.44 with a standard deviation of 0.49.

A primary hypothesis of this study was that RFDC_cal could outperform PROX_reg. This question can be addressed by NSE differences between RFDC_cal and PROX_reg. The mean NSE difference between them was -0.22 and the standard error was 0.051, providing an evaluation that the NSE differences were less than zero at a 95% confidence level. The paired t-test did not lend support the hypothesis (i.e., PROX_reg outperformed RFDC_cal significantly). However, we could

assume that $D_{RL}$ could improve predictive performance of FDC_cal. The mean NSE difference between FDC+DRL_cal and
370   FDC_cal was 0.12 and the standard error was 0.025, confirming the significance at a 95% confidence level.

Likewise, we tested several questions relevant to rainfall-runoff modelling in ungauged catchments using different combinations. In Table 3, we summarised the results of paired t-tests for scientific questions that may arise from this study. One interesting question would be "Did the behavioural parameters from Q_cal contain flow timing information for ungauged catchments?" We addressed this question by comparing between PROX_reg and FPROX_reg with a hypothesis
375   that predictability in ungauged catchments would decrease if the regionalised parameters were gained only from flow magnitudes. FPROX_reg uses FDC_cal for searching behavioural parameters at gauged catchments; thereby, it cannot transfer flow timing information to ungauged catchments through the behavioural parameters. The mean NSE difference between PROX_reg and FPROX_reg was 0.10, and the standard error was 0.031. The NSE differences were greater than zero significantly. The behavioural parameters from Q_cal were likely to have flow timing information affecting
380   predictability in ungauged catchments.

**5 Discussion and conclusions**

**5.1 RFDC_cal for rainfall-runoff modelling in ungauged catchments**

The use of regional FDCs as a single calibration criterion appears to be a good choice for searching behavioural parameters
385   in ungauged sites. As discussed earlier, the FDC is a compact representation of runoff variability at all time scales, and thus able to embed multiple hydrological features in catchment dynamics (Blöschl et al., 2013). A pilot study of Yokoo and Sivapalan (2011) discovered that the upper part of an FDC is controlled by interaction between extreme rainfall and fast runoff, while the lower part is governed by baseflow recession behaviour during dry periods. The middle part connecting the upper and the lower parts is related to the mean within year flow variations, which is controlled by interactions between
390   water availability, energy, and water storage (Yager et al., 2012; Yokoo and Sivapalan, 2011). It is well-documented that hydro-climatological processes within a catchment are reflected in the FDC (e.g., Cheng et al., 2012; Ye et al., 2012; Coopersmith et al., 2012; Yaeger et al., 2012; Botter et al., 2008), and therefore the model parameters identified solely by a regional FDC are expected to provide reliable predictions in ungauged catchments (e.g., Westerberg et al., 2014; Yu and Yang, 2000).
395   The comparative evaluation in this study, however, provides another expected lesson that the FDC calibration is good to reproduce the FDC itself, but it insufficiently captures functional responses of catchments due to the absence of flow timing information. A hydrograph is the most complete flow signature embedding numerous processes interacting within a catchment (Blöschl et al., 2013), being more informative than an FDC. Since any simplification of a hydrograph, including the FDC, should lose some amount of flow information, it is no surprise that the FDC calibration worsens the equifinality.
400   This study emphasises that the absence of flow timing in RFDC_cal may cause larger prediction errors than regionalised

parameters gained against observed hydrographs. The paired t-test between PROX_reg and FPROX_reg highlights that regionalised parameters gained from observed hydrographs were likely to contain intangible flow timing information even for ungauged catchments. The flow timing information implicitly transferred to ungauged catchment is a major gap between PROX_reg and RFDC_cal. The errors introduced by the FDC regionalisation were not significant due to high performance of the geostatistical method in this study.

Because the hydrograph calibration can compensate the errors in input-output data, one may convert the hydrograph into the FDC to avoid effects of disinformation on rainfall-runoff modelling. However, in this case, valuable flow timing information should be paid in trade-off. For RFDC_cal in this study, we began with converting the observed hydrographs into the flow quantiles to regionalise them; thus, the flow timing information was initially lost. As shown, the performance of RFDC_cal was generally lower than that of PROX_reg. Therefore, when condensing observed hydrographs into flow signatures, preserving all available flow information in the hydrograph would be a key for a successful rainfall-runoff modelling. This study shows that only using regionalised FDCs could lead to less reliable rainfall-runoff modelling in ungauged catchments than regionalised parameters. An FDC is unlikely to preserve all flow information in a hydrograph necessary for rainfall-runoff modelling.

**5.2 Suggestions for improving RFDC_cal**

Westerberg et al. (2014) suggested the necessity of further constraining to reduce predictive uncertainty in RFDC_cal. This study found that RFDC_cal could provide comparable performance to regenerate the flow signatures within which flow magnitudes are only involved (i.e., $R_{QP}$ and $I_{BF}$). To supplement regional FDCs, flow signatures associated with flow timing seems to be essential. Figure 9 shows potential of additional constraining with $D_{RL}$, and Q2 in Table 3 confirms it. Other flow signatures in temporal dimensions such as the high- and the low-flow event durations in Westerberg and McMillan (2015) can be candidates to improve RFDC_cal. However, uncertainty in those flow signatures will be a challenge to build regional models for ungauged catchments (Westerberg et al., 2016).

An alternative method of RFDC_cal is to directly regionalise hydrographs to ungauged catchments (e.g., Viglione et al., 2013). In data-rich regions, topological proximity could better capture spatial variation of daily flows than rainfall-runoff modelling with regionalised parameters (Viglione et al., 2013). Although a dynamic model may be required for regionalising observed daily flows at an expensive computational cost, flow timing information would be contained in regionalised hydrographs. The parameter identification against the regional hydrographs may become a better approach than RFDC_cal and/or other signature-based calibrations.

**5.3 Limitations and future research directions**

There are caveats in our comparative evaluation. First, uncertainty in input-output data was not considered in our assessment. McMillan et al. (2012) reported typical ranges of relative errors in discharge data as 10-20% for medium to high flow and 50-100% for low flows. We assumed that quality of the discharge data was adequate. However, other methods objectively

considering uncertainty could better estimate model performance and the equifinality (e.g., Westerberg et al., 2011, 2014). Second, we used a conceptual runoff model with a fixed structure for all the catchments. Uncertainty from the model structure would vary across the study catchments; nevertheless, the structural uncertainty was not measured here. Our comparative assessment was based on the basic premise that modelling conditions should be fixed for all study catchments. Third, we compared RFDC_cal and PROX_reg in a region with sufficient data lengths and quality at gauged catchments. The lessons from this study may not be expandable to ungauged catchments under poor data availability. Finally, though the proximity-based parameter regionalisation was good for the Korean catchments, comparison between RFDC_cal and other regionalisation methods, such as the regional calibration and the similarity-based parameter transfer, may provide beneficial information for rainfall-runoff modelling in ungauged catchments. Comparative assessment between RFDC_cal and other parameter regionalisation using more sample catchments under diverse climates will provide more meaningful lessons.

We can could no longer hypothesise that the parameters gained against regionalised FDCs would perform sufficiently, because an FDC contains less information than a hydrograph (i.e., the absence of flow timing). For improving RFDC_cal, we suggested to supplement RFDC_cal with flow signatures in temporal dimensions. Then, a question should be addressed on how to make flow signatures more informative than (or equally informative to) hydrographs. It may be impossible only using flow signatures originated from hydrographs (e.g., mean annual flow, baseflow index, recession rates, FDCs, etc.). Combinations of those signatures are unlikely more informative than their origins (i.e., hydrographs), though it depends on how much disinformation is present in the observed flows. Future research topics may include finding new signatures that supplement hydrographs, and how to combine them with existing flow signatures for rainfall-runoff modelling in ungauged catchments.

**5.4 Conclusions**

While the rainfall-runoff modelling against regional FDCs appeared a good approach for prediction in ungauged catchments, this study highlights its weakness in the absence of flow timing information, which may cause poorer predictive performance than the simple proximity-based parameter regionalisation. The following conclusions are worth emphasising:

(1) For ungauged catchments in South Korea where spatial proximity well captured functional similarity between gauged catchments, the model calibration against regional FDCs is unlikely to outperform the conventional proximity-based parameter transfer for daily runoff prediction. The absence of flow timing information in regional FDCs seems to cause a substantial equifinality problem in the parameter identification process and thus lower predictability.

(2) The model parameters gained from observed hydrographs would contain flow timing information even for ungauged catchments. This intangible flow timing information should be discarded if one calibrates a rainfall-runoff model against regional FDCs. This information loss may reduce predictability in ungauged catchments significantly.

(3) To improve the calibration against regional FDCs, flow metrics in temporal dimensions, such as the rising limb density, need to be included as additional constraints. As an alternative approach, if river gauging density is high,

regionalised hydrographs preserving flow timing information can be used for local calibrations at ungauged catchments.

(4) For better predictions in ungauged catchments, it is necessary to find new flow signatures that can supplement the observed hydrographs. How to  combine them with existing information will be a future research topic for rainfall-runoff modelling in ungauged catchments.

**Acknowledgements**

This study was supported by APEC Climate Center. We send special thanks to Ms. Yoe-min Jeong and Dr. Hyungil Eum for the PRISM  data sets. We greatly appreciate constructive comments and suggestions from the reviewers that significantly improved the manuscript. Data  required to reproduce the modelling results  are available upon request from the authors (d.kim@apcc21.org, jachum@apcc21.org).

**Table 1: Summary of hydrological features of the study catchments**

| | Average | CV | minimum | 25% | median | 75% | Maximum |
|---|---|---|---|---|---|---|---|
| Area (km$^2$) | 890 | 1.39 | 57 | 208 | 495 | 1013 | 6705 |
| Elevation (m a.s.l.) | 339 | 0.63 | 39 | 193 | 255 | 495 | 996 |
| Mean annual prcp. (mm yr$^{-1}$) | 1359 | 0.14 | 1145 | 1247 | 1286 | 1388 | 1997 |
| Mean annual temp. (°C) | 11.9 | 0.13 | 7.9 | 11.3 | 12.3 | 13.0 | 13.8 |
| Aridity index[1] (-) | 0.66 | 0.11 | 0.44 | 0.61 | 0.68 | 0.71 | 0.76 |
| $P_{snow}$[2] | 35 | 0.66 | 6 | 23 | 28 | 50 | 141 |
| Mean annual flow (mm yr$^{-1}$) | 739 | 0.25 | 232 | 624 | 740 | 838 | 1159 |
| $R_{PQ}$ (-) | 0.55 | 0.27 | 0.18 | 0.45 | 0.54 | 0.63 | 0.91 |
| $I_{BF}$ (-) | 0.49 | 0.16 | 0.27 | 0.44 | 0.49 | 0.56 | 0.62 |
| $D_{RL}$ (day$^{-1}$) | 0.63 | 0.10 | 0.50 | 0.60 | 0.63 | 0.66 | 0.77 |

620 [1]Ratio of potential ET to total precipitation, [2]Percentage of snowfall to total precipitation. Climatological features were calculated using spatial averages of the grid data, while the flow metrics were from the daily hydrographs for 2007-2015 as explained in Section 3.6.

**Table 2: Ranges of GR4J parameters used for parameter calibration (Demirel et al., 2013)**

| Parameter | Range |
|---|---|
| X1 (mm) | 10 to 2000 |
| X2 (mm) | -8 to +6 |
| X3 (mm) | 10 to 500 |
| X4 (days) | 0.5 to 4.0 |

625

**Table 3: Results of the paired t-tests for potential questions on rainfall-runoff modelling in ungauged catchments**

| Questions | Corresponding pair | [1]PM | [2]$\overline{\Delta PM}$ | [3]std. err. | Answer |
|---|---|---|---|---|---|
| Q1. Did RFDC_cal outperform PROX_reg? | RFDC_cal – PROX_reg | NSE | -0.22 | 0.051 | No[*] |
| Q2. Did $D_{RL}$ improve FDC_cal? | FDC+DRL_cal – FDC_cal | NSE | 0.12 | 0.025 | Yes[*] |
| Q3. Did parameters from Q_cal contain flow timing information for ungauged catchments? | PROX_reg – FPROX_reg | NSE | 0.10 | 0.031 | Yes[*] |
| Q4. Did absence of flow timing affect model efficiency? | Q_cal – FDC_cal | NSE | 0.23 | 0.026 | Yes[*] |
| Q5. Did PROX_reg outperform RFDC_cal in predicting low flows? | PROX_reg – RFDC_cal | LNSE | 0.09 | 0.031 | Yes[*] |
| Q6. Did PROX_reg outperform RFDC_cal in reproducing $I_{BF}$? | PROX_reg – RFDC_cal | $D_{FS}(I_{BF})$ | 0.06 | 0.028 | Unlikely |
| Q7. Did errors in regional FDCs affect RFDC_cal significantly? | RFDC_cal – FDC_cal | NSE | -0.09 | 0.069 | Unlikely |

[1]Performance metric used for t-test, [2]Mean PM difference between the corresponding pair, [3]Standard error of ΔPM. [*]ΔPM is significantly different from zero. The significance was evaluated at 95% confidence levels.

630

[Figure]

**Figure 1: Locations of the study catchments in South Korea. The numbers are labelled at the outlet of each catchment.**

[Figure]

635     **Figure 2: The schematised structure of GR4J (X1-X4: model parameters, PE: potential evapotranspiration, P: precipitation, Q: runoff, other letters indicate variables conceptualising internal catchment processes).**

[Figure]

**Figure 3:** (a) box plots of high flow (NSE) and low flow (LNSE) reproducibility of the behavioural parameters obtained from the hydrograph calibration at the 45 catchments, (b) the relationship between the input-output consistency and the model performance. The straight lines in the box plots connect the performance metrics for the calibration (2011-2015) and the validation periods (2007-2010) in each catchment.

645

[Figure]

**Figure 4: 1:1 scatter plot between the empirical flow quantiles and the flow quantiles predicted by the top-kriging FDC regionalisation method.**

[Figure]

**Figure 5: Box plots of NSE and LNSE values between the observed and the predicted hydrographs by RFDC_cal and PROX_reg for the 45 catchments under the cross validation mode.**

[Figure]

**Figure 6: The observed and predicted hydrographs, the prediction areas, and the observed and predicted FDCs given by (a) the hydrograph calibration and (b) the FDC calibration for Namgang Dam (<s>the</s> Catchment 2 in Figure 1).**

[Figure]

**Figure 7: The input-output consistency vs. equifinality increased by replacing the hydrograph calibration with the FDC**
665 **calibration. The equifinality ratio is defined as the ratio between the prediction areas of the 50 behavioural parameters gained from the FDC calibration and the hydrograph calibration.**

[Figure]

670 **Figure 8: Flow signature reproducibility comparison between RFDC_cal and PROX_reg in terms of $R_{QP}$ (a), $I_{BF}$ (b), and $D_{RL}$ (c).**

[Figure]

**Figure 9: Predictive performance of the FDC calibrations additionally conditioned by $R_{QP}$ (FDC+RQP), $I_{BF}$ (FDC+IBF), and $D_{RL}$**
675 **(FDC+DRL) in comparison to the other modelling approaches. Q_cal and FDC_cal refer to the hydrograph and the FDC calibration in gauged catchments respectively. 38 catchments with positive NSEs for all the modelling approaches were used in the box-plots.**